# Near-Optimal Randomized Exploration for Tabular Markov Decision Processes

**Zhihan Xiong**[1][*]
zhihanx@cs.washington.edu

**Ruoqi Shen**[1][*]
shenr3@cs.washington.edu

**Qiwen Cui**[1][*]
qwcui@cs.washington.edu

**Maryam Fazel**[2]
mfazel@uw.edu

**Simon S. Du**[1]
ssdu@cs.washington.edu

[1] Paul G. Allen School of Computer Science & Engineering, University of Washington
[2] Department of Electrical & Computer Engineering, University of Washington

## Abstract

We study algorithms using randomized value functions for exploration in reinforcement learning. This type of algorithms enjoys appealing empirical performance. We show that when we use 1) a single random seed in each episode, and 2) a Bernstein-type magnitude of noise, we obtain a worst-case $\widetilde{O}\left(H\sqrt{SAT}\right)$ regret bound for episodic time-inhomogeneous Markov Decision Process where $S$ is the size of state space, $A$ is the size of action space, $H$ is the planning horizon and $T$ is the number of interactions. This bound polynomially improves all existing bounds for algorithms based on randomized value functions, and for the first time, matches the $\Omega\left(H\sqrt{SAT}\right)$ lower bound up to logarithmic factors. Our result highlights that randomized exploration can be near-optimal, which was previously achieved only by optimistic algorithms. To achieve the desired result, we develop 1) a new clipping operation to ensure both the probability of being optimistic and the probability of being pessimistic are lower bounded by a constant, and 2) a new recursive formula for the absolute value of estimation errors to analyze the regret.

## 1 Introduction

This paper concerns learning in tabular Markov Decision Processes (MDP), arguably the most fundamental model for reinforcement learning (RL). Existing algorithms that achieve the near-optimal minimax $\widetilde{O}\left(H\sqrt{SAT}\right)$ regret bound are based on the principle of *Optimism in the face of Uncertainty* (OFU), such as upper confidence bound (UCB) [7, 53, 16, 57, 55].[1] Here $S$ is the number of states, $A$ is the number of actions, $H$ is the planning horizon, and $T$ is the total number of interactions between the agent and the environment.

---

[*]Equal Contribution
[1]This bound is for time-inhomogeneous MDP with each reward bounded by 1 and $T$ is sufficiently large.

36th Conference on Neural Information Processing Systems (NeurIPS 2022).

Another broad category is algorithms with randomized exploration such as Thompson Sampling [36, 5, 39]. These algorithms inject (carefully tuned) random noise to value function to encourage exploration. UCB-type algorithms enjoy well-established theoretical guarantees but suffer from difficult implementation since an upper confidence bound is usually infeasible for many practical models like neural networks. Instead, practitioners prefer randomized exploration such as noisy networks in [19], and algorithms with randomized exploration have been widely used in practice [37, 13, 11, 35]. However, how to design randomized exploration algorithms in a principled way and perform randomized exploration optimally is far from clear. While randomized exploration can have great performance in practice, theoretically, the best known worst-case regret bound for algorithms with randomized exploration is $\widetilde{O}\left(H^2 S\sqrt{AT}\right)$ [2], which is far worse than that of the UCB-type algorithms. In this paper, we introduce a new randomized exploration algorithm and show it enjoys a near-optimal $\widetilde{O}\left(H\sqrt{SAT}\right)$ worst-case regret bound, thus closing the gap. Our work sheds new light on randomized exploration on both the algorithmic side and the theoretical side.

**Our Contributions.**     Our contributions are summarized below:

• We propose a new algorithm, Single Seed Randomization (SSR), which incorporates a crucial algorithmic idea: using a single random seed for the entire episode, in contrast to previous methods of randomized exploration which use one seed for each time step. SSR is able to explore more efficiently than previous methods by avoiding having noise at different time steps canceling with each other. Theoretically, we show, thanks to this new idea, if one uses a Hoeffding-type magnitude of noise, SSR achieves an $\widetilde{O}\left(H^{1.5}\sqrt{SAT}\right)$ regret bound, improving upon the best existing result on randomized exploration algorithm [2].

• We further design a new Bernstein-type magnitude of noise for our algorithm, and achieve an $\widetilde{O}\left(H\sqrt{SAT}\right)$ regret bound, resolving an open problem raised in [2]. To our knowledge, this is the first time that a Bernstein-type bound is used in randomized exploration. More importantly, our upper bound matches the $\Omega\left(H\sqrt{SAT}\right)$ minimax lower bound up to logarithmic factors.

We note that our goal is not to show randomized exploration is better than optimistic algorithms [7] in the tabular setting. Instead, we aim to provide a solid theoretical understanding of a practically relevant algorithm. Indeed, understanding randomized exploration itself is an important theoretical research direction and has attracted much interest in the community [3, 4, 6, 41, 52, 48, 2, 36, 39, 37, 35].

**Main Challenge and Technical Overview.**     Besides the aforementioned algorithmic ideas (single random seed and Bernstein-type magnitude of noise), we also need additional ideas in analysis to prove the desired regret bound. The main challenge is that unlike UCB-type algorithms, the estimated value in algorithm with randomized exploration, is not an upper bound of the true optimal value. This leads to the failure of directly utilizing their analysis, which only need to analyze the one-sided error in estimation. We instead work on the *absolute value* of the estimation error, whose analysis is more complicated than that for the one-sided error in UCB-type algorithms. Working with absolute value forces us to ensure that both the probability that the estimated value is optimistic and the probability that the estimated value is pessimistic are lower bounded. However, the clipping strategy in existing algorithm cannot maintain pessimism. To tackle with this issue, we develop a new clipping method. Below we list our technical contributions.

**1.** First, we propose a new clipping strategy to constrain the estimated value function (cf. Eqn. (4)). Previous clipping strategies in [52, 2] are based on uncertainty and can only maintain optimism. Our clipping strategy directly works on the value function, which is similar to those used in UCB-type algorithms [7, 22, 57]. Our clipping strategy can maintain both the optimism and pessimism. In addition, the number of times that the clipping is used can still be bounded.

**2.** Second, we prove that the single seed randomization ensures that the estimated value function can both be optimistic or pessimistic with constant probability at all states and timesteps. This is stronger than previous randomized exploration algorithms that are only shown to be optimistic at the initial state with constant probability. With this property, we can then bound the difference between the optimal value function and estimated value function from both above and below, which results in a bound on its absolute value. See Section 5.1, Appendix C and Appendix D.

**3.** Third, we prove a novel recursion argument on the absolute value of the policy estimation error. As mentioned in [2], the recursion in UCB-type algorithms can not be directly utilized because our estimated value function is not a high-probability upper bound of the true optimal value function. With the bound of absolute value, we are able to prove new recursion formulas and together we can control the policy estimation error. See Section 5.2 and Appendix E.

**4.** At last, we bound the sum of variance in a novel manner. In [7], the UCB-type estimation guarantees that the policy estimation error is always positive so the difference of the variance can be directly bounded. We generalize the argument to the absolute value of the estimation error to bound the sum of variance. See Section 5.3.1 and Appendix G.

## 2 Related Work

In this section we review existing provably efficient algorithms for tabular MDP. There is a long list of sample complexity guarantees for tabular MDP [27, 9, 25, 43, 44, 28, 8, 21, 45, 30, 36, 14, 7, 15, 38, 5, 22, 20, 46, 16, 18, 42, 41, 54, 12, 56, 51, 40, 34, 55, 49, 2, 41, 5, 17, 33, 31]. The state-of-the-art methods are based on upper confidence bound (UCB) [7, 53, 16, 57, 55, 33, 31]. For the setting considered in this paper where the transition is time-inhomogeneous and the reward is bounded by $1$, one can achieve an $\widetilde{O}\left(H\sqrt{SAT}\right)$ in the regime where $T$ is sufficiently large.

Algorithms with randomized exploration have been proved to enjoy favorable regret bounds in bandit problems [29, 3, 26, 10, 4]. In certain settings, randomized exploration can match the worst-case regret bound of UCB-based approaches and achieve nearly minimax optimal regret bounds [24, 4]. However, for RL, existing theory for randomized exploration are far from optimal [2, 41, 5, 50, 52]. For the setting considered in this paper, the sharpest existing regret bound among algorithms with randomized exploration is $\widetilde{O}\left(H^2 S\sqrt{AT}\right)$ proved in [2]. Our paper closes this gap and thus deepens our understanding about randomized exploration.

## 3 Preliminaries

We consider time-inhomogeneous finite-horizon MDP $M = (H, \mathcal{S}, \mathcal{A}, P, R, s_1)$, where $|\mathcal{S}| = S$ and $|\mathcal{A}| = A$. Here, $\mathcal{S} = \{1, ..., S\}$ is the finite state space. $\mathcal{A} = \{1, ..., A\}$ is the finite action space. $H$ is the length of an episode. For convenience, we take $s_1$ to be the fixed initial state, although a more general initial distribution will not change the conclusion. $P : \mathcal{S} \times \mathcal{A} \times [H] \to \triangle(\mathcal{S})$ is the transition function, where if the agent stays at state $s$ and takes action $a$ at time $h$, it transits to state $s'$ with probability $P_{h,s,a}(s') \in [0,1]$. $R : \mathcal{S} \times \mathcal{A} \times [H] \to [0,1]$ is the reward function, where if the agent stays at $s$ and takes action $a$ at time $h$, it will receive reward $r_{h,s,a} \in [0,1]$ such that $\mathbb{E}[r_{h,s,a}] = R_{h,s,a}$.

A deterministic policy for such a MDP is defined as a tuple $\pi = (\pi_1, \ldots, \pi_H)$, where $\pi_h : \mathcal{S} \mapsto \mathcal{A}$. The associated value function at state $s \in \mathcal{S}$ and level $h \in \{1, \ldots, H\}$ is recursively defined as

$$V_h^\pi(s) = R_{h,s,\pi_h(s)} + \sum_{s' \in \mathcal{S}} P_{h,s,\pi_h(s)}(s') V_{h+1}^\pi(s').$$

For convenience, we set $V_{H+1}^\pi = \mathbf{0} \in \mathbb{R}^S$. The corresponding optimal value function is $V_h^*(s) = \max_{\pi \in \Pi} V_h^\pi(s)$, where $\Pi$ is the set of all possible deterministic policies. For a particular algorithm Alg, let $\pi^k$ denote the policy that Alg employs during episode $k$. Then, the regret of running Alg on MDP $M$ for $K$ episodes is defined as

$$\text{Reg}(M, K, \text{Alg}) = \sum_{k=1}^{K} \left(V_1^*(s_1) - V_1^{\pi^k}(s_1)\right). \tag{1}$$

Note that the regret, $\text{Reg}(M, K, \text{Alg})$, is a random variable due to randomness in state transition and the algorithm, Alg. In this paper, we show the regret of our proposed algorithm can be upper bounded with high probability, and the upper bound matches the known lower bound up to logarithmic factors.

To facilitate our later analysis, we introduce some notations for empirical estimation. At episode $k$, we collect a trajectory $(s_1^k, a_1^k, r_1^k, \cdots, s_H^k, a_H^k, r_H^k)$ as specified in Algorithm 1. Let $n_k(h, s, a) =$

---

**Algorithm 1:** Single Seed Randomization (SSR)

---

**Input:** $H, S, A$, perturbation type $\text{ty} \in \{\text{Ho}, \text{Be}\}$

1 **for** *episodes* $k = 1, 2, \ldots, K$ **do**
2      Sample $\hat{z}_k \sim \mathcal{N}(0, 1)$
3      Define terminal value function $\overline{Q}_{H+1,k} = \mathbf{0} \in \mathbb{R}^{SA}$ and $\overline{V}_{H+1,k} = \mathbf{0} \in \mathbb{R}^S$
4      **for** *time periods* $h = H, \ldots, 1$ **do**
5          $\overline{Q}_{h,k}(s, a) \leftarrow \hat{R}_{h,s,a}^k + \left\langle \hat{P}_{h,s,a}^k, \overline{V}_{h+1,k} \right\rangle + \sigma_{\text{ty}}^k(h, s, a)\, \hat{z}_k;$       `// ` $\sigma_{\text{ty}}^k(h,s,a)$ `is`
           `defined in (5) and (6).`
6          Define $\overline{V}_{h,k}(s) = \text{clip}_{2(H-h+1)}\left(\max_{a \in \mathcal{A}} \overline{Q}_{h,k}(s,a)\right)$ for all $s \in \mathcal{S}$
7      **end**
8      Agent takes actions $a_h^k = \text{argmax}_{a \in \mathcal{A}} \overline{Q}_{h,k}(s_h^k, a)$ throughout this episode
9      Observe data $s_1^k, a_1^k, r_1^k, \ldots, s_H^k, a_H^k, r_H^k$ and compute $\hat{R}_{h,s,a}^{k+1}, \hat{P}_{h,s,a}^{k+1}$ and $n_{k+1}(h, s, a)$ for
        all $(h, s, a) \in [H] \times \mathcal{S} \times \mathcal{A}$
10 **end**

---

$\sum_{l=1}^{k-1} \mathbb{1}\{(s_h^l, a_h^l) = (s, a)\}$ be the number of times action $a$ is taken at state $s$ and time $h$ before episode $k$, where $\mathbb{1}\{\cdot\}$ is the indicator function. We define

$$\hat{R}_{h,s,a}^k = \frac{\sum_{l=1}^{k-1} \mathbb{1}\{(s_h^l, a_h^l) = (s, a)\} r_{h, s_h^l, a_h^l}^l}{n_k(h, s, a) + 1}, \tag{2}$$

$$\hat{P}_{h,s,a}^k(s') = \frac{\sum_{l=1}^{k-1} \mathbb{1}\{(s_h^l, a_h^l, s_{h+1}^l) = (s, a, s')\}}{n_k(h, s, a) + 1}. \tag{3}$$

Then, define empirical MDP based on our observation and estimation before episode $k$ as the tuple $\hat{M}^k = (H, \mathcal{S}, \mathcal{A}, \hat{P}^k, \hat{R}^k, s_1)$. Since $\hat{P}_{h,s,a}^k$ is not a valid distribution over $\mathcal{S}$, for being rigorous, we can imagine there is an additional virtual absorbing state that every state will transit to with remaining probability.

In addition to the above notations, let $\widetilde{O}(\cdot)$, $\widetilde{\Theta}(\cdot)$ and $\widetilde{\Omega}(\cdot)$ be asymptotic notations ignoring all poly-logarithmic terms. For distribution $D \in \Delta^{\mathcal{S}}$ and value function $V \in \mathbb{R}^S$, let $\mathbb{V}(D, V)$ denote the variance of $V$ under distribution $D$, which is defined as $\mathbb{V}(D, V) = \sum_{s \in \mathcal{S}} D(s)(V(s) - \langle D, V \rangle)^2$. For constant $a > 0$, we define the corresponding clipping function as $\text{clip}_a(\cdot) = \max\{-a, \min\{a, \cdot\}\}$. Immediately we have $|\text{clip}_a(x)| \le a$ for any $a > 0$. We introduce the definitions of other notations when used. In appendix, we summarize the notations and definitions used in this paper.

## 4 Main Results

### 4.1 Algorithm

The main contribution of this paper is that we show algorithm with randomized value functions can achieve regret that matches the known lower bound $\Omega\left(H\sqrt{SAT}\right)$ [21, 17] up to logarithmic factors in the tabular setting. To facilitate exploration, this type of algorithms uses random value perturbation instead of deterministic bonus. The algorithm we consider is summarized in Algorithm 1. In our algorithm, SSR, the random perturbation ensures that optimism/pessimism can be obtained with constant probability in each episode. Moreover, randomized value function has its origin from posterior sampling for reinforcement learning (Thompson sampling). The randomized perturbation can be interpreted as approximate sampling from the posterior distribution of the value function on randomized training data [41].

We first give an overview of SSR. In Algorithm 1, the policy used at episode $k$ is computed using the empirical MDP, $\hat{M}^k = (H, \mathcal{S}, \mathcal{A}, \hat{P}^k, \hat{R}^k, s_1)$, which is based on observation and estimation before episode $k$. However, instead of directly choosing optimal policy for $\hat{M}^k$, we add a small random perturbation when computing the value of each state and action pair. To be more precise, at each

episode $k$, we first estimate the reward and transition function for each state $s$ and action $a$ based on (2) and (3). Then, we compute the value function for state $s$ and action $a$,

$$\overline{Q}_{h,k}(s,a) \leftarrow \hat{R}^k_{h,s,a} + \left\langle \hat{P}^k_{h,s,a}, \overline{V}_{h+1,k} \right\rangle + \sigma^k_{\text{ty}}(h,s,a)\,\hat{z}_k.$$

Here, $\hat{z}_k \sim \mathcal{N}(0,1)$ is a standard Gaussian random variable sampled once every episode. The magnitude of the perturbation, $\sigma^k_{\text{ty}}$ depends on how many samples $n_k(h,s,a)$ we have observed and how confident we are on the estimations $\hat{R}^k_{h,s,a}$ and $\hat{P}^k_{h,s,a}$. We will discuss more about the choice of the magnitude later in this section.

In order to prevent estimated value function from behaving badly, we add a clipping to the value function:

$$\overline{V}_{h,k}(s) = \text{clip}_{2(H-h+1)}\left(\max_{a\in\mathcal{A}}\overline{Q}_{h,k}(s,a)\right) \tag{4}$$

As our analysis will show, this kind of clipping can bound the value function, maintain optimism and pessimism and also guarantee that clipping will not happen for a lot of times. The constant 2 (instead of 1) plays a crucial role because it means the value function grows at an additive rate of 2 from $h = H$ to $h = 1$. If we do not consider the added noise, then the value function should at most grow 1 at each timestep because the reward is at most 1. For our clipping technique, if a clip is triggered, there exists a timestep such that the added noise is more than 1, which is equivalent to a small number of visits (cf. Definition 23 and Lemma 8). As our later analysis will show, the clipping only affects the lower-order term and will not compromise the long-term performance of the algorithm. Finally, after computing the value function and clipping, SSR chooses the action $a^k_h$ that maximizes $\overline{Q}_{h,k}(s^k_h, a)$ at each time step, $h = 1, ..., H$, throughout the episode.

Note that from a Bayesian perspective, when there is no clipping, in Algorithm 1, $\overline{Q}_{h,k}$ follows distribution

$$\overline{Q}_{h,k}(s,a) \mid \overline{V}_{h+1,k} \sim \mathcal{N}\left(\hat{R}^k_{h,s,a} + \left\langle \hat{P}^k_{h,s,a}, \overline{V}_{h+1,k} \right\rangle, \left(\sigma^k_{\text{ty}}(h,s,a)\right)^2\right).$$

This resembles posterior sampling because when estimating some parameter $\theta^* \sim \mathcal{N}\left(0, \beta^2\right)$ based on noisy observations $\theta_1, ..., \theta_n \sim \mathcal{N}(\theta, \beta^2)$, the posterior distribution of $\theta^*$ given $\{\theta_i\}_{i=1}^n$ is $\theta^* \mid \{\theta_i\}_{i=1}^n \sim \mathcal{N}\left(\frac{1}{n+1}\sum_{i=1}^n \theta_i, \frac{\beta^2}{n+1}\right)$. Although exact posterior sampling may not be possible in complex reinforcement learning settings, in SSR, $\sigma^k_{\text{ty}}(h,s,a)$ is chosen at scale $\widetilde{\Theta}\left(1/\sqrt{n_k(h,s,a)}\right)$ and therefore can be interpreted as doing approximate posterior sampling. Moreover, SSR can be viewed as a variant of Randomized Least Square Value Iteration (RLSVI). The major differences are at the clipping function and a single random seed used in each episode instead of different random seeds at different tuples $(h,s,a)$. We will discuss more about the choice of the random seed later in this section. We refer to [37] and [41] for a more detailed discussion on the relationship among RLSVI, posterior sampling and randomized value function.

In the following paragraphs, we discuss in more details about the three major algorithmic innovations:

**Single Random Seed in Each Episode.** SSR is similar to the algorithms analyzed in [41] and [2]. The major difference is that in the algorithm we propose, we use a single random seed $\hat{z}_k$ to generate the perturbations for all time steps $h = 1, ..., H$ in an episode $k$.

When using different random seeds in an episode, the algorithm can be optimistic in some time step while being pessimistic in others. Then, the effects of the perturbations at different time steps will cancel with each other. As a result, to ensure sufficient exploration, the magnitude of the perturbation has to large. This issue was also pointed out in [2, 1].

A large perturbation magnitude can increase the instability of the algorithm and worsen the algorithm's performance. When a single random seed is used, a small perturbation magnitude is enough to guarantee that the algorithm is optimistic with constant probability in any episode. We are able to show that using a single random seed can significantly increase the stability of the algorithm and therefore enjoy much smaller regret. Coincidentally, [48] also uses a similar single randomization in bandit problems to build a near-optimal randomized exploration algorithm and our work can be treated as its natural extension to RL problems.

**Clipping.** To obtain a tight regret bound, the estimated value function needs to be well bounded. In [41], no clipping is used and the estimated value function is at the order of $\widetilde{O}(H^{5/2}S)$, which results in a suboptimal regret bound. Generally there are two types of clipping methods. The first one is uncertainty-based, i.e. the value is clipped to $H - h + 1$ at timestep $h$ whenever the uncertainty is large [52, 2]. However this type of clipping cannot maintain pessimism which is critical in our analysis. The other kind of clipping is value-based, mostly in UCB-type algorithms [23]. These algorithms truncate estimated value greater than a certain threshold, i.e. $H - h + 1$ at time step $h$. The problem here is that the number of clippings cannot be bounded because if the true value function is close to $H - h + 1$ at timestep $h$, the clipping will happen with some constant probability.

Our clipping method leverages both type of clipping methods in the existing literature. Though our clipping is based on the value function, we show that whenever the clip is triggered, the estimation error must be large, which implies that the uncertainty at that state is large. This clipping method inherits the desired properties from both uncertainty-based and value-based clipping, i.e. the optimism/pessimism is maintained and the number of clippings can be bounded.

**Magnitude of Perturbation.** A large magnitude of perturbation can encourage exploration, but at the same time increase instability. In our algorithm, the magnitudes are chosen as the smallest values so that the algorithm can be optimistic with constant probability. Since the value function can roughly be bounded by $O(H)$, a naive choice of the perturbation magnitude can be $\Theta\left(H/\sqrt{n_k(h,s,a)}\right)$. In this way, by Hoeffding's inequality, as long as the random Gaussian variable sampled $\hat{z}_k$ is bigger than a constant, which happens with constant probability, the estimated value function will be optimistic. By similar reasoning, we can see that the estimated value function will also be pessimistic with constant probability.

To make the magnitude even smaller, inspired by [7] who showed one can use an (empirical) Bernstein's inequality to derive a sharp exploration bonus for UCB-based algorithms, we propose a new choice of perturbation magnitude based on Bernstein's inequality. The Bernstein-based perturbation uses the empirical variance of the value function, which makes it smaller than the Hoeffding-based one mostly, but still maintains optimism with constant probability.

In our paper, we study both types of magnitudes. In particular, we show that the regret of SSR based on Bernstein's inequality matches the known lower bound $\Omega\left(H\sqrt{SAT}\right)$. Following are the two choices:

$$\sigma_{\mathrm{Ho}}^k(h,s,a) = H\sqrt{\frac{\log\left(2HSAk^2\right)}{n_k(h,s,a)+1}} + \frac{H}{n_k(h,s,a)+1}, \tag{5}$$

$$\sigma_{\mathrm{Be}}^k(h,s,a) = \sqrt{\frac{16\mathbb{V}\left(\tilde{P}_{h,s,a}^k, \overline{V}_{k,h+1}\right)\log\left(2HSAk^2\right)}{n_k(h,s,a)+1}} + \frac{65H\log\left(2HSAk^2\right)}{n_k(h,s,a)+1} + \sqrt{\frac{\log\left(2HSAk^2\right)}{n_k(h,s,a)+1}}, \tag{6}$$

where subscript "Ho" represents that the perturbation is based on Hoeffding's inequality and "Be" represents Bernstein's inequality, correspondingly. Here, for proof convenience, $\tilde{P}_{h,s,a}^k$ is defined by replacing the denominator in $\hat{P}_{h,s,a}^k$ by $\max\{n_k(h,s,a),1\}$. To clarify, when subscript "ty" is used, which stands for "type" as a placeholder for "Ho" or "Be", it means that there is no need to write two copies of expressions for Hoeffding-based and Bernstein-based noises separately.

**Practical Considerations.** Here, we explain why randomized exploration is widely used in practice and why our algorithmic formulation practically has advantage over UCB-type algorithms. In randomized exploration, there are usually two important components: (1) the algorithm (e.g., Algorithm 1) and (2) the noise magnitude ($\sigma_{\mathrm{ty}}$). In practice, the main advantage of randomized exploration lies in the algorithm component. The generalization from the tabular setting to the function approximation setting is straightforward: one can just add a random regularization term in the value estimation step, whose details can be found in [35]. On the other hand, the generalization of optimistic algorithms from the tabular setting to the function approximation setting is more non-trivial because it often requires an explicit construction of the confidence set. For the second component, although generalizing our strategy of tuning noise magnitude to the real-world function approximation setting is indeed not straightforward, it is often set as a hyper-parameter in practice.

## 4.2 Regret Analysis

We analyze the regret, defined in (1), of our algorithm SSR using both types of perturbations. Our main theorems are presented in Theorem 1 and 2. In particular, Theorem 2 shows SSR with Bernstein-based perturbation can achieve the regret that matches the known lower bound $\Omega\left(H\sqrt{SAT}\right)$ up to logarithmic factors. We sketch the proof of Theorem 1 and Theorem 2 in Section 5.

**Theorem 1.** *If the Hoeffding-type noise* (5) *is used, then for any MDP* $M = (H, \mathcal{S}, \mathcal{A}, P, R, s_1)$, *with probability at least* $1 - \delta$, *Algorithm 1 satisfies*

$$\text{Reg}(M, K, \textsf{SSR}_{\text{Ho}}) \leq \widetilde{O}\left(H^{1.5}\sqrt{SAT} + H^4 S^2 A\right).$$

*In particular, when* $T \geq \widetilde{\Omega}\left(H^5 S^3 A\right)$, *it holds that* $\text{Reg}(M, K, \textsf{SSR}_{\text{Ho}}) \leq \widetilde{O}\left(H^{1.5}\sqrt{SAT}\right)$.

**Theorem 2.** *If the Bernstein-type noise* (6) *is used, then when* $T \geq \widetilde{\Omega}\left(H^5 S^2 A\right)$, *for any MDP* $M = (H, \mathcal{S}, \mathcal{A}, P, R, s_1)$, *with probability at least* $1 - \delta$, *Algorithm 1 satisfies*

$$\text{Reg}(M, K, \textsf{SSR}_{\text{Be}}) \leq \widetilde{O}\left(H\sqrt{SAT} + H^4 S^2 A\right).$$

*In particular, if we further have* $T \geq \widetilde{\Omega}\left(H^6 S^3 A\right)$, *it then holds that* $\text{Reg}(M, K, \textsf{SSR}_{\text{Be}}) \leq \widetilde{O}\left(H\sqrt{SAT}\right)$.

We give a brief comparison between SSR and other related works. [41] shows that RLSVI, an algorithm similar to SSR, can achieve $\tilde{O}\left(H^{2.5}S^{1.5}\sqrt{AT}\right)$ regret in expectation over the randomness of MDP and the algorithm. In [2], an improved high probability regret bound $\widetilde{O}\left(H^2 S\sqrt{AT}\right)$ is proposed, which is the sharpest bound for randomized algorithms prior to this work. Our paper closes the gap between those previous bounds and the lower bound in tabular setting.

We also run numerical simulations to empirically compare SSR and RLSVI in the deep-sea environment, which is commonly used as a benchmark to test an algorithm's ability to explore. The results show that SSR significantly outperforms RLSVI as predicted by our regret analysis. More details about our experiment can be found in Appendix J.

## 5 Proof Outline

In this section, we present an proof outline of Theorem 1 and 2. Since their proofs follow the same framework, we will present an unified outline and explain the individual steps particularly for each case when necessary. The details of complete proof are deferred to the appendix.

**Notation**   For the ease of exposition, we will use a simplified notations during this sketch. Specifically, let $x = (h, s, a)$ and $x_h^k = (h, s_h^k, a_h^k)$.

### 5.1 Concentration and Optimism/Pessimism

We start by introducing a set of MDPs $\mathcal{M}_{\text{ty}}^k$ as a confidence set such that the empirical MDP $\hat{M}^k$ belongs to it with high probability, meaning that we have a good estimation of the true MDP. Specifically, with $M' = (H, \mathcal{S}, \mathcal{A}, P', R', s_1)$, we define

$$\mathcal{M}_{\text{ty}}^k := \left\{ M' : \forall x = (h, s, a), \left|(R_x' - R_x) + \left\langle P_x' - P_x, V_{h+1}^* \right\rangle\right| \leq \sqrt{e_{\text{ty}}^k(x)} \right\},$$

where $\sqrt{e_{\text{Ho}}^k(x)} = \sigma_{\text{Ho}}^k(x)$ and $\sqrt{e_{\text{Be}}^k(x)} \approx \sigma_{\text{Be}}^k(x)$.

Define the event $\mathcal{C}_{\text{ty}}^k := \left\{\hat{M}^k \in \mathcal{M}_{\text{ty}}^k\right\}$. Then, by applying Hoeffding's inequality or Bernstein's inequality, for both types of perturbation, it is possible to show that

$$\sum_{k=1}^{\infty} \mathbb{P}\left(\left(\mathcal{C}_{\text{ty}}^k\right)^c\right) = \sum_{k=1}^{\infty} \mathbb{P}\left(\hat{M}^k \notin \mathcal{M}_{\text{ty}}^k\right) \leq \frac{\pi^2}{3}.$$

Since the value function is bounded in $[0, H]$, this inequality tells us that the regret incurred by bad estimation is at most $\widetilde{O}(H)$. To be precise, it holds with high probability that

$$\sum_{k=1}^{K} \mathbb{1}\left\{\left(\mathcal{C}_{\text{ty}}^k\right)^c\right\}\left(V_1^* - V_{1,k}^{\pi^k}\right)(s_1^k) \leq \widetilde{O}(H). \tag{7}$$

Then, to better control the estimated value function, we need it to be bounded, which requires us to clip it. Specifically, we will use two crucial properties of our clipping method. First, if $\overline{Q}_{h,k}(s,a) \geq Q_h^*(s,a), \forall (s,a) \in \mathcal{S} \times \mathcal{A}$, then we have $\overline{V}_{h,k}(s) \geq V_h^*(s), \forall s \in \mathcal{S}$. Similarly if $\overline{Q}_{h,k}(s,a) \leq Q_h^*(s,a), \forall (s,a) \in \mathcal{S} \times \mathcal{A}$, then we have $\overline{V}_{h,k}(s) \leq V_h^*(s), \forall s \in \mathcal{S}$.

In addition, we can prove that whenever a clip is triggered for $s_h^k$, we have $n_k(h, s_h^k, a_h^k) \leq \alpha_k$ with $\alpha_k = \widetilde{O}(H^2)$. As a result, it is possible to show that the total regret incurred by clipping is at most $\widetilde{O}(H^4 SA)$, which is a lower-order term when $T$ is sufficiently large. That is, let $\mathcal{E}_{H,k}^{cum}$ denote the event that there is no clipping during episode $k$. Then, it holds with high probability that[2]

$$\sum_{k=1}^{K} \mathbb{1}\left\{\mathcal{C}_{\text{ty}}^k \cap \left(\mathcal{E}_{H,k}^{cum}\right)^c\right\}\left(V_1^* - V_{1,k}^{\pi^k}\right)(s_1^k) \leq \widetilde{O}(H^4 S^2 A). \tag{8}$$

As claimed before, because of the randomness in Gaussian noise, our algorithm SSR will encourage exploration and it takes effect when there is no clipping and the estimation is not too bad. In other words, it can be optimistic. However, also because of this randomness, its optimism only holds in a probabilistic sense. In precise, it is possible to show that

$$\mathbb{P}\left(\overline{V}_{h,k}(s) \geq V_h^*(s), \forall h \in [H], s \in \mathcal{S} \mid \mathcal{C}_{\text{ty}}^k\right) \geq C_{\text{ty}}, \tag{9}$$

where the value of constant $C_{\text{ty}}$ depends on the type of noise we choose. Meanwhile, we can also prove a very similar probabilistic pessimism, which means to have $\overline{V}_{h,k}(s) \leq V_h^*(s), \forall h \in [H], s \in \mathcal{S}$ with constant probability. The property of optimism and pessimism will help us upper bound the absolute value of $V_1^*(s_1^k) - \overline{V}_{1,k}(s_1^k)$, which will be discussed soon.

## 5.2 Regret Decomposition

Now, given equations (7) and (8), we can see that for each episode $k$, it only remains to bound $\mathbb{1}\left\{\mathcal{C}_{\text{ty}}^k \cap \mathcal{E}_{H,k}^{cum}\right\}\left(V_1^* - V_{1,k}^{\pi^k}\right)(s_1^k)$. Technically, the further defined the good event $\mathcal{G}_k$ will help make $\overline{V}_{h,k}$ better-behaved. Its precise definition will be given in the appendix. Therefore, it is sufficient to bound $\mathbb{1}\left\{\mathcal{G}_k\right\}\left(V_1^* - V_{1,k}^{\pi^k}\right)(s_1^k)$, which means to have

$$\text{Reg}(M, K, \text{SSR}_{\text{ty}}) \leq \sum_{k=1}^{K} \mathbb{1}\left\{\mathcal{G}_k\right\}(\underbrace{V_1^* - \overline{V}_{1,k}}_{\text{pessimism}} + \underbrace{\overline{V}_{1,k} - V_{1,k}^{\pi^k}}_{\text{estimation error}})(s_1^k) + \widetilde{O}(H^4 SA). \tag{10}$$

To proceed, we need to define two auxiliary value functions $\underline{V}_{h,k}$ and $\overline{\overline{V}}_{h,k}$, which are obtained by virtually running policy $\pi^k$ on some deliberately perturbed MDPs. In particular, they are designed such that $\underline{V}_{h,k} \leq \overline{V}_{h,k} \leq \overline{\overline{V}}_{h,k}$ holds under the good event $\mathcal{G}_k$.

**Pessimism Term** Here, as a technical novelty, we bound the pessimism term's absolute value. Meanwhile, different from [52] and [2], by applying both optimism and pessimism, we do not resort to an independent copy of the perturbed MDP to bound the pessimism term and give a conceptually simpler analysis. In particulary, by defining $C_1 = 1/\min\{C_{\text{Ho}}, C_{\text{Be}}\}$. it is possible to show that

$$\mathbb{1}\left\{\mathcal{G}_k\right\}\left|V_{h,k}^*(s_h^k) - \overline{V}_{h,k}(s_h^k)\right| \leq \mathbb{1}\left\{\mathcal{G}_k\right\} C_1 \left(\left|\overline{\overline{V}}_{h,k}^{\pi^k}(s_h^k) - V_{h,k}^{\pi^k}(s_h^k)\right| + \left|\underline{V}_{h,k}^{\pi^k}(s_h^k) - V_{h,k}^{\pi^k}(s_h^k)\right|\right). \tag{11}$$

The full proof is given in Appendix under Lemma 15.

---

[2]Technically, this is not precisely how we bound the regret incurred by clipping, but it aligns better with the intuition. Full technical details can be found in Appendix.

**Estimation Error Term**  The sum of pessimism term and estimation error term can be further bounded via the techniques of recursion used in [7]. However, we want to emphasize the difference that in their algorithm, the estimated value is optimistic with high probability, which makes $\overline{V}_{h,k}(s_h^k) - V_h^*(s_h^k)$ always positive. Instead, since our optimism only holds with constant probability, we use absolute value to keep the estimation error terms positive. As a result, we show that

$$\left|\overline{V}_{1,k} - V_{1,k}^{\pi^k}\right|(s_1^k) + \left|\overline{\overline{V}}_{1,k}^{\pi^k} - V_{1,k}^{\pi^k}\right|(s_1^k) + \left|\underline{V}_{1,k}^{\pi^k} - V_{1,k}^{\pi^k}\right|(s_1^k) \lesssim e^{3C}\sum_{h=1}^{H}\left(L\sigma_{\text{ty}}^k(x_h^k) + \mathcal{M}_{h,k}\right), \quad (12)$$

where $L$ denotes some poly-logarithmic term and $\mathcal{M}_{h,k}$ denotes some martingale difference sequence term at period $h$, episode $k$. The full proof is given in Appendix under Lemma 19,

## 5.3 Combining Different Terms

By combining equations (10), (11) and (12) and applying concentration inequalities to MDP $\mathcal{M}_{h,k}$, it is possible to show that

$$\text{Reg}\left(M, K, \mathsf{SSR}_{\text{ty}}\right) \leq e^{3C_1}\sum_{k=1}^{K}\sum_{h=1}^{H}\mathbb{1}\left\{\mathcal{G}_k\right\}L\sigma_{\text{ty}}^k(x_h^k) + \widetilde{O}\left(H\sqrt{T} + H^4 S^2 A\right). \quad (13)$$

Then, a final high-probability regret bound can be obtained by summing each individual terms over $k, h$ separately. It is well-known among literature that

$$\sum_{k=1}^{K}\sum_{h=1}^{H}\sqrt{\frac{1}{n_k(x_h^k)+1}} \leq \widetilde{O}\left(\sqrt{HSAT}\right), \qquad \sum_{k=1}^{K}\sum_{h=1}^{H}\frac{1}{n_k(x_h^k)+1} \leq \widetilde{O}\left(HSA\right). \quad (14)$$

Recall the definition of $\sigma_{\text{Ho}}^k$ in equation (5). By using these two inequalities, the bound in equation (13) can be made explicit if we use Hoeffding-type noise. As a result, we have

$$\text{Reg}\left(M, K, \mathsf{SSR}_{\text{Ho}}\right) \leq \widetilde{O}\left(H^{1.5}\sqrt{SAT} + H^4 S^2 A\right).$$

### 5.3.1 Bound on Sum of Variance

Analyses become more involved when Bernstein-type noise is used. Specifically, notice that inequalities in (14) cannot directly be used to bound $\sum_{k,h}\mathbb{V}\left(\tilde{P}_{x_h^k}^k, \overline{V}_{h+1,k}\right)$. Here, we apply some techniques developed in [7]. However, since the optimism only holds with constant probability, the details for specific terms are quite different.

For the ease of exposure, we will ignore all constants and define $\hat{\mathbb{V}}_{h,k}^* = \mathbb{V}\left(\tilde{P}_{x_h^k}^k, V_h^*\right), \hat{\overline{\mathbb{V}}}_{h,k} = \mathbb{V}\left(\tilde{P}_{x_h^k}^k, \overline{V}_{h,k}\right)$. Then, by using Cauchy-Schwartz inequality and equation (14), we can get

$$U \stackrel{\text{def}}{=} \sum_{k,h}\mathbb{1}\left\{\mathcal{G}_k\right\}\sqrt{\frac{L}{n_k(x_h^k)+1}}\left(\sqrt{\hat{\mathbb{V}}_{h,k}^*} + \sqrt{\hat{\overline{\mathbb{V}}}_{h,k}}\right) \leq \sqrt{\widetilde{O}\left(HSA\right)\sum_{k,h}\mathbb{1}\left\{\mathcal{G}_k\right\}\left(\hat{\mathbb{V}}_{h,k}^* + \hat{\overline{\mathbb{V}}}_{h,k}\right)} \quad (15)$$

Here, note that $U \approx \sum_{k,h}\sigma_{\text{Be}}^k(x_h^k)$. Then, after some steps of algebra, it is possible to show that

$$\sum_{k=1}^{K}\sum_{h=1}^{H}\mathbb{1}\left\{\mathcal{G}_k\right\}\left(\hat{\mathbb{V}}_{h,k}^* + \hat{\overline{\mathbb{V}}}_{h,k}\right) \leq \widetilde{O}\left(HT + H^2 U\right) \qquad \text{(When } T \geq \widetilde{\Omega}\left(H^5 S^2 A\right)\text{)}$$

$$\implies U \leq \widetilde{O}\left(\sqrt{HSA\left(HT + H^2 U\right)}\right) \leq \widetilde{O}\left(H\sqrt{SAT} + H^{1.5}\sqrt{U}\right). \quad \text{(By using equation (15))}$$

Now, we can see that $\sum_{k,h}\sigma_{\text{Be}}^k(x_h^k) \approx U \leq \widetilde{O}\left(H\sqrt{SAT}\right)$ satisfies this inequality. Finally, by plugging this result back into equation (13), we can have

$$\text{Reg}\left(M, K, \mathsf{SSR}_{\text{Be}}\right) \leq \widetilde{O}\left(H\sqrt{SAT} + H^4 S^2 A\right),$$

which matches the known lower bound when $T \geq \widetilde{\Omega}\left(H^6 S^3 A\right)$.

# 6   Conclusion

We gave a new algorithm with randomized exploration, SSR, for tabular MDP, which enjoys a near-optimal $\widetilde{O}\left(H\sqrt{SAT}\right)$ regret bound in the time-homogeneous model. Previously, near-optimal regret bounds can only be achieved by optimistic algorithms. Our result also highlights the importance of using a single random seed for the entire episode and using the variance information in tuning the magnitude of noise (cf. Bernstein's inequality).

One important open problem is whether randomized exploration can a achieve a horizon-free regret bound in the time-homogeneous model where the transition is the same at different levels [53, 49, 55]. Another possible future direction is to consider whether the sub-optimal lower order terms $\widetilde{O}\left(H^4S^2A\right)$ can be further improved to relax the current requirement $T \geq \widetilde{\Omega}\left(H^6S^3A\right)$ for being near-optimal.

## Acknowledgements

We sincerely thank Jinglin Chen and Chao Qin for pointing out mistakes in the initial draft of this paper. We are grateful for their careful reading and insightful discussion. This work was supported in part by NSF TRIPODS II-DMS 2023166, NSF CCF 2007036, NSF IIS 2110170, NSF DMS 2134106, NSF CCF 2212261, NSF IIS 2143493, NSF CCF 2019844.

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
