^k_H, a^k_H, r^k_H$ and compute $\hat{R}^{k+1}_{h,s,a}$, $\hat{P}^{k+1}_{h,s,a}$ and $n_{k+1}(h, s, a)$ for
     all $(h, s, a) \in [H] \times \mathcal{S} \times \mathcal{A}$
10 **end**

---

$\sum_{l=1}^{k-1} \mathbb{1}\{(s^l_h, a^l_h) = (s, a)\}$ be the number of times action $a$ is taken at state $s$ and time $h$ before episode $k$, where $\mathbb{1}\{\cdot\}$ is the indicator function. We define

$$\hat{R}^k_{h,s,a} = \frac{\sum_{l=1}^{k-1} \mathbb{1}\{(s^l_h, a^l_h) = (s, a)\} r^l_{h, s^l_h, a^l_h}}{n_k(h, s, a) + 1}, \tag{2}$$

$$\hat{P}^k_{h,s,a}(s') = \frac{\sum_{l=1}^{k-1} \mathbb{1}\{(s^l_h, a^l_h, s^l_{h+1}) = (

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

# Contents

## A Table of Notations

| Symbol | Meaning |
|---|---|
| $\mathcal{S}$ | The state space |
| $\mathcal{A}$ | The action space |
| $S$ | Size of state space |
| $A$ | Size of action space |
| $H$ | The length of horizon |
| $K$ | The total number of episodes |
| $T$ | The total number of steps, $T = HK$ |
| $\pi^k$ | The greedy policy generated in Algorithm 1 at episode $k$ |
| $R_{h,s,a}$ | Expected reward function at $(h,s,a)$ |
| $P_{h,s,a}(s')$ | Transition probability |
| $M$ | Underlying true MDP, $M = (H, \mathcal{S}, \mathcal{A}, R, P, s_1)$ |
| $n_k(h,s,a)$ | $\sum_{k'=1}^{k-1} \mathbb{1}\{(s_h^{k'}, a_h^{k'}) = (s,a)\}$ |
| $\hat{R}_{h,s,a}^k$ | Estimated reward function, $\frac{1}{n_k(h,s,a)+1} \sum_{l=1}^{k-1} \mathbb{1}\{(s_h^l, a_h^l) = (s,a)\} r_{h,s_h^l,a_h^l}^l$ |
| $\hat{P}_{h,s,a}^k(s')$ | Estimated transition kernel, $\frac{1}{n_k(h,s,a)+1} \sum_{l=1}^{k-1} \mathbb{1}\{(s_h^l, a_h^l, s_{h+1}^l) = (s,a,s')\}$ |
| $\tilde{P}_{h,s,a}^k(s')$ | Estimated transition probability with a slightly different denominator, $\frac{1}{\max\{n_k(h,s,a),1\}} \sum_{l=1}^{k-1} \mathbb{1}\{(s_h^l, a_h^l, s_{h+1}^l) = (s,a,s')\}$ |
| $\hat{M}^k$ | Estimated MDP, $\hat{M}^k = (H, \mathcal{S}, \mathcal{A}, \hat{P}, \hat{R}, s_1)$ |
| $\gamma_{\text{ty}}^k(h,s,a)$ | $\sigma_{\text{ty}}^k(h,s,a)\sqrt{\log(40k^4)}$ |
| $\hat{z}_k$ | Perturbation's single random source during episode $k$ from a standard Gaussian, $\hat{z}_k \sim \mathcal{N}(0,1)$ |
| $w_{\text{ty}}^k(h,s,a)$ | Noise of type "ty", $w_{\text{ty}}^k(h,s,a) = \sigma_{\text{ty}}^k(h,s,a)\hat{z}_k$ |
| $\underline{w}_{\text{ty}}^k(h,s,a)$ | $-\gamma_{\text{ty}}^k(h,s,a)$ |
| $\overline{w}_{\text{ty}}^k(h,s,a)$ | $\gamma_{\text{ty}}^k(h,s,a)$ |
| $\overline{M}_{\text{ty}}^k$ | Perturbed estimated MDP with ty-type noise, $\overline{M}^k = (H, \mathcal{S}, \mathcal{A}, \hat{P}, \hat{R} + w_{\text{ty}}^k, s_1)$ |
| $\underline{M}_{\text{ty}}^k$ | Negatively perturbed MDP, $\underline{M}_{\text{ty}}^k = (H, \mathcal{S}, \mathcal{A}, \hat{P}, \hat{R} + \underline{w}_{\text{ty}}^k, s_1)$ |
| $\overline{\overline{M}}_{\text{ty}}^k$ | Positively perturbed MDP, $\underline{M}_{\text{ty}}^k = (H, \mathcal{S}, \mathcal{A}, \hat{P}, \hat{R} + \overline{w}_{\text{ty}}^k, s_1)$ |
| $V_h^* / V_{h,k}^*$ | Optimal value function at step $h$ for true MDP $M$ |
| $V_h^{\pi^k} / V_{h,k}^{\pi^k}$ | Value function at step $h$ by running policy $\pi^k$ on true MDP $M$ |
| $\overline{Q}_{h,k}$ | $Q$-value function obtained by running Algorithm 1 |
| $\overline{V}_{h,k}$ | Value function obtained by running policy $\pi^k$ on $\overline{M}^k$ with a clipping of threshold $2(H - h + 1)$ |
| $\underline{V}_{h,k}$ | Value function obtained by running policy $\pi^k$ on $\underline{M}^k$ with a clipping of threshold $2(H - h + 1)$ |
| $\overline{\overline{V}}_{h,k}$ | Value function obtained by running policy $\pi^k$ on $\overline{\overline{M}}^k$ with a clipping of threshold $2(H - h + 1)$ |
| $\mathcal{R}_{h,s,a}^k$ | $\hat{R}_{h,s,a}^k - R_{h,s,a}$ |
| $\mathcal{P}_{h,s,a}^k$ | $\left\langle \hat{P}_{h,s,a}^k - P_{h,s,a}, V_{h+1}^* \right\rangle$ |
| $\mathcal{H}_h^k$ | The historical observations and actions till time $h$ in episode $k$, $\left\{(s_l^j, a_l^j, r_l^j) : j \leq k \text{ and } l \leq H \text{ if } j < k, \text{ else } l \leq h\right\}$ |
| $\overline{\mathcal{H}}_h^k$ | The historical observations and actions till time $h$ and episode $k$, plus the randomness in episode $k$, $\mathcal{H}_h^k \cup \{\hat{z}_k\}$ |
| $\mathbb{V}(P, V)$ | Variance of $V \in \mathbb{R}^S$ under distribution $P \in \Delta^S$, $\sum_{s \in \mathcal{S}} P(s)(V(s) - \langle P, V \rangle)^2$ |
| $\alpha_k$ | $200H^2 \log(2HSAk^2) \log(40k^4)$ |

| | |
|---|---|
| $\sigma^k_{\mathrm{ty}}(h,s,a)$ | Magnitude of perturbation. $\mathrm{ty} \in \{\mathrm{Ho}, \mathrm{Be}\}$ |
| ty | Reserved subscript for denoting perturbation type, $\mathrm{ty} \in \{\mathrm{Ho}, \mathrm{Be}\}$, where "Ho" denotes Hoeffding-type and "Be" denotes Bernstein-type |
| $\sigma^k_{\mathrm{Ho}}(h,s,a)$ | $H\sqrt{\dfrac{\log(2HSAk^2)}{n_k(h,s,a)+1}} + \dfrac{H}{n_k(h,s,a)+1}$ |
| $\sigma^k_{\mathrm{Be}}(h,s,a)$ | $\sqrt{\dfrac{16\mathbb{V}\left(\tilde{P}^k_{h,s,a}, \overline{V}_{k,h+1}\right)\log(2HSAk^2)}{n_k(h,s,a)+1}} + \dfrac{65H\log\left(2HSAk^2\right)}{n_k(h,s,a)+1} + \sqrt{\dfrac{\log(2HSAk^2)}{n_k(h,s,a)+1}}$ |
| $\sqrt{e^k_{\mathrm{Ho}}(h,s,a)}$ | $H\sqrt{\dfrac{\log(2HSAk^2)}{n_k(h,s,a)+1}} + \dfrac{H}{n_k(h,s,a)+1}$ |
| $\sqrt{e^k_{\mathrm{Be}}(h,s,a)}$ | $\sqrt{\dfrac{6\mathbb{V}\left(\tilde{P}^k_{h,s,a}, V^*_{h+1}\right)\log(2HSAk^2)}{n_k(h,s,a)+1}} + \dfrac{9H\log\left(2HSAk^2\right)}{n_k(h,s,a)+1} + \sqrt{\dfrac{\log(2HSAk^2)}{n_k(h,s,a)+1}}$ |
| $C_1$ | $\dfrac{1}{\Phi(1.5)-\Phi(1)}$ |

## B Good Events

**Definition 1.** *Let $M' = (H, \mathcal{S}, \mathcal{A}, P', R', s_1)$. We define the following confidence sets for both Bernstein-type and Hoeffding-type noise*

$$\mathcal{M}^k_{\mathrm{ty}} = \left\{ M' : \forall (h,s,a), \left|(R'_{h,s,a} - R_{h,s,a}) + \left\langle P'_{h,s,a} - P_{h,s,a}, V^*_{h+1}\right\rangle\right| \le \sqrt{e^k_{\mathrm{ty}}(h,s,a)} \right\}, \quad (16)$$

*where the confidence widths are set as*

$$\sqrt{e^k_{\mathrm{Be}}(h,s,a)} = \sqrt{\frac{6\mathbb{V}\left(\tilde{P}^k_{h,s,a}, V^*_{h+1}\right)\log\left(2HSAk^2\right)}{n_k(h,s,a)+1}} \quad (17)$$
$$+ \frac{9H\log\left(2HSAk^2\right)}{n_k(h,s,a)+1} + \sqrt{\frac{\log\left(2HSAk^2\right)}{n_k(h,s,a)+1}},$$

$$\sqrt{e^k_{\mathrm{Ho}}(h,s,a)} = H\sqrt{\frac{\log\left(2HSAk^2\right)}{n_k(h,s,a)+1}} + \frac{H}{n_k(h,s,a)+1}. \quad (18)$$

*We also define two events $\mathcal{E}^1_k$ and $\mathcal{E}^2_k$ as the following:*

$$\mathcal{E}^1_k = \left\{ \left|\hat{R}^k_{h,s,a} - R_{h,s,a}\right| \le \sqrt{\frac{\log\left(2HSAk^2\right)}{n_k(h,s,a)+1}} + \frac{1}{n_k(h,s,a)+1}, \forall (h,s,a) \right\}, \quad (19)$$

$$\mathcal{E}^2_k = \left\{ \left|\left\langle \hat{P}^k_{h,s,a} - P_{h,s,a}, V^*_{h+1}\right\rangle\right| \le \sqrt{\frac{6\mathbb{V}\left(\tilde{P}^k_{h,s,a}, V^*_{h+1}\right)\log\left(2HSAk^2\right)}{n_k(h,s,a)+1}} \right. \quad (20)$$
$$\left. + \frac{8H\log\left(2HSAk^2\right)}{n_k(h,s,a)+1}, \forall (h,s,a) \right\}.$$

We have the following lemmas about concentration of events.

**Lemma 1.** *For fixed $(k,h,s,a)$, let $n = n_k(h,s,a)$. Then, if $n \ge 1$, for any fixed $\delta > 0$, we have*

$$\mathbb{P}\left( |\hat{R}^k_{h,s,a} - R_{h,s,a}| \ge \sqrt{\frac{\log(2/\delta)}{n+1}} + \frac{1}{n+1} \right) \le \delta.$$

*Proof.* Let $\hat{R}^k_{h,s,a} = \frac{1}{n+1}\sum_{i=1}^n r_{(h,s,a),i}$, where $r_{(h,s,a),i} \sim \mathscr{R}_{h,s,a}$ are i.i.d. samples. By definition of the MDP, we have $\mathbb{E}\left[r_{(h,s,a),i}\right] = R_{h,s,a}$. Then, notice that

$$\hat{R}^k_{h,s,a} = \frac{1}{n+1}\sum_{i=1}^n r_{(h,s,a),i} = \frac{1}{n}\sum_{i=1}^n r_{(h,s,a),i} - \frac{1}{n(n+1)}\sum_{i=1}^n r_{(h,s,a),i}.$$

Since the reward is assumed to be bounded in $[0, 1]$, we have $\frac{1}{n(n+1)} \sum_{i=1}^{n} r_{(h,s,a),i} \leq \frac{1}{n+1}$. Then, for fixed $\delta > 0$, we have

$$\mathbb{P}\left(|\hat{R}_{h,s,a}^k - R_{h,s,a}| \geq \sqrt{\frac{\log(2/\delta)}{n+1}} + \frac{1}{n+1}\right)$$

$$=\mathbb{P}\left(\left|\frac{1}{n} \sum_{i=1}^{n} r_{(h,s,a),i} - R_{h,s,a} - \frac{1}{n(n+1)} \sum_{i=1}^{n} r_{(h,s,a),i}\right| \geq \sqrt{\frac{\log(2/\delta)}{n+1}} + \frac{1}{n+1}\right)$$

$$\leq\mathbb{P}\left(\left|\frac{1}{n} \sum_{i=1}^{n} r_{(h,s,a),i} - R_{h,s,a}\right| + \frac{1}{n+1} \geq \sqrt{\frac{\log(2/\delta)}{n+1}} + \frac{1}{n+1}\right) \quad \text{(By triangle inequality)}$$

$$\leq\mathbb{P}\left(\left|\frac{1}{n} \sum_{i=1}^{n} r_{(h,s,a),i} - R_{h,s,a}\right| \geq \sqrt{\frac{\log(2/\delta)}{2n}}\right) \quad \text{(Since } n+1 \leq 2n \text{ for } n \geq 1)$$

$$\leq\delta. \quad \text{(By standard Hoeffding's inequality)}$$

$\square$

**Lemma 2.** *For fixed* $(k, h, s, a)$*, let* $n = n_k(h, s, a)$ *and* $V \in \mathbb{R}^S$ *be some non-negative value function such that* $\|V\|_\infty \leq H$*. Then, if* $n \geq 1$*, for any fixed* $\delta > 0$*, we have*

$$\mathbb{P}\left(\left|\left\langle \hat{P}_{h,s,a}^k - P_{h,s,a}, V\right\rangle\right| \geq H\sqrt{\frac{\log(2/\delta)}{n+1}} + \frac{H}{n+1}\right) \leq \delta, \tag{21}$$

$$\mathbb{P}\left(\left|\left\langle \hat{P}_{h,s,a}^k - P_{h,s,a}, V\right\rangle\right| \geq \sqrt{\frac{6\mathbb{V}\left(\tilde{P}_{h,s,a}^k, V\right)\log(2/\delta)}{n+1}} + \frac{8H\log(2/\delta)}{n+1}\right) \leq \delta. \tag{22}$$

*Proof.* For fixed $(h, s, a)$, we generate $n$ i.i.d. samples of $s_{(h,s,a),i} \sim P_{h,s,a}$ and consider $V(s_{(h,s,a),i})$. Then, by taking $n_k(h, s, a) = n$, we have

$$\left\langle \hat{P}_{h,s,a}^k, V\right\rangle = \frac{1}{n} \sum_{i=1}^{n} V\left(s_{(h,s,a),i}\right) - \frac{1}{n(n+1)} \sum_{i=1}^{n} V\left(s_{(h,s,a),i}\right).$$

The first result in equation (21) can be proved very similarly as Lemma 1 using Hoeffding's inequality by simply replacing the upper bound of 1 in reward by $H$.

Then, for second result, we first consider $n \geq 2$. For some $\delta > 0$, define

$$b_{(h,s,a),n} = \sqrt{\frac{2\mathbb{V}\left(\tilde{P}_{h,s,a}^k, V\right)\log(2/\delta)}{n-1}} + \frac{7H\log(2/\delta)}{3(n-1)} + \frac{H}{n+1}.$$

By noticing that $F(s) \leq H$ and applying similar technique in proof of Lemma 1, we have

$$\mathbb{P}\left(\left|\left\langle \hat{P}_{h,s,a}^k - P_{h,s,a}, V\right\rangle\right| \geq b_{(h,s,a),n}\right)$$

$$\leq\mathbb{P}\left(\left|\left\langle \tilde{P}_{h,s,a}^k - P_{h,s,a}, V\right\rangle\right| \geq \sqrt{\frac{2\mathbb{V}\left(\tilde{P}_{h,s,a}^k, V\right)\log(2/\delta)}{n-1}} + \frac{7H\log(2/\delta)}{3(n-1)}\right)$$

$$\leq\delta. \quad \text{(By Lemma 31, the empirical Bernstein's inequality)}$$

Then, since $3(n-1) \geq n+1$ when $n \geq 2$, we can easily check that

$$b_{(h,s,a),n} \leq \sqrt{\frac{6\mathbb{V}\left(\tilde{P}_{h,s,a}^k, V\right)\log(2/\delta)}{n+1}} + \frac{8H\log(2/\delta)}{n+1}.$$

Finally, since $\|V\|_\infty \le H$, when $n = 1$, we trivially have

$$\left|\left\langle \hat{P}^k_{h,s,a} - P_{h,s,a}, V \right\rangle\right| \le H \le \sqrt{\frac{6\mathbb{V}\left(\tilde{P}^k_{h,s,a}, V\right)\log\left(2/\delta\right)}{n+1}} + \frac{8H\log\left(2/\delta\right)}{n+1}.$$

Therefore, we can conclude that

$$\mathbb{P}\left(\left|\left\langle \hat{P}^k_{h,s,a} - P_{h,s,a}, V \right\rangle\right| \ge \sqrt{\frac{6\mathbb{V}\left(\tilde{P}^k_{h,s,a}, V\right)\log\left(2/\delta\right)}{n+1}} + \frac{8H\log\left(2/\delta\right)}{n+1}\right) \le \delta.$$

$\square$

**Lemma 3.** $\sum_{k=1}^\infty \mathbb{P}\left(\left(\mathcal{E}^1_k\right)^c\right) \le \frac{\pi^2}{6}.$

*Proof.* Let $n = n_k(h,s,a)$. Then, for some fixed $(h,s,a)$, $n \ge 1$ and $\delta_n > 0$, by Lemma 1, we have

$$\mathbb{P}\left(|\hat{R}^k_{h,s,a} - R_{h,s,a}| \ge \sqrt{\frac{\log\left(2/\delta_n\right)}{n+1}} + \frac{1}{n+1}\right) \le \delta_n.$$

Therefore, by taking $\delta_n = \frac{1}{HSAn^2}$, a union bound will give us

$$\sum_{n=1}^\infty \sum_{h,s,a} \mathbb{P}\left(\left\{|\hat{R}^k_{h,s,a} - R_{h,s,a}| \ge \sqrt{\frac{\log\left(2HSAn^2\right)}{n+1}} + \frac{1}{n+1}\right\}\right) \le \sum_{n=1}^\infty \frac{1}{n^2} = \frac{\pi^2}{6}.$$

Therefore, we have

$$\sum_{k=1}^\infty \mathbb{P}\left(\exists(h,s,a): n_k(h,s,a) > 0, |\hat{R}^k_{h,s,a} - R_{h,s,a}| \ge \sqrt{\frac{\log\left(2HSAn_k(h,s,a)^2\right)}{n_k(h,s,a)+1}} + \frac{1}{n_k(h,s,a)+1}\right)$$
$$\le \frac{\pi^2}{6}.$$

Since the MDP is time-inhomogeneous, each $(h,s,a)$ can only be visited at most once during one episode, which implies $n_k(h,s,a) \le k$. Therefore, we have

$$\sqrt{\frac{\log\left(2HSAn_k(h,s,a)\right)}{n_k(h,s,a)+1}} + \frac{1}{n_k(h,s,a)+1} \le \sqrt{\frac{\log\left(2HSAk^2\right)}{n_k(h,s,a)+1}} + \frac{1}{n_k(h,s,a)+1}$$

and thus the proof is complete. $\square$

**Lemma 4.** $\sum_{k=1}^\infty \mathbb{P}\left(\left(\mathcal{E}^2_k\right)^c\right) \le \frac{\pi^2}{6}.$

*Proof.* This proof will be very similar to proof of Lemma 3. In specific, for fixed $(h,s,a)$, let $n = n_k(h,s,a) \ge 1$. Then, for any $\delta_n > 0$, since $\left\|V^*_{h+1}\right\|_\infty \le H$, by Lemma 2, we have

$$\mathbb{P}\left(\left|\left\langle \hat{P}^k_{h,s,a} - P_{h,s,a}, V^*_{h+1} \right\rangle\right| \ge \sqrt{\frac{6\mathbb{V}\left(\tilde{P}^k_{h,s,a}, V^*_{h+1}\right)\log\left(2/\delta_n\right)}{n+1}} + \frac{8H\log\left(2/\delta_n\right)}{n+1}\right) \le \delta_n.$$

Therefore, by taking $\delta_n = \frac{1}{HSAn^2}$ and applying a similar union bound argument used in the proof of Lemma 3, we can conclude $\sum_{k=1}^\infty \mathbb{P}\left(\left(\mathcal{E}^2_k\right)^c\right) \le \frac{\pi^2}{6}.$ $\square$

We further define the event $\mathcal{C}^k_{\text{ty}} = \left\{\hat{M}^k \in \mathcal{M}^k_{\text{ty}}\right\}$. With what we have proved above, it will be straightforward to show the following results about $\mathcal{C}^k_{\text{ty}}$.

**Lemma 5.** $\sum_{k=1}^{\infty} \mathbb{P}\left(\left(\mathcal{C}_{\mathrm{Be}}^k\right)^c\right) = \sum_{k=1}^{\infty} \mathbb{P}\left(\hat{M}^k \notin \mathcal{M}_{\mathrm{Be}}^k\right) \leq \frac{\pi^2}{3}$

*Proof.* We can easily notice $\mathcal{E}_k^1 \cap \mathcal{E}_k^2 \implies \hat{M}^k \in \mathcal{M}_{\mathrm{Be}}^k$, which implies $\hat{M}^k \notin \mathcal{M}_{\mathrm{Be}}^k \implies (\mathcal{E}_k^1)^c \cup (\mathcal{E}_k^2)^c$. The first result then follows straightforwardly by applying Lemma 3 and Lemma 4. $\square$

**Lemma 6.** $\sum_{k=1}^{\infty} \mathbb{P}\left(\left(\mathcal{C}_{\mathrm{Ho}}^k\right)^c\right) = \sum_{k=1}^{\infty} \mathbb{P}\left(\hat{M}^k \notin \mathcal{M}_{\mathrm{Ho}}^k\right) \leq \frac{\pi^2}{3}$.

*Proof.* Similarly, for fixed $(h, s, a)$, we generate $n$ i.i.d. samples $s_{(h,s,a),i} \sim P_{h,s,a}$ and $r_{(h,s,a),i} \sim \mathcal{R}_{h,s,a}$ for $i = 1, \ldots, n$ respectively. Define $Y_{(h,s,a),i} = r_{(h,s,a),i} + V_{h+1}^*(s_{(h,s,a),i})$ and we have $\mathbb{E}\left[Y_{(h,s,a),i}\right] = R_{h,s,a} + \left\langle P_{h,s,a}, V_{h+1}^* \right\rangle$.

By definition of MDP, we know that $Y_{(h,s,a),i} \leq H$. Thus, we can use an argument similar to the proof of Lemma 1. In specific, let $n = n_k(h, s, a)$ and for $\delta_n > 0$, we have

$$\mathbb{P}\left(\left|\frac{1}{n+1}\sum_{i=1}^{n} Y_{(h,s,a),i} - \mathbb{E}\left[Y_{(h,s,a),i}\right]\right| \geq H\sqrt{\frac{\log(2/\delta_n)}{n+1}} + \frac{H}{n+1}\right)$$

$$=\mathbb{P}\left(\left|\left(\hat{R}_{h,s,a}^k - R_{h,s,a}\right) + \left\langle \hat{P}_{h,s,a}^k - P_{h,s,a}, V_{h+1}^* \right\rangle\right| \geq H\sqrt{\frac{\log(2/\delta_n)}{n+1}} + \frac{H}{n+1}\right)$$

$$\leq \delta_n.$$

Then, we can take $\delta_n = \frac{1}{HSAn^2}$ and apply a similar union bound argument in used in the proof of Lemma 3. As a result, we can obtain

$$\sum_{k=1}^{\infty} \mathbb{P}\left(\hat{M}^k \notin \mathcal{M}_{\mathrm{Ho}}^k\right) \leq \frac{\pi^2}{6} \leq \frac{\pi^2}{3}.$$

$\square$

We can also have well-behaved bounds on magnitude of noise and estimated value functions.

**Definition 2.** *We define* $w_{\mathrm{ty}}^k(h, s, a) = \sigma_{\mathrm{ty}}^k(h, s, a)\hat{z}_k$ *and* $\gamma_{\mathrm{ty}}^k(h, s, a) = \sigma_{\mathrm{ty}}^k(h, s, a)\sqrt{\log(40k^4)}$. *We define the event* $\mathcal{E}_k^w$ *as*

$$\mathcal{E}_k^w = \left\{ \forall (h, s, a), |w_{\mathrm{ty}}^k(h, s, a)| \leq \gamma_{\mathrm{ty}}^k(h, s, a) \right\}.$$

**Lemma 7.** $\sum_{k=1}^{K} \mathbb{P}\left((\mathcal{E}_k^w)^c\right) \leq \frac{\pi^2}{3}$ *regardless the type of noise we choose.*

*Proof.* For any $k \in [K]$, by the tail bound of Gaussian distribution,

$$\mathbb{P}\left(|\hat{z}_k| \geq \sqrt{\log(40k^4)}\right) \leq 2\exp\left(-\frac{\log(40k^4)}{2}\right) \leq \frac{2}{k^2}.$$

Summing over $k \in [K]$,

$$\sum_{k=1}^{K} \mathbb{P}\left(\left(\mathcal{E}_w^k\right)^c\right) = \sum_{k=1}^{K} \mathbb{P}\left(|\hat{z}_k| \geq \sqrt{\log(40k^4)}\right) \leq \sum_{k=1}^{\infty} \frac{2}{k^2} \leq \frac{\pi^2}{3}.$$

Note that this result does not depend on the type of noise we choose. $\square$

Now, we define the following good events that hold with high probability and will be used throughout the whole proof.

**Definition 3** (Good events $\mathcal{G}_k$)**.** *Let* $\mathcal{G}_{k,\mathrm{ty}} = \left\{ \mathcal{C}_{\mathrm{ty}}^k \cap \mathcal{E}_k^w \right\}$.

The subscript "ty" will be ignored later since it is clear from the context.

**Definition 4.** *With* $\alpha_k = 200H^2\log(2HSAk^2)\log(40k^4)$, *we define events* $\mathcal{E}_{h,k}^{th}$ *and* $\mathcal{E}_{h,k}^{cum}$ *as*

$$\mathcal{E}_{h,k}^{th} = \left\{ n_k(h, s_h^k, a_h^k) \geq \alpha_k \right\}, \quad \mathcal{E}_{h,k}^{cum} = \bigcap_{i=1}^{h} \mathcal{E}_{i,k}^{th}. \tag{23}$$

We will show that under events $\mathcal{E}_k^w$, $\mathcal{E}_{h,k}^{th}$ and $\hat{M}^k \in \mathcal{M}_{\text{ty}}^k$, no clipping happens on $s_h^k$.

**Lemma 8.** *Assume that $\mathcal{E}_k^w$, $\mathcal{E}_{h,k}^{th}$ and $\hat{M}^k \in \mathcal{M}_{\text{ty}}^k$ hold. Then, regardless the type of noise we choose, it holds that*

$$|\overline{Q}_{h,k}(s_h^k, a_h^k)| \leq 2(H - h + 1),$$

*which immediately tells us that no clipping is triggered for any $(s_h^k, a_h^k)$.*

*Proof.* We have that

$$\overline{Q}_{h,k}(s_h^k, a_h^k) = \hat{R}_{h,s_h^k,a_h^k}^k + \left\langle \hat{P}_{h,s_h^k,a_h^k}^k, \overline{V}_{h+1,k} \right\rangle + \sigma_{\text{ty}}^k(h, s_h^k, a_h^k)\hat{z}_k.$$

As we have $\left|\overline{V}_{h+1,k}\right| \leq 2(H - h)$ by clipping and $\hat{R}_{h,s_h^k,a_h^k}^k \in [0, 1]$, we only need to show that $\sigma_{\text{ty}}^k(h, s_h^k, a_h^k)\hat{z}_k \leq 1$. Under event $\mathcal{E}_w^k$, we have $\left|\sigma_{\text{ty}}^k(h, s_h^k, a_h^k)\hat{z}_k\right|$ is bounded by $\gamma_{\text{ty}}^k(h, s_h^k, a_h^k) = \sigma_{\text{ty}}^k(h, s_h^k, a_h^k)\sqrt{\log(40k^4)}$. Note that we have $\overline{V}_{h+1,k}(s) \in [-2H, 2H]$ by clipping for any $s \in \mathcal{S}$. Thus, by Lemma 32, we have $\mathbb{V}\left(\tilde{P}_{h,s,a}^k, \overline{V}_{h+1,k}\right) \leq 4H^2$ for any $(h, s, a)$.

By taking $\alpha_k = 200H^2 \log(2HSAk^2) \log(40k^4)$ and referring to the definitions of $\sigma_{\text{Be}}^k(h, s, a)$ in Equation (6), we can check that

$$\gamma_{\text{Be}}^k(h, s_h^k, a_h^k)$$
$$= \sigma_{\text{Be}}^k(h, s_h^k, a_h^k)\sqrt{\log(40k^4)}$$
$$= \left( \sqrt{\frac{16\mathbb{V}\left(\tilde{P}_{h,s_h^k,a_h^k}^k, \overline{V}_{k,h+1}\right)\log(2HSAk^2)}{n_k(h, s_h^k, a_h^k) + 1}} + \frac{65H\log(2HSAk^2)}{n_k(h, s_h^k, a_h^k) + 1} \right)\sqrt{\log(40k^4)}$$

$$\quad + \sqrt{\frac{\log(2HSAk^2)}{n_k(h, s_h^k, a_h^k) + 1}} \cdot \sqrt{\log(40k^4)}$$

$$\leq \left( \sqrt{\frac{64H^2\log(2HSAk^2)}{\alpha_k}} + \frac{65H\log(2HSAk^2)}{\alpha_k} + \sqrt{\frac{\log(2HSAk^2)}{\alpha_k}} \right)\sqrt{\log(40k^4)}$$
$$\text{(Event } \mathcal{E}_k^{th} \text{ implies } n_k(h, s_h^k, a_h^k) \geq \alpha_k)$$

$$\leq \sqrt{\frac{64}{200}} + \frac{65}{200H} + \sqrt{\frac{1}{200H^2}} \leq 1.$$

Thus, we have $\gamma_{\text{Be}}^k(h, s, a) \leq 1$ and we can similarly check that $\gamma_{\text{Ho}}^k(h, s, a) \leq 1$. As a result, we have

$$\left|\overline{Q}_{h,k}(s_h^k, a_h^k)\right| \leq 2(H - h + 1),$$

which completes the proof. $\qquad\square$

## C  Optimism

Let $\mathcal{H}_h^k$ denote the history trajectory, which is defined as

$$\mathcal{H}_h^k = \left\{ (s_l^j, a_l^j, r_l^j) : j \leq k \text{ and } l \leq H \text{ if } j < k, \text{ else } l \leq h \right\}. \tag{24}$$

We will prove that for both types of noise, $\overline{V}_{h,k}$ is optimistic with constant probability under certain conditions.

### C.1  Hoeffding-type Noise

**Lemma 9.** *Condition on history $\mathcal{H}_H^{k-1}$, if $\mathcal{G}_{k,\text{Ho}}$ holds and Hoeffding-based noise is applied, then $\overline{V}_{h,k}$ is optimisitic with constant probability for any $h \in [H]$. Specifically, we have*

$$\mathbb{P}\left(\overline{V}_{h,k}(s) \geq V_h^*(s), \forall h \in [H], s \in \mathcal{S} \mid \mathcal{H}_H^{k-1}, \mathcal{G}_{k,\text{Ho}}\right) \geq \Phi(1.9) - \Phi(1) := C_{\text{Ho}}.$$

*Proof.* We will show that if $\hat{z}_k \geq 1$, then for all $h \in [H]$ and $s \in \mathcal{S}$, we have $\overline{V}_{h,k}(s) \geq V_h^*(s)$. The proof will use induction and the argument is true for $h = H+1$ as $\overline{V}_{H+1,k}(s) = V_{H+1}^*(s) = 0$. Suppose the argument is true for timestep $h+1$ and for timestep $h$ we have

$$
\begin{aligned}
\overline{V}_{h,k}(s) &= \text{clip}_{2(H-h+1)}\left(\max_{a \in \mathcal{A}} \overline{Q}_{h,k}(s,a)\right) \\
&\geq \min\left\{2(H-h+1), \max_{a \in \mathcal{A}} \overline{Q}_{h,k}(s,a)\right\} \\
&\geq \min\left\{(H-h+1), \overline{Q}_{h,k}(s,\pi_h^*(s))\right\} \\
&\geq \min\left\{(H-h+1), \hat{R}_{h,s,\pi_h^*(s)}^k + \left\langle \hat{P}_{h,s,\pi_h^*(s)}^k, \overline{V}_{h+1,k}\right\rangle + \sigma_{\text{ty}}^k(h,s,\pi_h^*(s))\hat{z}_k\right\} \\
&\geq \min\left\{(H-h+1), \hat{R}_{h,s,\pi_h^*(s)}^k + \left\langle \hat{P}_{h,s,\pi_h^*(s)}^k, V_{h+1,k}^*\right\rangle + \sigma_{\text{ty}}^k(h,s,\pi_h^*(s))\hat{z}_k\right\} \\
&\qquad\qquad\qquad\qquad\qquad\qquad\qquad\qquad\qquad\qquad\qquad \text{(Inductive hypothesis)} \\
&\geq \min\left\{(H-h+1), R_{h,s,\pi_h^*(s)}^k + \left\langle P_{h,s,\pi_h^*(s)}^k, V_{h+1,k}^*\right\rangle\right\} \\
&\qquad\qquad\qquad \text{(Since } \hat{M}^k \in \mathcal{M}_{\text{Ho}}^k \text{ inferred by } \mathcal{G}_{k,\text{Ho}} \text{ and } \hat{z}_k \geq 1\text{)} \\
&\geq \min\left\{(H-h+1), Q_h^*(s,\pi_h^*(s))\right\} \\
&\geq V_h^*(s).
\end{aligned}
$$

Then by induction we have that the optimism is achieved for all $h \in [H]$ and $s \in \mathcal{S}$ simultaneously. Meanwhile, as stated in Definition 2, we have $\hat{z}_k \leq \sqrt{\log(40k^4)}$ under event $\mathcal{E}_k^w$ and numerically, $\sqrt{\log(40k^4)} \geq 1.9$. Therefore, the probability that $\hat{z}_k \geq 1$ under $\mathcal{E}_k^w$, inferred by $\mathcal{G}_{k,\text{Ho}}$, is at least

$$
\mathbb{P}\left(\hat{z}_k \geq 1 \mid \mathcal{H}_H^{k-1}, \mathcal{G}_{k,\text{Ho}}\right) = \frac{\Phi(1.9) - \Phi(1)}{\Phi(1.9) - \Phi(-1.9)} \geq \Phi(1.9) - \Phi(1) := C_{\text{Ho}}.
$$

Thus, we can conclude that

$$
\mathbb{P}\left(\overline{V}_{h,k}(s) \geq V_h^*(s), \forall h \in [H], s \in \mathcal{S} \mid \mathcal{H}_H^{k-1}, \mathcal{G}_{k,\text{Ho}}\right) \geq C_{\text{Ho}}.
$$

$\square$

### C.2 Bernstein-type Noise

The following proof of optimism applies some techniques used in [55]. We first present a technical lemma.

**Lemma 10.** *Let* $f_z : \Delta^S \times \mathbb{R}_+^S \times \mathbb{R} \times \mathbb{R} \mapsto \mathbb{R}$ *with* $f_z(p,v,n,L) = \frac{n}{n+1}\langle p,v\rangle + \max\left\{4\sqrt{\frac{\mathbb{V}(p,v)L}{n+1}}, \frac{64HL}{n+1}\right\} \cdot z$ *for some constant* $H > 0$ *and* $z \in \mathbb{R}$. *Then,* $f_z$ *satisfies*

*(i)* $f_z(p,v,n,L)$ *is non-decreasing in* $v(s)$ *for all* $p \in \Delta^S$, $\|v\|_\infty \leq 2H$, $L > 0$, $n \geq 3$ *and* $z \in [-1.5, 1.5]$

*(ii)* $f_z(p,v,n,L) \geq \frac{n}{n+1}\langle p,v\rangle + \left(3\sqrt{\frac{\mathbb{V}(p,v)L}{n+1}} + \frac{8HL}{n+1}\right) \cdot z$ *for* $z \in [1, 1.5]$.

*(iii)* $f_z(p,v,n,L) \leq \frac{n}{n+1}\langle p,v\rangle + \left(3\sqrt{\frac{\mathbb{V}(p,v)L}{n+1}} + \frac{8HL}{n+1}\right) \cdot z$ *for* $z \in [-1.5, -1]$.

*Proof.* It is obvious that $f_z(p,v,n,L)$ is continuous in $v(s)$ and not differentiable at only one point where $4\sqrt{\frac{\mathbb{V}(p,v)L}{n+1}} = \frac{64HL}{n+1}$. Therefore, to prove statement (i), we only need to show that $\frac{\partial f_z(p,v,n,L)}{\partial v(s)} \geq 0$. Specifically, we have

$$
\frac{\partial f_z(p,v,n,L)}{\partial v(s)} = \frac{n}{n+1} \cdot p(s) + \mathbb{1}\left\{4\sqrt{\frac{\mathbb{V}(p,v)L}{n+1}} \geq \frac{64HL}{n+1}\right\} \frac{4p(s)(v(s) - \langle p,v\rangle)L}{\sqrt{(n+1)\mathbb{V}(p,v)L}} \cdot z
$$

$$\overset{(a)}{\geq} \frac{n}{n+1} \cdot p(s) + \mathbb{1}\left\{4\sqrt{\frac{\mathbb{V}(p,v)L}{n+1}} \geq \frac{64HL}{n+1}\right\} \frac{-8HL}{\sqrt{(n+1)\mathbb{V}(p,v)L}} \cdot z$$

$$\overset{(b)}{\geq} p(s)\left(\frac{n}{n+1} - \frac{z}{2}\right)$$

$$\geq 0.$$

Here, The inequality (a) holds because $\|v\|_\infty \leq 2H$ and $v$ is non-negative, which means to have $v(s) - \langle p, v\rangle \geq -2H$. The inequality (b) above holds because when the condition inside indicator $\mathbb{1}\{\cdot\}$ holds, we will have $\sqrt{(n+1)\mathbb{V}(p,v)L} \geq 16HL$. The last inequality holds because we have $n \geq 3$ and $z \leq 1.5$. Therefore, $f_z(p,v,n,L)$ is non-decreasing in $v(s)$.

For the statement (ii), we consider two cases. First, when $4\sqrt{\frac{\mathbb{V}(p,v)L}{n+1}} \geq \frac{64HL}{n+1}$ holds, we have $\frac{8HL}{n+1} \leq \frac{1}{2}\sqrt{\frac{\mathbb{V}(p,v)L}{n+1}}$, which means to have

$$\frac{n}{n+1}\langle p, v\rangle + \left(3\sqrt{\frac{\mathbb{V}(p,v)L}{n+1}} + \frac{8HL}{n+1}\right) \cdot z \leq \frac{n}{n+1}\langle p, v\rangle + \frac{7z}{2}\sqrt{\frac{\mathbb{V}(p,v)L}{n+1}} \leq f_z(p,v,n,L).$$

When $4\sqrt{\frac{\mathbb{V}(p,v)L}{n+1}} \leq \frac{64HL}{n+1}$ holds, we have $3\sqrt{\frac{\mathbb{V}(p,v)L}{n+1}} \leq \frac{48HL}{n+1}$, which similarly leads to

$$\frac{n}{n+1}\langle p, v\rangle + \left(3\sqrt{\frac{\mathbb{V}(p,v)L}{n+1}} + \frac{8HL}{n+1}\right) \cdot z \leq f_z(p,v,n,L).$$

The state (iii) can be shown similarly and thus the proof is complete. $\qquad\square$

**Lemma 11.** *Condition on history $\mathcal{H}_H^{k-1}$, if $\mathcal{G}_{k,\mathrm{Ho}}$ holds and Bernstein-based noise is applied, then $\overline{V}_{h,k}$ is optimisitic with constant probability for any $h \in [H]$. Specifically, we have*

$$\mathbb{P}\left(\overline{V}_{h,k}(s) \geq V_h^*(s), \forall h \in [H], s \in \mathcal{S} \mid \mathcal{H}_H^{k-1}, \mathcal{G}_{k,\mathrm{Be}}\right) \geq \Phi(1.5) - \Phi(1) := C_{\mathrm{Be}}$$

*Proof.* Similar to what we have discussed in the proof of Lemma 9, under event $\mathcal{E}_k^w$, we have $\hat{z}_k \in [1, 1.5]$ with probability at least $\Phi(1.5) - \Phi(1) = C_{\mathrm{Be}}$. Then, we will show that $\overline{Q}_{h,k}(s,a) \geq Q_h^*(s,a)$ for any $h$ with arbitrary $s, a$ and $\hat{z}_k \in [1, 1.5]$. The proof will use induction. For simplicity, let $L = \log(2HSAk^2)$.

For $h = H + 1$, the inequality holds trivially because both sides are 0. Then, by assuming $\overline{Q}_{h+1,k}(s,a) \geq Q_h^*(s,a)$ for any $(s,a)$ such that $n_k(h,s,a) \geq 3$, we have

$$\overline{Q}_{h,k}(s,a) = \hat{R}_{h,s,a}^k + \left\langle \hat{P}_{h,s,a}^k, \overline{V}_{h+1,k}\right\rangle + \sigma_{\mathrm{Be}}^k(h,s,a)\hat{z}_k$$

$$\geq R_{h,s,a} + \left\langle \hat{P}_{h,s,a}^k, \overline{V}_{h+1,k}\right\rangle + \left(4\sqrt{\frac{\mathbb{V}\left(\tilde{P}_{h,s,a}^k, \overline{V}_{h+1,k}\right)L}{n_k(h,s,a)+1}} + \frac{64HL}{n_k(h,s,a)+1}\right) \cdot \hat{z}_k$$

(Replace $\hat{R}_{h,s,a}$ by $R_{h,s,a}$ through applying event $\mathcal{E}_k^1$ defined in (19))

$$\geq R_{h,s,a} + \left\langle \hat{P}_{h,s,a}^k, \overline{V}_{h+1,k}\right\rangle + \max\left\{4\sqrt{\frac{\mathbb{V}\left(\tilde{P}_{h,s,a}^k, \overline{V}_{h+1,k}\right)L}{n_k(h,s,a)+1}}, \frac{64HL}{n_k(h,s,a)+1}\right\} \cdot \hat{z}_k$$

$$\overset{(a)}{\geq} R_{h,s,a} + \left\langle \hat{P}_{h,s,a}^k, V_{h+1}^*\right\rangle + \max\left\{4\sqrt{\frac{\mathbb{V}\left(\tilde{P}_{h,s,a}^k, V_{h+1}^*\right)L}{n_k(h,s,a)+1}}, \frac{64HL}{n_k(h,s,a)+1}\right\} \cdot \hat{z}_k$$

$$\geq R_{h,s,a} + \left\langle \hat{P}_{h,s,a}^k, V_{h+1}^*\right\rangle + \left(3\sqrt{\frac{\mathbb{V}\left(\tilde{P}_{h,s,a}^k, V_{h+1}^*\right)L}{n_k(h,s,a)+1}} + \frac{8HL}{n_k(h,s,a)+1}\right) \cdot \hat{z}_k$$

(By applying statement (ii) of Lemma 10)

$$\geq R_{h,s,a} + \left\langle \hat{P}^k_{h,s,a}, V^*_{h+1} \right\rangle + \sqrt{\frac{6\mathbb{V}\left(\tilde{P}^k_{h,s,a}, V^*_{h+1}\right)}{n_k(h,s,a)+1} + \frac{8HL}{n_k(h,s,a)+1}} \qquad \text{(Since } \hat{z}_k \geq 1\text{)}$$

$$\geq R_{h,s,a} + \langle P_{h,s,a}, V^*_{h+1} \rangle \qquad \text{(By applying event } \mathcal{E}^2_k \text{ defined in (20))}$$

$$= Q^*_h(s,a).$$

Here, the above inequality (a) holds by applying inductive hypothesis and statement (i) in Lemma 10. It is applicable because when $\mathcal{E}^w_k$ holds, and by the clipping function, $\left\| \overline{V}_{h+1,k} \right\|_\infty \leq 2H$. When $n_k(h,s,a) < 3$, $\overline{Q}_{h,k}(s,a) \geq Q^*_h(s,a)$ holds trivially because $Q^*_h(s,a) \leq H$ by definition. Therefore, the induction is complete.

Now, for arbitrary $(k,h,s)$, set $a = \mathrm{argmax}_{a\in\mathcal{A}} \overline{Q}_{h,k}(s,a)$ and we have

$$\overline{V}_{h,k}(s) = \mathrm{clip}_{2(H-h+1)}\left(\max_{a\in\mathcal{A}} \overline{Q}_{h,k}(s,a)\right)$$

$$\geq \min\left\{2(H-h+1), \max_{a\in\mathcal{A}} \overline{Q}_{h,k}(s,a)\right\}$$

$$\geq \min\left\{(H-h+1), \overline{Q}_{h,k}(s, \pi^*_h(s))\right\}$$

$$\geq \min\left\{(H-h+1), Q^*_{h,k}(s, \pi^*_h(s))\right\}$$

$$\geq V^*_{h,k}(s)$$

$\square$

# D  Pessimism

Similar to what we have proved in Section C, in this section we will prove that for both types of noise, $\overline{V}_{h,k}$ is pessimistic with constant probability under certain conditions.

## D.1  Hoeffding-type Noise

**Lemma 12.** *Condition on history $\mathcal{H}^{k-1}_H$, if $\mathcal{G}_{k,\mathrm{Ho}}$ holds and Hoeffding-based noise is applied, then $\overline{V}_{h,k}$ is optimisitic with constant probability for any $h \in [H]$. Specifically, we have*

$$\mathbb{P}\left(\overline{V}_{h,k}(s) \leq V^*_h(s), \forall h \in [H], s \in \mathcal{S} \mid \mathcal{H}^{k-1}_H, \mathcal{G}_{k,\mathrm{Ho}}\right) \geq \Phi(1.9) - \Phi(1) := C_{\mathrm{Ho}}.$$

*Proof.* We will show that if $\hat{z}_k \leq -1$, then for all $h \in [H]$ and $s \in \mathcal{S}$, we have $\overline{V}_{h,k}(s) \leq V^*_h(s)$. The proof will use induction and the argument is true for $h = H+1$ as $\overline{V}_{H+1,k}(s) = V^*_{H+1}(s) = 0$. Suppose the argument is true for timestep $h+1$ and we consider timestep $h$. Set $a = \mathrm{argmax}_{a\in\mathcal{A}} \overline{Q}_{h,k}(s,a)$.

$$\overline{V}_{h,k}(s) = \mathrm{clip}_{2(H-h+1)}\left(\overline{Q}_{h,k}(s,a)\right)$$

$$\leq \max\left\{-2(H-h+1), \overline{Q}_{h,k}(s,a)\right\}$$

$$\leq \max\left\{-(H-h+1), \overline{Q}_{h,k}(s,a)\right\}$$

$$\leq \max\left\{-(H-h+1), \hat{R}^k_{h,s,a} + \left\langle \hat{P}^k_{h,s,a}, \overline{V}_{h+1,k}\right\rangle + \sigma^k_{\mathrm{ty}}(h,s,a)\hat{z}_k\right\}$$

$$\leq \max\left\{-(H-h+1), \hat{R}^k_{h,s,a} + \left\langle \hat{P}^k_{h,s,a}, V^*_{h+1,k}\right\rangle + \sigma^k_{\mathrm{ty}}(h,s,a)\hat{z}_k\right\}$$
$$\text{(Induction Hypothesis)}$$

$$\leq \max\left\{-(H-h+1), R^k_{h,s,a} + \left\langle P^k_{h,s,a}, V^*_{h+1,k}\right\rangle\right\} \quad \text{(Since } \hat{M}^k \in \mathcal{M}^k_{\mathrm{Ho}} \text{ and } \hat{z}_k \leq -1\text{)}$$

$$\leq \max\left\{-(H-h+1), Q^*_h(s,a)\right\}$$

$$\leq \max\left\{-(H-h+1), \max_{a\in\mathcal{A}} Q^*_h(s,a)\right\}$$

$$\leq V^*_h(s).$$

Then by induction we have that the optimism is achieved for all $h \in [H]$ and $s \in \mathcal{S}$ simultaneously. By using argument similar to the proof of Lemma 9, we can see that when $\hat{z}_k \leq -1$, we have $\overline{V}_{h,k}(s) \leq V_h^*(s)$ an this hold simultaneously for any $h \in [H]$, $s \in \mathcal{S}$. Furtherm as stated in Definition 2, we have $|\hat{z}_k| \leq \sqrt{\log(40k^4)}$ under event $\mathcal{E}_k^w$ and numerically, $\sqrt{\log(40k^4)} \geq 1.9$. Therefore, the probability that $\hat{z}_k \leq -1$ under $\mathcal{E}_k^w$ is at least

$$\mathbb{P}\left(\hat{z}_k \leq -1 \mid \mathcal{H}_H^{k-1}, \mathcal{G}_{k,\mathrm{Ho}}\right) = \frac{\Phi(1.9) - \Phi(1)}{\Phi(1.9) - \Phi(-1.9)} \geq \Phi(1.9) - \Phi(1) = C_{\mathrm{Ho}}.$$

Thus, we can conclude that

$$\mathbb{P}\left(\overline{V}_{h,k}(s) \leq V_h^*(s), \forall h \in [H], s \in \mathcal{S} \mid \mathcal{H}_H^{k-1}, \mathcal{G}_{k,\mathrm{Ho}}\right) \geq C_{\mathrm{Ho}}.$$

$\square$

### D.2 Bernstein-type Noise

**Lemma 13.** *Condition on history $\mathcal{H}_H^{k-1}$, if $\mathcal{G}_{k,\mathrm{Ho}}$ holds and Bernstein-based noise is applied, then $\overline{V}_{h,k}$ is pessimistic with constant probability for any $h \in [H]$. Specifically, we have*

$$\mathbb{P}\left(\overline{V}_{h,k}(s) \leq V_h^*(s), \forall h \in [H], s \in \mathcal{S} \mid \mathcal{H}_H^{k-1}, \mathcal{G}_{k,\mathrm{Be}}\right) \geq C_{\mathrm{Be}}$$

*Proof.* Similar to what we have discussed in the proof of Lemma 12, under event $\mathcal{E}_k^w$, we have $\hat{z}_k \in [-1.5, -1]$ with probability at least $\Phi(1.5) - \Phi(1) = C_{\mathrm{Be}}$. Then, we will show that $\overline{Q}_{h,k}(s,a) \leq Q_h^*(s,a)$ for any $h$ with arbitrary $s, a$ and $\hat{z}_k \in [-1.5, -1]$. The proof will go by induction. For simplicity, let $L = \log(2HSAk^2)$.

For $h = H + 1$, the inequality holds trivially because both sides are 0. Then, by assuming $\overline{Q}_{h+1,k}(s,a) \leq Q_h^*(s,a)$ for any $(s,a)$ such that $n_k(h,s,a) \geq 3$, we have

$$\overline{Q}_{h,k}(s,a) = \hat{R}_{h,s,a}^k + \left\langle \hat{P}_{h,s,a}^k, \overline{V}_{h+1,k} \right\rangle + \sigma_{\mathrm{Be}}^k(h,s,a)\hat{z}_k$$

$$\leq R_{h,s,a} + \left\langle \hat{P}_{h,s,a}^k, \overline{V}_{h+1,k} \right\rangle - \left(4\sqrt{\frac{\mathbb{V}\left(\tilde{P}_{h,s,a}^k, \overline{V}_{h+1,k}\right)L}{n_k(h,s,a)+1}} + \frac{64HL}{n_k(h,s,a)+1}\right)$$

(Replace $\hat{R}_{h,s,a}$ by $R_{h,s,a}$ through applying event $\mathcal{E}_k^1$ defined in (19))

$$\overset{(a)}{\leq} R_{h,s,a} + \left\langle \hat{P}_{h,s,a}^k, V_{h+1}^* \right\rangle - \max\left\{4\sqrt{\frac{\mathbb{V}\left(\tilde{P}_{h,s,a}^k, V_{h+1}^*\right)L}{n_k(h,s,a)+1}}, \frac{64HL}{n_k(h,s,a)+1}\right\}$$

$$\leq R_{h,s,a} + \left\langle \hat{P}_{h,s,a}^k, V_{h+1}^* \right\rangle - \left(3\sqrt{\frac{\mathbb{V}\left(\tilde{P}_{h,s,a}^k, V_{h+1}^*\right)L}{n_k(h,s,a)+1}} + \frac{8HL}{n_k(h,s,a)+1}\right)$$

(By applying statement (iii) of Lemma 10)

$$\leq R_{h,s,a} + \left\langle \hat{P}_{h,s,a}^k, V_{h+1}^* \right\rangle - \sqrt{\frac{6\mathbb{V}\left(\tilde{P}_{h,s,a}^k, V_{h+1}^*\right)}{n_k(h,s,a)+1}} - \frac{8HL}{n_k(h,s,a)+1}$$

$$\leq R_{h,s,a} + \left\langle P_{h,s,a}, V_{h+1}^* \right\rangle \qquad \text{(By applying event $\mathcal{E}_k^2$ defined in (20))}$$

$$= Q_h^*(s,a).$$

Here, the above inequality (a) holds by applying inductive hypothesis and statement (i) in Lemma 10. It is applicable because when $\mathcal{E}_k^w$ holds, by the clipping function, $\left\|\overline{V}_{h+1,k}\right\|_\infty \leq 2H$. When $n_k(h,s,a) < 3$, $\overline{Q}_{h,k}(s,a) \leq Q_h^*(s,a)$ holds trivially because $0 \leq Q_h^*(s,a) \leq H$ by definition. Therefore, the induction is complete.

Now, for arbitrary $(k,h,s)$, set $a = \mathrm{argmax}_{a \in \mathcal{A}} \overline{Q}_{h,k}(s,a)$ and we have

$$\overline{V}_{h,k}(s) = \mathrm{clip}_{2(H-h+1)}\left(\overline{Q}_{h,k}(s,a)\right)$$

$$\leq \max\left\{-2(H-h+1), \overline{Q}_{h,k}(s,a)\right\}$$
$$\leq \max\left\{-(H-h+1), \overline{Q}_{h,k}(s,a)\right\}$$
$$\leq \max\left\{-(H-h+1), Q^*_{h,k}(s,a)\right\}$$
$$\leq \max\left\{-(H-h+1), Q^*_{h,k}(s,\pi^*_h(s))\right\}$$
$$\leq V^*_{h,k}(s).$$

$\square$

# E  Regret Decomposition

In this section, we prove the multiple lemmas necessary for bounding the regret. The regret is mainly composed of two terms, the pessimism term and the estimation error term. The pessimism term, $V^*_{1,k}(s^k_1) - \overline{V}_{1,k}(s^k_1)$, measures how much regret is due to the value the algorithm uses, $\overline{V}_{1,k}$, is smaller than the true value, $V^*_{1,k}(s^k_1)$. The estimation error term, $\overline{V}_{1,k}(s^k_1) - V^{\pi^k}_{1,k}(s^k_1)$ measure how much regret is due to the value, $\overline{V}_{1,k}$, does not estimate $V^{\pi^k}_{1,k}(s^k_1)$, the true value of the policy $\pi^k$ accurately.

We first introduce a few definitions key to this section. In this section, we omit $k$ if it is clear from the context. Let $a^k_h = \pi^k_h(s^k_h)$ unless specified otherwise.

**Definition 5.** *Let* $\mathcal{P}^k_{h,s,a} = \left\langle \hat{P}^k_{h,s,a} - P_{h,s,a}, V^*_{h+1} \right\rangle$ *and* $\mathcal{R}^k_{h,s,a} = \hat{R}^k_{h,s,a} - R_{h,s,a}$.

**Definition 6** ($\underline{M}^k_{\text{ty}}$ *and* $\underline{V}_{h,k}$)**.** *Given history* $\mathcal{H}^{k-1}_H$ *(defined in equation* (24)*),* $\hat{P}^k$ *and* $\hat{R}^k$*, we define* $\underline{w}^k_{\text{ty}}(h,s,a) = -\gamma^k_{\text{ty}}(h,s,a)$ *and* $\underline{V}_{h,k}$ *be the value function obtained by running policy* $\pi^k$ *on the MDP* $\underline{M}^k_{\text{ty}} = (H, \mathcal{S}, \mathcal{A}, \hat{P}^k, \hat{R}^k + \underline{w}^k_{\text{ty}}, s^k_1)$ *plus a magnitude clipping with threshold* $2(H-h+1)$.

**Definition 7** ($\overline{\overline{M}}^k_{\text{ty}}$ *and* $\overline{\overline{V}}_{h,k}$)**.** *Given history* $\mathcal{H}^{k-1}_H$ *(defined in equation* (24)*),* $\hat{P}^k$ *and* $\hat{R}^k$*, we define* $\overline{w}^k_{\text{ty}}(h,s,a) = \gamma^k_{\text{ty}}(h,s,a)$ *and* $\overline{\overline{V}}_{h,k}$ *be the value function obtained by running policy* $\pi^k$ *on the MDP* $\overline{\overline{M}}^k_{\text{ty}} = (H, \mathcal{S}, \mathcal{A}, \hat{P}^k, \hat{R}^k + \overline{w}^k_{\text{ty}}, s^k_1)$ *plus a magnitude clipping with threshold* $2(H-h+1)$.

*Similar to Lemma* 8*, we can also show that under good event* $\mathcal{G}_k$ *and* $\mathcal{E}^{th}_{h,k}$*, no clipping happens on* $s^k_h$ *for* $\underline{V}_{h,k}(s^k_h)$ *and* $\overline{\overline{V}}_{h,k}(s^k_h)$.

**Lemma 14.** *Under the good event* $\mathcal{G}_k$*, we have* $\underline{V}_{h,k}(s) \leq \overline{V}_{h,k}(s) \leq \overline{\overline{V}}_{h,k}(s)$ *for all* $h \in [H]$*,* $s \in \mathcal{S}$.

*Proof.* This is an immediate result by noticing that under good event $\mathcal{G}_k$, we have $\underline{w}^k_{\text{ty}}(h,s,a) \leq w^k_{\text{ty}}(h,s,a) \leq \overline{w}^k_{\text{ty}}(h,s,a)$ for all $h \in [H]$ and $s \in \mathcal{S}$. $\square$

**Definition 8.** *Define* $\underline{\delta}^\pi_h(s_h), \overline{\delta}^\pi_h(s_h), \overline{\overline{\delta}}^\pi_h(s_h), \delta^\pi_h(s_h), \underline{\delta}_h(s_h), \overline{\delta}_h(s_h)$ *and* $\overline{\overline{\delta}}_h(s_h)$ *as*
$$\underline{\delta}^\pi_h(s_h) = \underline{V}_h(s_h) - V^\pi_h(s_h),$$
$$\overline{\delta}^\pi_h(s_h) = \overline{V}_h(s_h) - V^\pi_h(s_h),$$
$$\overline{\overline{\delta}}^\pi_h(s_h) = \overline{\overline{V}}_h(s_h) - V^\pi_h(s_h),$$
$$\delta^\pi_h(s_h) = V^*_h(s_h) - V^\pi_h(s_h),$$
$$\underline{\delta}_h(s_h) = \underline{V}_h(s_h) - V^*_h(s_h),$$
$$\overline{\delta}_h(s_h) = \overline{V}_h(s_h) - V^*_h(s_h),$$
$$\overline{\overline{\delta}}_h(s_h) = \overline{\overline{V}}_h(s_h) - V^*_h(s_h).$$

**Definition 9.** *We denote the history trajectory* $\overline{\mathcal{H}}^k_h = \mathcal{H}^k_h \cup \{\hat{z}_k\}$*. With filtration sets* $\left\{\overline{\mathcal{H}}^k_h\right\}_{h,k}$*, we define the following sequences:*
$$\mathcal{M}_{\delta_h(s_h)} = \mathbb{1}\{\mathcal{G}_k \cap \mathcal{E}^{cum}_{h,k}\}\left[\langle P_{h,s_h,a_h}, \delta_{h+1}\rangle - \delta_{h+1}(s_{h+1})\right],$$
*where* $\delta \in \{\underline{\delta}^\pi, \overline{\delta}^\pi, \overline{\overline{\delta}}^\pi, \delta^\pi, \underline{\delta}, \overline{\delta}, \overline{\overline{\delta}}\}$*. We will show the sequences are martingales in Lemma* 22*.*

Finally, the regret can be decomposed as

$$\text{Regret}\left(M, K, \mathsf{SSR}_{\text{ty}}\right)$$

$$= \sum_{k=1}^{K}\left(V_1^*(s_1^k) - V_{1,k}^{\pi^k}(s_1^k)\right)$$

$$= \sum_{k=1}^{K}\mathbb{1}\left(\mathcal{C}_{\text{ty}}^k\right)\left(V_1^*(s_1^k) - V_{1,k}^{\pi^k}(s_1^k)\right) + \sum_{k=1}^{K}\mathbb{1}\left(\left(\mathcal{C}_{\text{ty}}^k\right)^c\right)\left(V_1^*(s_1^k) - V_{1,k}^{\pi^k}(s_1^k)\right)$$

$$= \sum_{k=1}^{K}\mathbb{1}\left\{\mathcal{C}_{\text{ty}}^k\right\}\left(\underbrace{V_{1,k}^*(s_1^k) - \overline{V}_{1,k}(s_1^k)}_{\text{pessimism term} = -\overline{\delta}_{1,k}(s_1^k)} + \underbrace{\overline{V}_{1,k}(s_1^k) - V_{1,k}^{\pi^k}(s_1^k)}_{\text{estimation error term} = \overline{\delta}_{1,k}^{\pi^k}(s_1^k)}\right)$$

$$+ \underbrace{\sum_{k=1}^{K}\mathbb{1}\left(\left(\mathcal{C}_{\text{ty}}^k\right)^c\right)\left(V_1^*(s_1^k) - V_{1,k}^{\pi^k}(s_1^k)\right)}_{\text{(a)}}.$$

By Lemma 5 and 6, we know that

$$\mathbb{E}\left[\sum_{k=1}^{K}\mathbb{1}\left(\left(\mathcal{C}_{\text{ty}}^k\right)^c\right)\right] = \sum_{k=1}^{K}\mathbb{P}\left(\left(\mathcal{C}_{\text{ty}}^k\right)^c\right) \leq \sum_{k=1}^{\infty}\mathbb{P}\left(\left(\mathcal{C}_{\text{ty}}^k\right)^c\right) \leq \frac{\pi^2}{3}.$$

Therefore, by standard Hoeffding's inequality, it holds with probability at least $1 - \delta$ that

$$\sum_{k=1}^{K}\mathbb{1}\left(\left(\mathcal{C}_{\text{ty}}^k\right)^c\right) \leq \frac{\pi^2}{3} + \sqrt{\frac{\log(1/\delta)}{2K}}.$$

Since the value functions of true MDP is bounded in $[0, H]$, with probability at least $1 - \delta$, we have

$$\text{(a)} \leq H\sum_{k=1}^{K}\mathbb{1}\left(\left(\mathcal{C}_{\text{ty}}^k\right)^c\right) \leq \frac{\pi^2 H}{3} + H\sqrt{\frac{\log(1/\delta)}{2K}} = \widetilde{O}\left(H\right).$$

Further, notice that the good event $\mathcal{G}_k = \mathcal{C}_{\text{ty}}^k \cap \mathcal{E}_k^w$ and by Lemma 7, we have $\sum_{k=1}^{\infty}\mathbb{P}\left(\left(\mathcal{E}_k^w\right)^c\right) \leq \frac{\pi^2}{3}$. Therefore, we can similarly address the regret incurred by $\left(\mathcal{E}_k^w\right)^c$ as the bound for term (a). As a result, it will be sufficient to only consider $\mathbb{1}\left\{\mathcal{G}_k\right\}\left(V_{1,k}^*(s_1^k) - V_{1,k}^{\pi^k}(s_1^k)\right)$ when bounding pessimism and estimation error terms. That is, with probability at least $1 - \delta$, it holds that

$$\text{Regret}\left(M, K, \mathsf{SSR}_{\text{ty}}\right) \leq \sum_{k=1}^{K}\mathbb{1}\left\{\mathcal{G}_k\right\}\left(\left|\overline{\delta}_{1,k}(s_1^k)\right| + \left|\overline{\delta}_{1,k}^{\pi^k}(s_1^k)\right|\right) + \widetilde{O}\left(H\right). \tag{25}$$

Then, we decompose the estimation error term in Section E.2. We decompose the pessimism term in Section E.1. We combine the decomposition of the pessimism term and the estimation error term in Section E.3.

## E.1 Pessimism Term

**Lemma 15.** *Let* $C_1 = \max\left\{\frac{1}{\Phi(1.9) - \Phi(1)}, \frac{1}{\Phi(1.5) - \Phi(1)}\right\} = \frac{1}{\Phi(1.5) - \Phi(1)} \approx 10.9$. *Then, for any* $h, k, s_h^k$ *and the type of noise we used, under the good event* $\mathcal{G}_k$, *the following bound holds,*

$$\mathbb{1}\left\{\mathcal{G}_k\right\}\left|\overline{\delta}_{h,k}(s_h^k)\right| \leq \mathbb{1}\left\{\mathcal{G}_k\right\}C_1\left(\left|\overline{\overline{\delta}}_{h,k}^{\pi^k}(s_h^k)\right| + \left|\underline{\delta}_{h,k}^{\pi^k}(s_h^k)\right|\right). \tag{26}$$

*Proof.* Let $\mathcal{O}_k$ be the event that $\overline{V}_{h,k}(s) \geq V_h^*(s)$ simultaneously for all $s \in \mathcal{S}$ and $h \in [H]$. By Lemma 9 and 11, we know that $\mathbb{P}\left(\mathcal{O}_k \mid \mathcal{H}_H^{k-1}, \mathcal{G}_k\right) \geq \min\left\{\Phi(1.9) - \Phi(1), \Phi(1.5) - \Phi(1)\right\} = \Phi(1.5) - \Phi(1)$, which means $\frac{1}{\mathbb{P}(\mathcal{O}_k)} \leq C_1$ regardless the type of noise used.

The definition of $\mathcal{O}_k$ implies $V_h^* \leq \mathbb{E}\left[\overline{V}_{h,k} \mid \mathcal{O}_k, \mathcal{H}_H^{k-1}, \mathcal{G}_k\right]$. Meanwhile, notice that

$$
\begin{aligned}
&\mathbb{1}\left\{\mathcal{G}_k\right\}\left(\mathbb{E}\left[\overline{V}_{h,k} \mid \mathcal{H}_H^{k-1}, \mathcal{G}_k\right] - \underline{V}_{h,k}\right)\\
=&\mathbb{1}\left\{\mathcal{G}_k\right\}\mathbb{P}\left(\mathcal{O}_k \mid \mathcal{H}_H^{k-1}, \mathcal{G}_k\right)\left(\mathbb{E}\left[\overline{V}_{h,k} \mid \mathcal{O}_k, \mathcal{H}_H^{k-1}, \mathcal{G}_k\right] - \underline{V}_{h,k}\right)\\
&+\mathbb{1}\left\{\mathcal{G}_k\right\}\mathbb{P}\left((\mathcal{O}_k)^c \mid \mathcal{H}_H^{k-1}, \mathcal{G}_k\right)\underbrace{\left(\mathbb{E}\left[\overline{V}_{h,k} \mid (\mathcal{O}_k)^c, \mathcal{H}_H^{k-1}, \mathcal{G}_k\right] - \underline{V}_{h,k}\right)}_{(a)\geq 0}\\
\geq&\mathbb{1}\left\{\mathcal{G}_k\right\}\mathbb{P}\left(\mathcal{O}_k \mid \mathcal{H}_H^{k-1}, \mathcal{G}_k\right)\left(\mathbb{E}\left[\overline{V}_{h,k} \mid \mathcal{O}_k, \mathcal{H}_H^{k-1}, \mathcal{G}_k\right] - \underline{V}_{h,k}\right)\\
\implies & \mathbb{1}\left\{\mathcal{G}_k\right\}\left(\mathbb{E}\left[\overline{V}_{h,k} \mid \mathcal{O}_k, \mathcal{H}_H^{k-1}, \mathcal{G}_k\right] - \underline{V}_{h,k}\right) \leq \mathbb{1}\left\{\mathcal{G}_k\right\}C_1\left(\mathbb{E}\left[\overline{V}_{h,k} \mid \mathcal{H}_H^{k-1}, \mathcal{G}_k\right] - \underline{V}_{h,k}\right).
\end{aligned}
$$

Here, we have term $(a) \geq 0$ since $\underline{V}_{h,k} \leq \overline{V}_{h,k}$ under event $\mathcal{G}_k$, by Lemma 14.

Therefore, we have

$$
\begin{aligned}
\mathbb{1}\{\mathcal{G}_k\}\left(V_h^*(s_h^k) - \overline{V}_{h,k}(s_h^k)\right) &\leq \mathbb{1}\left\{\mathcal{G}_k\right\}\left(\mathbb{E}\left[\overline{V}_{h,k} \mid \mathcal{O}_k, \mathcal{H}_H^{k-1}, \mathcal{G}_k\right](s_h^k) - \overline{V}_{h,k}(s_h^k)\right)\\
&\leq \mathbb{1}\left\{\mathcal{G}_k\right\}\left(\mathbb{E}\left[\overline{V}_{h,k} \mid \mathcal{O}_k, \mathcal{H}_H^{k-1}, \mathcal{G}_k\right](s_h^k) - \underline{V}_{h,k}(s_h^k)\right)\\
&\leq \mathbb{1}\left\{\mathcal{G}_k\right\}C_1\left(\mathbb{E}\left[\overline{V}_{h,k} \mid \mathcal{H}_H^{k-1}, \mathcal{G}_k\right](s_h^k) - \underline{V}_{h,k}(s_h^k)\right). \quad (27)
\end{aligned}
$$

We can similarly use constant probability pessimism shown in Lemma 12 and 13. In particular, let $\mathcal{N}_k$ be the event that $\overline{V}_{h,k}(s) \leq V_h^*(s)$ for all $s \in \mathcal{S}$ and $h \in [H]$. Then, we have

$$
\begin{aligned}
&\mathbb{1}\left\{\mathcal{G}_k\right\}\left(\mathbb{E}\left[\overline{V}_{h,k} \mid \mathcal{H}_H^{k-1}, \mathcal{G}_k\right] - \overline{\overline{V}}_{h,k}\right)\\
=&\mathbb{1}\left\{\mathcal{G}_k\right\}\mathbb{P}\left(\mathcal{N}_k \mid \mathcal{H}_H^{k-1}, \mathcal{G}_k\right)\left(\mathbb{E}\left[\overline{V}_{h,k} \mid \mathcal{N}_k, \mathcal{H}_H^{k-1}, \mathcal{G}_k\right] - \overline{\overline{V}}_{h,k}\right)\\
&+\mathbb{1}\left\{\mathcal{G}_k\right\}\mathbb{P}\left((\mathcal{N}_k)^c \mid \mathcal{H}_H^{k-1}, \mathcal{G}_k\right)\underbrace{\left(\mathbb{E}\left[\overline{V}_{h,k} \mid (\mathcal{N}_k)^c, \mathcal{H}_H^{k-1}, \mathcal{G}_k\right] - \overline{\overline{V}}_{h,k}\right)}_{(b)\leq 0}\\
\leq&\mathbb{1}\left\{\mathcal{G}_k\right\}\mathbb{P}\left(\mathcal{N}_k \mid \mathcal{H}_H^{k-1}, \mathcal{G}_k\right)\left(\mathbb{E}\left[\overline{V}_{h,k} \mid \mathcal{N}_k, \mathcal{H}_H^{k-1}, \mathcal{G}_k\right] - \overline{\overline{V}}_{h,k}\right)\\
\implies & \mathbb{1}\left\{\mathcal{G}_k\right\}\left(\mathbb{E}\left[\overline{V}_{h,k} \mid \mathcal{N}_k, \mathcal{H}_H^{k-1}, \mathcal{G}_k\right] - \overline{\overline{V}}_{h,k}\right) \geq \mathbb{1}\left\{\mathcal{G}_k\right\}C_1\left(\mathbb{E}\left[\overline{V}_{h,k} \mid \mathcal{H}_H^{k-1}, \mathcal{G}_k\right] - \overline{\overline{V}}_{h,k}\right).
\end{aligned}
$$

Thus, we have

$$
\begin{aligned}
\mathbb{1}\{\mathcal{G}_k\}\left(V_h^*(s_h^k) - \overline{V}_{h,k}(s_h^k)\right) &\geq \mathbb{1}\left\{\mathcal{G}_k\right\}\left(\mathbb{E}\left[\overline{V}_{h,k} \mid \mathcal{N}_k, \mathcal{H}_H^{k-1}, \mathcal{G}_k\right](s_h^k) - \overline{\overline{V}}_{h,k}(s_h^k)\right)\\
&\geq \mathbb{1}\left\{\mathcal{G}_k\right\}C_1\left(\mathbb{E}\left[\overline{V}_{h,k} \mid \mathcal{H}_H^{k-1}, \mathcal{G}_k\right](s_h^k) - \overline{\overline{V}}_{h,k}(s_h^k)\right). \quad (28)
\end{aligned}
$$

Since good event $\mathcal{G}_k$ implies $\underline{V}_{h,k} \leq \overline{V}_{h,k} \leq \overline{\overline{V}}_{h,k}$ by Lemma 14, the RHS of (27) is non-negative and the RHS of (28) is non-positive. Therefore, we can then conclude

$$
\begin{aligned}
&\mathbb{1}\left\{\mathcal{G}_k\right\}\left|V_h^*(s_h^k) - \overline{V}_{h,k}(s_h^k)\right|\\
\leq&\mathbb{1}\left\{\mathcal{G}_k\right\}C_1\left(\left(\mathbb{E}\left[\overline{V}_{h,k} \mid \mathcal{H}_H^{k-1}, \mathcal{G}_k\right](s_h^k) - \underline{V}_{h,k}(s_h^k)\right) - \left(\mathbb{E}\left[\overline{V}_{h,k} \mid \mathcal{H}_H^{k-1}, \mathcal{G}_k\right](s_h^k) - \overline{\overline{V}}_{h,k}(s_h^k)\right)\right)\\
=&\mathbb{1}\left\{\mathcal{G}_k\right\}C_1\left(\overline{\overline{V}}_{h,k}(s_h^k) - \underline{V}_{h,k}(s_h^k)\right)\\
\leq&\mathbb{1}\left\{\mathcal{G}_k\right\}C_1\left(\left|\overline{\overline{\delta}}_{h,k}^{\pi^k}(s_h^k)\right| + \left|\underline{\delta}_{h,k}^{\pi^k}(s_h^k)\right|\right).
\end{aligned}
$$

$\square$

## E.2 Estimation Error Term

We first bound the estimation error of $\overline{\overline{V}}$, which can be regarded as the optimistic estimate used in UCB-type algorithms. For convenience, we will ignore notation $\mathbb{1}\left\{\mathcal{G}_k\right\}$ in this section since all statements are proved under the good event $\mathcal{G}_k$.

**Lemma 16.** *With probability at least $1 - \delta$, for all $(k, h, s_i^k)$, under the good event $\mathcal{G}_k$ it holds that*

$$\mathbb{1}\left\{\mathcal{E}_{h,k}^{cum}\right\}\left|\overline{\overline{\delta}}_{h,k}^{\pi^k}(s_h^k)\right|$$

$$\leq \mathbb{1}\left\{\mathcal{E}_{h,k}^{cum}\right\}\left(\left|\mathcal{P}_{h,s_h^k,a_h^k}^k + \mathcal{R}_{h,s_h^k,a_h^k}^k + \overline{w}_{\text{ty}}^k(h, s_h^k, a_h^k)\right| + \mathcal{M}_{\left|\overline{\overline{\delta}}_{h,k}^{\pi^k}(s_h^k)\right|} + \mathcal{M}_{\left|\overline{\delta}_{h,k}(s_h^k)\right|} + \frac{2SH^2L}{n_k(h, s_h^k, a_h^k)}\right)$$

$$+ \mathbb{1}\left\{\mathcal{E}_{h+1,k}^{cum}\right\}\left(\frac{C_1}{H}\left|\underline{\delta}_{h+1,k}^{\pi^k}(s_{h+1}^k)\right| + \frac{H+1+C_1}{H}\left|\overline{\overline{\delta}}_{h+1,k}^{\pi^k}(s_{h+1}^k)\right| + \frac{1}{H}\left|\overline{\delta}_{h+1,k}^{\pi^k}(s_{h+1}^k)\right|\right)$$

$$+ \mathbb{1}\left\{\mathcal{E}_{h,k}^{cum} \cap \left(\mathcal{E}_{h+1,k}^{th}\right)^c\right\}\left(\frac{C_1}{H}\left|\underline{\delta}_{h+1,k}^{\pi^k}(s_{h+1}^k)\right| + \frac{H+1+C_1}{H}\left|\overline{\overline{\delta}}_{h+1,k}^{\pi^k}(s_{h+1}^k)\right| + \frac{1}{H}\left|\overline{\delta}_{h+1,k}^{\pi^k}(s_{h+1}^k)\right|\right),$$

*where $L = \log(2HS^2AK/\delta)$.*

*Proof.* Since both $\overline{\overline{V}}$ and $V^{\pi^k}$ are obtained by choosing actions based on policy $\pi^k$ under event $\mathcal{G}_k$, we have

$$\mathbb{1}\left\{\mathcal{E}_{h,k}^{cum}\right\}\left|\overline{\overline{\delta}}_{h,k}^{\pi^k}(s_h^k)\right|$$

$$=\mathbb{1}\left\{\mathcal{E}_{h,k}^{cum}\right\}\left|\overline{\overline{V}}_{h,k}(s_h^k) - V_{h,k}^{\pi^k}(s_h^k)\right|$$

$$=\mathbb{1}\left\{\mathcal{E}_{h,k}^{cum}\right\}\left|\overline{\overline{Q}}_{h,k}(s_h^k, a_h^k) - Q_{h,k}^{\pi^k}(s_h^k, a_h^k)\right| \qquad \text{(Since no clipping under } \mathcal{E}_{h,k}^{cum} \text{ for } \overline{\overline{V}}_{h,k}(s_h^k))$$

$$=\mathbb{1}\left\{\mathcal{E}_{h,k}^{cum}\right\}\left|\hat{R}_{h,s_h^k,a_h^k}^k - R_{h,s_h^k,a_h^k} + \overline{w}_{\text{ty}}^k(h, s_h^k, a_h^k) + \left\langle\hat{P}_{h,s_h^k,a_h^k}^k, \overline{\overline{V}}_{h+1,k}\right\rangle - \left\langle P_{h,s_h^k,a_h^k}^k, V_{h+1,k}^{\pi^k}\right\rangle\right|$$

$$=\mathbb{1}\left\{\mathcal{E}_{h,k}^{cum}\right\}\left|\hat{R}_{h,s_h^k,a_h^k}^k - R_{h,s_h^k,a_h^k} + \overline{w}_{\text{ty}}^k(h, s_h^k, a_h^k) + \left\langle\hat{P}_{h,s_h^k,a_h^k}^k, \overline{\overline{V}}_{h+1,k}\right\rangle - \left\langle P_{h,s_h^k,a_h^k}, V_{h+1,k}^{\pi^k}\right\rangle\right.$$

$$\left.+ \left\langle\hat{P}_{h,s_h^k,a_h^k}^k - P_{h,s_h^k,a_h^k}, V_{h+1}^*\right\rangle - \left\langle\hat{P}_{h,s_h^k,a_h^k}^k - P_{h,s_h^k,a_h^k}, V_{h+1}^*\right\rangle\right|$$

$$\leq\mathbb{1}\left\{\mathcal{E}_{h,k}^{cum}\right\}\left(\left|\mathcal{P}_{h,s_h^k,a_h^k}^k + \mathcal{R}_{h,s_h^k,a_h^k}^k + \overline{w}_{\text{ty}}^k(h, s_h^k, a_h^k)\right| + \left\langle P_{h,s_h^k,a_h^k}, \left|\overline{\overline{V}}_{h+1,k} - V_{h+1,k}^{\pi^k}\right|\right\rangle\right)$$

$$+ \mathbb{1}\left\{\mathcal{E}_{h,k}^{cum}\right\}\left|\left\langle\hat{P}_{h,s_h^k,a_h^k}^k - P_{h,s_h^k,a_h^k}, \overline{\overline{V}}_{h+1,k} - V_{h+1}^*\right\rangle\right|$$

$$=\mathbb{1}\left\{\mathcal{E}_{h,k}^{cum}\right\}\left(\left|\mathcal{P}_{h,s_h^k,a_h^k}^k + \mathcal{R}_{h,s_h^k,a_h^k}^k + \overline{w}_{\text{ty}}^k(h, s_h^k, a_h^k)\right| + \left|\overline{\overline{\delta}}_{h+1,k}^{\pi^k}(s_{h+1}^k)\right| + \mathcal{M}_{\left|\overline{\overline{\delta}}_{h,k}^{\pi^k}(s_h^k)\right|}\right)$$

$$+ \mathbb{1}\left\{\mathcal{E}_{h,k}^{cum}\right\}\left|\left\langle\hat{P}_{h,s_h^k,a_h^k}^k - P_{h,s_h^k,a_h^k}, \overline{\overline{V}}_{h+1,k} - V_{h+1}^*\right\rangle\right|.$$

For the last term, we use Lemma 33 and then for $L = \log(2HS^2AK/\delta)$, with probability at least $1 - \delta$, we have

$$\left|\left\langle\hat{P}_{h,s_h^k,a_h^k}^k - P_{h,s_h^k,a_h^k}, \overline{\overline{V}}_{h+1,k} - V_{h+1}^*\right\rangle\right|$$

$$\leq \sum_{s_{h+1}\in\mathcal{S}}\left|\hat{P}_{h,s_h^k,a_h^k}^k(s_{h+1}) - P_{h,s_h^k,a_h^k}(s_{h+1})\right|\left|\overline{\overline{V}}_{h+1,k}(s_h + 1) - V_{h+1}^*(s_{h+1})\right|$$

$$\leq \sum_{s_{h+1}\in\mathcal{S}}\left(2\sqrt{\frac{P_{h,s_h^k,a_h^k}(s_{h+1})L}{n_k(h, s_h^k, a_h^k)}} + \frac{4L}{3n_k(h, s_h^k, a_h^k)}\right)\left|\overline{\overline{\delta}}_{h+1,k}(s_{h+1})\right|$$

$$= \sum_{s_{h+1}:P_{h,s_h^k,a_h^k}(s_{h+1})n_k(h,s_h^k,a_h^k)\geq 4LH^2}2P_{h,s_h^k,a_h^k}(s_{h+1})\sqrt{\frac{L}{P_{h,s_h^k,a_h^k}(s_{h+1})n_k(h, s_h^k, a_h^k)}}\left|\overline{\overline{\delta}}_{h+1,k}(s_{h+1})\right|$$

$$+ \sum_{s_{h+1}:P_{h,s_h^k,a_h^k}(s_{h+1})n_k(h,s_h^k,a_h^k)<4LH^2}2\sqrt{\frac{LP_{h,s_h^k,a_h^k}(s_{h+1})n_k(h, s_h^k, a_h^k)}{n_k(h, s_h^k, a_h^k)^2}}\left|\overline{\overline{\delta}}_{h+1,k}(s_{h+1})\right|$$

$$+ \frac{4SHL}{3n_k(h, s_h^k, a_h^k)}$$

$$
\leq \sum_{s_{h+1} \in \mathcal{S}} P_{h,s_h^k,a_h^k}(s_{h+1}) \frac{1}{H} \left| \bar{\bar{\bar{\delta}}}_{h+1,k}(s_{h+1}) \right| + \frac{4SHL + 2SH^2\sqrt{L}}{3n_k(h,s_h^k,a_h^k)}
$$

$$
\leq \frac{1}{H} \left| \bar{\bar{\bar{\delta}}}_{h+1,k}(s_{h+1}^k) \right| + \mathcal{M}_{\left| \bar{\bar{\bar{\delta}}}_{h,k}(s_h^k) \right|} + \frac{2SH^2L}{n_k(h,s_h^k,a_h^k)}
$$

$$
\leq \frac{1}{H} \left| \bar{\bar{\bar{\delta}}}_{h+1,k}^{\pi^k}(s_{h+1}^k) \right| + \frac{1}{H} \left| \bar{\bar{\delta}}_{h+1,k}^{\pi^k}(s_{h+1}^k) \right| + \frac{1}{H} \left| \bar{\delta}_{h+1,k}(s_{h+1}^k) \right| + \mathcal{M}_{\left| \bar{\bar{\bar{\delta}}}_{h,k}(s_h^k) \right|} + \frac{2SH^2L}{n_k(h,s_h^k,a_h^k)}
$$
$$
\text{(By triangle inequality)}
$$

$$
\leq \frac{1+C_1}{H} \left| \bar{\bar{\bar{\delta}}}_{h+1,k}^{\pi^k}(s_{h+1}^k) \right| + \frac{1}{H} \left| \bar{\delta}_{h+1,k}^{\pi^k}(s_{h+1}^k) \right| + \frac{C_1}{H} \left| \underline{\delta}_{h+1,k}^{\pi^k}(s_{h+1}^k) \right| + \mathcal{M}_{\left| \bar{\bar{\bar{\delta}}}_{h,k}(s_h^k) \right|} + \frac{2SH^2L}{n_k(h,s_h^k,a_h^k)}.
$$
$$
\text{(By using Lemma 15)}
$$

Combining the above two arguments, we can prove the argument:

$$
\mathbb{1}\left\{ \mathcal{E}_{h,k}^{cum} \right\} \left| \bar{\bar{\bar{\delta}}}_{h,k}^{\pi^k}(s_h^k) \right|
$$

$$
\leq \mathbb{1}\left\{ \mathcal{E}_{h,k}^{cum} \right\} \left( \left| \mathcal{P}_{h,s_h^k,a_h^k}^k + \mathcal{R}_{h,s_h^k,a_h^k}^k + \overline{w}_{\mathrm{ty}}^k(h,s_h^k,a_h^k) \right| + \mathcal{M}_{\left| \bar{\bar{\bar{\delta}}}_{h,k}^{\pi^k}(s_h^k) \right|} + \mathcal{M}_{\left| \bar{\bar{\bar{\delta}}}_{h,k}(s_h^k) \right|} + \frac{2SH^2L}{n_k(h,s_h^k,a_h^k)} \right)
$$

$$
+ \mathbb{1}\left\{ \mathcal{E}_{h,k}^{cum} \right\} \left( \frac{H+1+C_1}{H} \left| \bar{\bar{\bar{\delta}}}_{h+1,k}^{\pi^k}(s_{h+1}^k) \right| + \frac{1}{H} \left| \bar{\delta}_{h+1,k}^{\pi^k}(s_{h+1}^k) \right| + \frac{C_1}{H} \left| \underline{\delta}_{h+1,k}^{\pi^k}(s_{h+1}^k) \right| \right).
$$

Then, the proof is complete by noticing that $\mathcal{E}_{h+1,k}^{cum} = \mathcal{E}_{h,k}^{cum} \cap \mathcal{E}_{h+1,k}^{th}$. $\qquad\square$

**Lemma 17.** *With probability at least $1 - \delta$, for all $(k,h,s_h^k)$, under good event $\mathcal{G}_k$ it holds that*

$$
\mathbb{1}\left\{ \mathcal{E}_{h,k}^{cum} \right\} \left| \underline{\delta}_{h,k}^{\pi^k}(s_h^k) \right|
$$

$$
\leq \mathbb{1}\left\{ \mathcal{E}_{h,k}^{cum} \right\} \left( \left| \mathcal{P}_{h,s_h^k,a_h^k}^k + \mathcal{R}_{h,s_h^k,a_h^k}^k + \underline{w}_{\mathrm{ty}}^k(h,s_h^k,a_h^k) \right| + \mathcal{M}_{\left| \underline{\delta}_{h,k}^{\pi^k}(s_h^k) \right|} + \mathcal{M}_{\left| \underline{\delta}_{h,k}(s_h^k) \right|} + \frac{2SH^2L}{n_k(h,s_h^k,a_h^k)} \right)
$$

$$
+ \mathbb{1}\left\{ \mathcal{E}_{h+1,k}^{cum} \right\} \left( \frac{C_1}{H} \left| \bar{\bar{\bar{\delta}}}_{h+1,k}^{\pi^k}(s_{h+1}^k) \right| + \frac{H+1+C_1}{H} \left| \underline{\delta}_{h+1,k}^{\pi^k}(s_{h+1}^k) \right| + \frac{1}{H} \left| \bar{\delta}_{h+1,k}^{\pi^k}(s_{h+1}^k) \right| \right)
$$

$$
+ \mathbb{1}\left\{ \mathcal{E}_{h,k}^{cum} \cap \left( \mathcal{E}_{h+1,k}^{th} \right)^c \right\} \left( \frac{C_1}{H} \left| \bar{\bar{\bar{\delta}}}_{h+1,k}^{\pi^k}(s_{h+1}^k) \right| + \frac{H+1+C_1}{H} \left| \underline{\delta}_{h+1,k}^{\pi^k}(s_{h+1}^k) \right| + \frac{1}{H} \left| \bar{\delta}_{h+1,k}^{\pi^k}(s_{h+1}^k) \right| \right).
$$

*Proof.* The proof exactly follows the proof of Lemma 16. $\qquad\square$

**Lemma 18.** *With probability at least $1 - \delta$, for all $(k,h,s_h^k)$, under good event $\mathcal{G}_k$ it holds that*

$$
\mathbb{1}\left\{ \mathcal{E}_{h,k}^{cum} \right\} \left| \bar{\delta}_{h,k}^{\pi^k}(s_h^k) \right|
$$

$$
\leq \mathbb{1}\left\{ \mathcal{E}_{h,k}^{cum} \right\} \left( \left| \mathcal{P}_{h,s_h^k,a_h^k}^k + \mathcal{R}_{h,s_h^k,a_h^k}^k + w_{\mathrm{ty}}^k(h,s_h^k,a_h^k) \right| + \mathcal{M}_{\left| \bar{\delta}_{h,k}^{\pi^k}(s_h^k) \right|} + \mathcal{M}_{\left| \bar{\delta}_{h,k}(s_h^k) \right|} + \frac{2SH^2L}{n_k(h,s_h^k,a_h^k)} \right)
$$

$$
+ \mathbb{1}\left\{ \mathcal{E}_{h+1,k}^{cum} \right\} \left( \frac{C_1}{H} \left| \bar{\bar{\bar{\delta}}}_{h+1,k}^{\pi^k}(s_{h+1}^k) \right| + \frac{C_1}{H} \left| \underline{\delta}_{h+1,k}^{\pi^k}(s_{h+1}^k) \right| + \frac{H+1}{H} \left| \bar{\delta}_{h+1,k}^{\pi^k}(s_{h+1}^k) \right| \right)
$$

$$
+ \mathbb{1}\left\{ \mathcal{E}_{h,k}^{cum} \cap \left( \mathcal{E}_{h+1,k}^{th} \right)^c \right\} \left( \frac{C_1}{H} \left| \bar{\bar{\bar{\delta}}}_{h+1,k}^{\pi^k}(s_{h+1}^k) \right| + \frac{C_1}{H} \left| \underline{\delta}_{h+1,k}^{\pi^k}(s_{h+1}^k) \right| + \frac{H+1}{H} \left| \bar{\delta}_{h+1,k}^{\pi^k}(s_{h+1}^k) \right| \right).
$$

*Proof.* The proof exactly follows the proof of Lemma 16. $\qquad\square$

**Lemma 19.** *With probability at least $1 - \delta$, for all $(k,i,s_i^k)$, under good event $\mathcal{G}_k$ it holds that*

$$
\mathbb{1}\left\{ \mathcal{E}_{i,k}^{cum} \right\} \left( \left| \bar{\delta}_{i,k}^{\pi^k}(s_i^k) \right| + \left| \bar{\bar{\bar{\delta}}}_{i,k}^{\pi^k}(s_i^k) \right| + \left| \underline{\delta}_{i,k}^{\pi^k}(s_i^k) \right| \right)
$$

$$
\leq 3e^{3C_1} \left( \sum_{h=i}^{H} \sqrt{e_{\mathrm{ty}}^k(h,s_h^k,a_h^k)} + \sum_{h=i}^{H} \gamma_{\mathrm{ty}}^k(h,s_h^k,a_h^k) + \sum_{h=i}^{H} \frac{SH^2L}{n_k(h,s_h^k,a_h^k)} \right)
$$

$$+ e^{3C_1} \sum_{h=i+1}^{H} \mathbb{1}\left\{\left(\mathcal{E}_{i+1,k}^{th}\right)^c\right\} \left(\left|\overline{\delta}_{i+1,k}^{\pi^k}(s_{i+1}^k)\right| + \left|\overline{\overline{\delta}}_{i+1,k}^{\pi^k}(s_{i+1}^k)\right| + \left|\underline{\delta}_{i+1,k}^{\pi^k}(s_{i+1}^k)\right|\right)$$

$$+ \sum_{h=i}^{H}(1 + \frac{3C_1}{H})^{h-1}\mathbb{1}\left\{\mathcal{E}_{h,k}^{cum}\right\} \mathcal{M}_{h,k},$$

$$where \; \mathcal{M}_{h,k} = \mathcal{M}_{\left|\overline{\overline{\delta}}_{h,k}^{\pi^k}(s_h^k)\right|} + \mathcal{M}_{\left|\overline{\overline{\delta}}_{h,k}(s_h^k)\right|} + \mathcal{M}_{\left|\underline{\delta}_{h,k}^{\pi^k}(s_h^k)\right|}$$

$$+ \mathcal{M}_{\left|\underline{\delta}_{h,k}(s_h^k)\right|} + \mathcal{M}_{\left|\overline{\delta}_{h,k}^{\pi^k}(s_h^k)\right|} + \mathcal{M}_{\left|\overline{\delta}_{h,k}(s_h^k)\right|}.$$

*Proof.* By summing results in Lemma 16, Lemma 17 and Lemma 18, we have

$$\mathbb{1}\left\{\mathcal{E}_{h,k}^{cum}\right\} \left(\left|\overline{\delta}_{h,k}^{\pi^k}(s_h^k)\right| + \left|\overline{\overline{\delta}}_{h,k}^{\pi^k}(s_h^k)\right| + \left|\underline{\delta}_{h,k}^{\pi^k}(s_h^k)\right|\right)$$

$$\leq \mathbb{1}\left\{\mathcal{E}_{h+1,k}^{cum}\right\} \left(1 + \frac{3C_1}{H}\right) \left(\left|\overline{\delta}_{h+1,k}^{\pi^k}(s_{h+1}^k)\right| + \left|\overline{\overline{\delta}}_{h+1,k}^{\pi^k}(s_{h+1}^k)\right| + \left|\underline{\delta}_{h+1,k}^{\pi^k}(s_{h+1}^k)\right|\right)$$

$$+ \left|w_{\mathrm{ty}}^k(h, s_h^k, a_h^k)\right| + \left|\overline{w}_{\mathrm{ty}}^k(h, s_h^k, a_h^k)\right| + \left|\underline{w}_{\mathrm{ty}}^k(h, s_h^k, a_h^k)\right| + \frac{6SH^2L}{n_k(h, s_h^k, a_h^k)} + \mathbb{1}\left\{\mathcal{E}_{h,k}^{cum}\right\} \mathcal{M}_{h,k}$$

$$+ \mathbb{1}\left\{\mathcal{E}_{h,k}^{cum} \cap \left(\mathcal{E}_{h+1,k}^{th}\right)^c\right\} \left(1 + \frac{3C_1}{H}\right) \left(\left|\overline{\delta}_{h+1,k}^{\pi^k}(s_{h+1}^k)\right| + \left|\overline{\overline{\delta}}_{h+1,k}^{\pi^k}(s_{h+1}^k)\right| + \left|\underline{\delta}_{h+1,k}^{\pi^k}(s_{h+1}^k)\right|\right)$$

$$+ 3\left|\mathcal{P}_{h,s_h^k,a_h^k}^k + \mathcal{R}_{h,s_h^k,a_h^k}^k\right|$$

$$\overset{(i)}{\leq} \mathbb{1}\left\{\mathcal{E}_{h+1,k}^{cum}\right\} \left(1 + \frac{3C_1}{H}\right) \left(\left|\overline{\delta}_{h+1,k}^{\pi^k}(s_{h+1}^k)\right| + \left|\overline{\overline{\delta}}_{h+1,k}^{\pi^k}(s_{h+1}^k)\right| + \left|\underline{\delta}_{h+1,k}^{\pi^k}(s_{h+1}^k)\right|\right)$$

$$+ 3\sqrt{e_{\mathrm{ty}}^k(h, s_h^k, a_h^k)} + 3\gamma_{\mathrm{ty}}^k(h, s_h^k, a_h^k) + \frac{6SH^2L}{n_k(h, s_h^k, a_h^k)} + \mathbb{1}\left\{\mathcal{E}_{h,k}^{cum}\right\} \mathcal{M}_{h,k}$$

$$+ \mathbb{1}\left\{\left(\mathcal{E}_{h+1,k}^{th}\right)^c\right\} \left(1 + \frac{3C_1}{H}\right) \left(\left|\overline{\delta}_{h+1,k}^{\pi^k}(s_{h+1}^k)\right| + \left|\overline{\overline{\delta}}_{h+1,k}^{\pi^k}(s_{h+1}^k)\right| + \left|\underline{\delta}_{h+1,k}^{\pi^k}(s_{h+1}^k)\right|\right)$$

Here, the inequality (i) above holds because of two reasons. Firstly, under event $\mathcal{G}_k$, we have $|w_{\mathrm{ty}}^k(h, s_h^k, a_h^k)| \leq |\underline{w}_{\mathrm{ty}}^k(h, s_h^k, a_h^k)|$ and

$$\left|\mathcal{P}_{h,s_h^k,a_h^k}^k + \mathcal{R}_{h,s_h^k,a_h^k}^k\right| = \left|\left\langle \hat{P}_{h,s_h^k,a_h^k}^k - P_{h,s_h^k,a_h^k}, V_{h+1}^*\right\rangle + \left(\hat{R}_{h,s_h^k,a_h^k}^k - R_{h,s_h^k,a_h^k}\right)\right|$$
$$\text{(By Definition 5)}$$

$$\leq \sqrt{e_{\mathrm{ty}}^k(h, s_h^k, a_h^k)}. \qquad \text{(Under event } \mathcal{G}_k, \hat{M} \in \mathcal{M}_{\mathrm{ty}}^k)$$

Then, the proof is complete by using this recursion from $h = i$ to $h = H$ and utilizing the fact that $(1 + \frac{3C_1}{H})^H \leq e^{3C_1}$. $\qquad\square$

### E.3 Combining Estimation and Pessimism Terms

**Lemma 20.** *With probability at least $1 - \delta$, it holds that*

$$\mathrm{Regret}\,(M, K, \mathsf{SSR}_{\mathrm{ty}}) \leq \mathbb{1}\{\mathcal{G}_k\} 3C_1 e^{3C_1} \sum_{k=1}^{K} \sum_{h=i}^{H} \left(\sqrt{e_{\mathrm{ty}}^k(h, s_h^k, a_h^k)} + \gamma_{\mathrm{ty}}^k(h, s_h^k, a_h^k)\right)$$

$$+ \widetilde{O}\left(H^4 S^2 A + H\sqrt{T}\right).$$

*Proof.* Recall in equation (25), with probability at least $1 - \delta$, we have

$$\mathrm{Regret}\,(M, K, \mathsf{SSR}_{\mathrm{ty}})$$

$$\leq \sum_{k=1}^{K} \mathbb{1}\left\{\mathcal{G}_k\right\} \left( \left|\overline{\delta}_{1,k}(s_1^k)\right| + \left|\overline{\delta}_{1,k}^{\pi^k}(s_1^k)\right| \right) + \widetilde{O}(H)$$

$$\leq \sum_{k=1}^{K} \mathbb{1}\left\{\mathcal{G}_k\right\} C_1 \left( \left|\overline{\delta}_{1,k}^{\pi^k}(s_1^k)\right| + \left|\overline{\overline{\delta}}_{1,k}^{\pi^k}(s_1^k)\right| + \left|\underline{\delta}_{1,k}^{\pi^k}(s_1^k)\right| \right) + \widetilde{O}(H) \qquad \text{(By using Lemma 15)}$$

$$= \sum_{k=1}^{K} \mathbb{1}\left\{\mathcal{G}_k \cap \mathcal{E}_{1,k}^{cum}\right\} C_1 \left( \left|\overline{\delta}_{1,k}^{\pi^k}(s_1^k)\right| + \left|\overline{\overline{\delta}}_{1,k}^{\pi^k}(s_1^k)\right| + \left|\underline{\delta}_{1,k}^{\pi^k}(s_1^k)\right| \right) + \widetilde{O}(H)$$

$$\quad + \sum_{k=1}^{K} \mathbb{1}\left\{\left(\mathcal{E}_{1,k}^{th}\right)^c\right\} \left( \left|\overline{\delta}_{1,k}^{\pi^k}(s_1^k)\right| + \left|\overline{\overline{\delta}}_{1,k}^{\pi^k}(s_1^k)\right| + \left|\underline{\delta}_{1,k}^{\pi^k}(s_1^k)\right| \right)$$

$$\leq \mathbb{1}\left\{\mathcal{G}_k\right\} 3C_1 e^{3C_1} \left( \sum_{k=1}^{K}\sum_{h=i}^{H} \sqrt{e_{\text{ty}}^k(h,s_h^k,a_h^k)} + \sum_{k=1}^{K}\sum_{h=i}^{H} \gamma_{\text{ty}}^k(h,s_h^k,a_h^k) + \sum_{k=1}^{K}\sum_{h=i}^{H} \frac{SH^2 L}{n_k(h,s_h^k,a_h^k)} \right)$$

$$\quad + \sum_{k=1}^{K}\sum_{h=i}^{H}(1+\frac{3C_1}{H})^{h-1}\mathbb{1}\left\{\mathcal{G}_k \cap \mathcal{E}_{h,k}^{cum}\right\} \mathcal{M}_{h,k} + \widetilde{O}(H)$$

$$\quad + \sum_{k=1}^{K}\sum_{h=1}^{H} \mathbb{1}\left\{\left(\mathcal{E}_{h,k}^{th}\right)^c\right\} \left( \left|\overline{\delta}_{h,k}^{\pi^k}(s_1^k)\right| + \left|\overline{\overline{\delta}}_{h,k}^{\pi^k}(s_h^k)\right| + \left|\underline{\delta}_{h,k}^{\pi^k}(s_h^k)\right| \right) \qquad \text{(By using Lemma 19)}$$

$$\overset{(i)}{\leq} \mathbb{1}\left\{\mathcal{G}_k\right\} 3C_1 e^{3C_1} \sum_{k=1}^{K}\sum_{h=i}^{H} \left( \sqrt{e_{\text{ty}}^k(h,s_h^k,a_h^k)} + \gamma_{\text{ty}}^k(h,s_h^k,a_h^k) \right) + \widetilde{O}\left(H^3 S^2 A + H\sqrt{T}\right)$$

$$\quad + \widetilde{O}(H) \sum_{k=1}^{K}\sum_{h=1}^{H} \mathbb{1}\left\{\left(\mathcal{E}_{h,k}^{th}\right)^c\right\}$$

$$\leq \mathbb{1}\left\{\mathcal{G}_k\right\} 3C_1 e^{3C_1} \sum_{k=1}^{K}\sum_{h=i}^{H} \left( \sqrt{e_{\text{ty}}^k(h,s_h^k,a_h^k)} + \gamma_{\text{ty}}^k(h,s_h^k,a_h^k) \right) + \widetilde{O}\left(H^4 S^2 A + H\sqrt{T}\right).$$

$$\text{(By using Lemma 21)}$$

The inequality (i) above holds for two reasons. First, it uses Lemma 22 and 24. Second, by our clipping threshold, we know that $\left( \left|\overline{\delta}_{h,k}^{\pi^k}(s_1^k)\right| + \left|\overline{\overline{\delta}}_{h,k}^{\pi^k}(s_h^k)\right| + \left|\underline{\delta}_{h,k}^{\pi^k}(s_h^k)\right| \right) \leq \widehat{O}(H)$. $\qquad \square$

**Lemma 21** (Lemma 20 in [2]).

$$\sum_{k=1}^{K}\sum_{h=1}^{H} \mathbb{1}\left\{\left(\mathcal{E}_{h,k}^{th}\right)^c\right\} \leq \widetilde{O}\left(H^3 SA\right).$$

*Proof.* It holds that

$$\sum_{k=1}^{K}\sum_{h=1}^{H} \mathbb{1}\left\{\left(\mathcal{E}_{h,k}^{th}\right)^c\right\} = \sum_{k=1}^{K}\sum_{h=1}^{H} \mathbb{1}\left\{n_k(h,s_h^k,a_h^k) \leq \alpha_k\right\}$$

$$\leq \sum_{s\in\mathcal{S}}\sum_{a\in\mathcal{A}}\sum_{h=1}^{H} \alpha_k$$

$$\leq 200 H^3 SA \log\left(2HSAK^2\right) \log\left(40K^4\right) \qquad \text{(By our choice of } \alpha_k)$$

$$= \widetilde{O}\left(H^3 SA\right).$$

$$\square$$

# F Bounds on Individual Terms

## F.1 Bounds on Martingale Difference

**Lemma 22.** *For $i \in [H]$, the sequences starting from $0$ and with difference between two consecutive terms given by $\mathbb{1}\{\mathcal{G}_k\}\mathcal{M}_{h,k}$ for $h = i, ..., H$, $k = 1, ...K$ are martingales with respect to filtration $\left\{\overline{\mathcal{H}}_h^k\right\}_{\substack{h=i,...,H, \\ k=1,...,K}}$. Moreover, for any $\delta' > 0$, with probability at least $1 - \delta'$, for any $i \in [H]$, the following hold,*

$$\left|\sum_{k=1}^{K}\sum_{h=i}^{H}\left(1 + \frac{3C_1}{H}\right)^h \mathbb{1}\{\mathcal{G}_k \cap \mathcal{E}_{h,k}^{cum}\}\mathcal{M}_{h,k}\right| = \tilde{O}\left(H\sqrt{T}\right).$$

*Proof.* We first show the sequence starting from $0$ and with difference between two consecutive terms given by $\mathbb{1}\{\mathcal{G}_k \cap \mathcal{E}_{h,k}^{cum}\}\left(1 + \frac{3C_1}{H}\right)^h \mathcal{M}_{|\overline{\delta}_{h,k}^{\pi^k}(s_h^k)|}$ is a martingale sequence. For any $h \in \{i, ..., H\}$ and $k \in [K]$,

$$\mathbb{E}\left[\mathbb{1}\{\mathcal{G}_k \cap \mathcal{E}_{h,k}^{cum}\}\mathcal{M}_{|\overline{\delta}_{h,k}^{\pi^k}(s_h^k)|} \mid \overline{\mathcal{H}}_h^k\right]$$

$$=\mathbb{E}\left[\mathbb{1}\{\mathcal{G}_k \cap \mathcal{E}_{h,k}^{cum}\}\left(\left\langle P_{h,s_h^k,a_h^k}, \left|\overline{\delta}_{h+1,k}^{\pi^k}(s_{h+1}^k)\right|\right\rangle - \left|\overline{\delta}_{h+1,k}^{\pi^k}(s_{h+1}^k)\right|\right) \mid \overline{\mathcal{H}}_h^k\right] = 0.$$

Similarly, we have $\mathbb{1}\{\mathcal{G}_k \cap \mathcal{E}_{h,k}^{cum}\}\mathcal{M}_{|\overline{\overline{\delta}}_{h,k}^{\pi^k}(s_h^k)|}$, $\mathbb{1}\{\mathcal{G}_k \cap \mathcal{E}_{h,k}^{cum}\}\mathcal{M}_{|\underline{\delta}_{h,k}^{\pi^k}(s_h^k)|}$, $\mathbb{1}\{\mathcal{G}_k \cap \mathcal{E}_{h,k}^{cum}\}\mathcal{M}_{|\overline{\delta}_{h,k}(s_h^k)|}$, $\mathbb{1}\{\mathcal{G}_k \cap \mathcal{E}_{h,k}^{cum}\}\mathcal{M}_{|\overline{\overline{\delta}}_{h,k}(s_h^k)|}$, $\mathbb{1}\{\mathcal{G}_k \cap \mathcal{E}_{h,k}^{cum}\}\mathcal{M}_{|\underline{\delta}_{h,k}(s_h^k)|}$ are martingale difference sequences. As $\mathbb{1}\{\mathcal{G}_k \cap \mathcal{E}_{h,k}^{cum}\}\mathcal{M}_{h,k}$ is the sum of several martingale difference sequences, it is a martingale difference sequence.

Next, we bound $\left|\mathbb{1}\{\mathcal{G}_k \cap \mathcal{E}_{h,k}^{cum}\}\mathcal{M}_{|\overline{\delta}_{h,k}^{\pi^k}(s_h^k)|}\right|$. When $h = H$, $\left|\mathbb{1}\{\mathcal{G}_k \cap \mathcal{E}_{h,k}^{cum}\}\mathcal{M}_{|\overline{\delta}_{h,k}^{\pi^k}(s_h^k)|}\right| = 0$. When $\mathcal{G}_k$ holds, for $h < H$ and any state $x$,

$$\left|\overline{\delta}_{h+1,k}^{\pi^k}(x)\right| = \left|\overline{V}_{h+1}(x) - V_{h+1}^\pi(x)\right| = \left|\left\langle P_{h+2,x,\pi(x)}, \overline{V}_{h+2} - V_{h+2}^\pi\right\rangle + w_{\text{ty}}^k(h+1, x, \pi(x))\right|$$

$$\leq \left\langle P_{h+2,x,\pi(x)}, \left|\overline{V}_{h+2} - V_{h+2}^\pi\right|\right\rangle + \left|w_{\text{ty}}^k(h+1, x, \pi(x))\right|$$

By our choice of $\alpha_k$, when $\mathcal{G}_k$ holds, $\left|w_{\text{ty}}^k(h, s, a)\right| \leq \gamma_{\text{ty}}^k(h, s, a) \leq 1$ for all $k, h, s, a$ as shown in Lemma 8. Then, by expanding $\left|\overline{\delta}_{h+1,k}^{\pi^k}(x)\right| = \left|\overline{V}_{h+1}(x) - V_{h+1}^\pi(x)\right|$ recursively from $h + 1$ to $H$, we have

$$\left|\mathbb{1}\{\mathcal{G}_k \cap \mathcal{E}_{h,k}^{cum}\}\mathcal{M}_{|\overline{\delta}_{h,k}^{\pi^k}(s_h^k)|}\right| \leq 2H\gamma_{\text{ty}}^k(h, s, a) \leq 2H.$$

Similarly, we have the bound on $\mathbb{1}\{\mathcal{G}_k \cap \mathcal{E}_{h,k}^{cum}\}\mathcal{M}_{|\overline{\overline{\delta}}_{h,k}^{\pi^k}(s_h^k)|}$, $\mathbb{1}\{\mathcal{G}_k \cap \mathcal{E}_{h,k}^{cum}\}\mathcal{M}_{|\underline{\delta}_{h,k}^{\pi^k}(s_h^k)|}$, $\mathbb{1}\{\mathcal{G}_k \cap \mathcal{E}_{h,k}^{cum}\}\mathcal{M}_{|\overline{\delta}_{h,k}(s_h^k)|}$, $\mathbb{1}\{\mathcal{G}_k \cap \mathcal{E}_{h,k}^{cum}\}\mathcal{M}_{|\overline{\overline{\delta}}_{h,k}(s_h^k)|}$ and $\mathbb{1}\{\mathcal{G}_k \cap \mathcal{E}_{h,k}^{cum}\}\mathcal{M}_{|\underline{\delta}_{h,k}(s_h^k)|}$.

As a result, $\left|\mathbb{1}\{\mathcal{G}_k \cap \mathcal{E}_{h,k}^{cum}\}\left(1 + \frac{3C_1}{H}\right)^h \mathcal{M}_{h,k}\right|$ is bounded by $12e^{3C_1}H$. By Azuma-Hoeffding inequality, with probability at least $1 - \delta'$, we have

$$\left|\sum_{k=1}^{K}\sum_{h=i}^{H}\left(1 + \frac{3C_1}{H}\right)^h \mathbb{1}\{\mathcal{G}_k \cap \mathcal{E}_{h,k}^{cum}\}\mathcal{M}_{h,k}\right| \leq \tilde{O}\left(\sqrt{\sum_{k=1}^{K}\sum_{h=i}^{H}H^2}\right) = \tilde{O}\left(H\sqrt{T}\right).$$

$\square$

### F.2 Bounds on Lower-order Terms

The following two lemmas are standard results in literature and we present their proofs here for completeness.

**Lemma 23.**

$$\sum_{k=1}^{K}\sum_{h=1}^{H}\sqrt{\frac{\log\left(2HSAk^2\right)}{n_k\left(h,s_h^k,a_h^k\right)+1}} \leq \widetilde{O}\left(\sqrt{HSAT}\right).$$

*Proof.* Let $L = \log\left(2HSAK^2\right)$. Then, it can be bounded as

$$\sum_{k=1}^{K}\sum_{h=1}^{H}\sqrt{\frac{\log\left(2HSAk^2\right)}{n_k\left(h,s_h^k,a_h^k\right)+1}} \leq \sqrt{L}\sum_{k=1}^{K}\sum_{h=1}^{H}\sqrt{\frac{1}{n_k(h,s_h^k,a_h^k)+1}}$$

$$= \sqrt{L}\sum_{h,s,a}\sum_{n=1}^{n_K(h,s,a)}\sqrt{\frac{1}{n+1}}$$

$$\leq \sqrt{L}\sum_{h,s,a}\int_{0}^{n_K(h,s,a)}\sqrt{\frac{1}{x}}\mathrm{d}x$$

$$\leq 2\sqrt{L}\sum_{h,s,a}\sqrt{n_K\left(h,s,a\right)}$$

$$\leq 2\sqrt{L}\cdot\sqrt{HSA\sum_{h,s,a}n_K\left(h,s,a\right)}$$

(By Cauchy-Schwartz inequality)

$$= \widetilde{O}\left(\sqrt{HSAT}\right). \qquad \text{(Since } \sum_{h,s,a}n_K\left(h,s,a\right)=T\text{)}$$

$\square$

**Lemma 24.**

$$\sum_{k=1}^{K}\sum_{h=1}^{H}\frac{\log\left(2HSAk^2\right)}{n_k\left(h,s_h^k,a_h^k\right)+1} \leq \widetilde{O}\left(HSA\right).$$

*Proof.* Let $L = \log\left(2HSAK^2\right)$. Then, it can be bounded as

$$\sum_{k=1}^{K}\sum_{h=1}^{H}\frac{\log\left(2HSAk^2\right)}{n_k\left(h,s_h^k,a_h^k\right)+1} \leq L\sum_{k=1}^{K}\sum_{h=1}^{H}\frac{1}{n_k(h,s_h^k,a_h^k)+1}$$

$$= L\sum_{h,s,a}\sum_{n=1}^{n_K(h,s,a)}\frac{1}{n+1}$$

$$\leq L\sum_{h,s,a}\log\left(n_K(h,s,a)\right) \qquad \text{(Since } \sum_{n=1}^{N}\frac{1}{n}\leq\log(N)+1\text{)}$$

$$\leq LHSA\cdot\max_{h,s,a}\log(n_K(h,s,a))$$

$$\leq LHSA\log(T)$$

$$= \widetilde{O}\left(HSA\right).$$

$\square$

## G Bounds on Sum of Variance

When we use the Bernstein-type noise, the regret analysis needs to bound the sum of variance. This proof applies some techniques developed in [7]. However, since our optimism only holds with

constant probability instead of deterministically, the details are quite different. For simplicity, we first define

$$
\hat{\mathbb{V}}^*_{h+1,k} = \mathbb{V}\left(\tilde{P}^k_{h,s^k_h,a^k_h}, V^*_{h+1}\right), \quad \mathbb{V}^*_{h+1,k} = \mathbb{V}\left(P_{h,s^k_h,a^k_h}, V^*_{h+1}\right),
$$

$$
\hat{\overline{\mathbb{V}}}_{h+1,k} = \mathbb{V}\left(\tilde{P}^k_{h,s^k_h,a^k_h}, \overline{V}_{h+1,k}\right), \quad \overline{\mathbb{V}}_{h+1,k} = \mathbb{V}\left(P_{h,s^k_h,a^k_h}, \overline{V}_{h+1,k}\right),
$$

$$
\mathbb{V}^{\pi^k}_{h+1,k} = \mathbb{V}\left(P_{h,s^k_h,a^k_h}, V^{\pi^k}_{h+1,k}\right),
$$

$$
U_{h,k,1} = \sqrt{\frac{\hat{\mathbb{V}}^*_{h+1,k}\log\left(2HSAK^2\right)}{n_k(h,s^k_h,a^k_h)+1}}, \quad U_{h,k,2} = \sqrt{\frac{\hat{\overline{\mathbb{V}}}_{h+1,k}\log\left(2HSAK^2\right)}{n_k(h,s^k_h,a^k_h)+1}},
$$

We will first give a full proof of the bound on sum of variance and then present all the auxiliary lemmas in Section G.1.

**Lemma 25.** *Let $U_{h,k} = U_{h,k,1} + U_{h,k,2}$. For any $\delta > 0$, with probability at least $1 - \delta$, when $T \geq \Omega\left(H^5 S^2 A\right)$, it holds that*

$$
\sum_{k=1}^{K}\sum_{h=1}^{H-1} \mathbb{1}\left\{\mathcal{G}_k\right\} U_{h,k} \leq \tilde{O}\left(H\sqrt{SAT}\right).
$$

*Proof.* First, we have

$$
\sum_{k=1}^{K}\sum_{h=1}^{H-1} \mathbb{1}\left\{\mathcal{G}_k\right\} U_{h,k} \leq \sum_{k=1}^{K}\sum_{h=1}^{H-1} \mathbb{1}\left\{\mathcal{G}_k\right\} \sqrt{\frac{\log\left(2HSAK^2\right)}{n_k(h,s^k_h,a^k_h)+1}} \left(\sqrt{\hat{\mathbb{V}}^*_{h+1,k}} + \sqrt{\hat{\overline{\mathbb{V}}}_{h+1,k}}\right)
$$

$$
\leq \sum_{k=1}^{K}\sum_{h=1}^{H-1} \mathbb{1}\left\{\mathcal{G}_k\right\} \sqrt{\frac{\log\left(2HSAK^2\right)}{n_k(h,s^k_h,a^k_h)+1}} \cdot \sqrt{2}\sqrt{\hat{\mathbb{V}}^*_{h+1,k} + \hat{\overline{\mathbb{V}}}_{h+1,k}}
$$

$$
\text{(Since } \sqrt{a} + \sqrt{b} \leq \sqrt{2(a+b)} \text{ for } a, b \geq 0\text{)}
$$

$$
\leq \sqrt{2}\sqrt{\left(\sum_{k=1}^{K}\sum_{h=1}^{H-1} \frac{\log\left(2HSAK^2\right)}{n_k(h,s^k_h,a^k_h)+1}\right)\left(\sum_{k=1}^{K}\sum_{h=1}^{H-1} \mathbb{1}\left\{\mathcal{G}_k\right\}\left(\hat{\mathbb{V}}^*_{h+1,k} + \hat{\overline{\mathbb{V}}}_{h+1,k}\right)\right)}
$$

$$
\text{(By Cauchy-Schwartz inequality)}
$$

$$
\leq \sqrt{\tilde{O}\left(HSA\right)\left(\sum_{k=1}^{K}\sum_{h=1}^{H-1} \mathbb{1}\left\{\mathcal{G}_k\right\}\left(\hat{\mathbb{V}}^*_{h+1,k} + \hat{\overline{\mathbb{V}}}_{h+1,k}\right)\right)}, \tag{29}
$$

where the last inequality above applies Lemma 24.

We will then bound the two sums of variance separately. Specifically, by applying Lemma 26 and Lemma 28, we have with probability at least $1 - \delta/3$,

$$
\sum_{k=1}^{K}\sum_{h=1}^{H-1} \mathbb{1}\left\{\mathcal{G}_k\right\} \hat{\mathbb{V}}^*_{h+1,k} \tag{30}
$$

$$
= \frac{3}{2}\sum_{k=1}^{K}\sum_{h=1}^{H-1} \mathbb{1}\left\{\mathcal{G}_k\right\} \mathbb{V}^{\pi^k}_{h+1,k} + \sum_{k=1}^{K}\sum_{h=1}^{H-1} \mathbb{1}\left\{\mathcal{G}_k\right\}\left(\hat{\mathbb{V}}^*_{h+1,k} - \frac{3}{2}\mathbb{V}^{\pi^k}_{h+1,k}\right)
$$

$$
\leq \tilde{O}\left(HT + H^2\sqrt{T} + H^3 + H^3 S^2 A + H\sum_{k=1}^{K}\sum_{h=1}^{H-1} \mathbb{1}\left\{\mathcal{G}_k\right\} \delta^{\pi^k}_{h+1,k}(s^k_{h+1})\right). \tag{31}
$$

By similarly applying Lemma 26 and Lemma 29, we have with probability at least $1 - \delta/3$,

$$
\sum_{k=1}^{K}\sum_{h=1}^{H-1} \mathbb{1}\left\{\mathcal{G}_k\right\} \hat{\overline{\mathbb{V}}}_{h+1,k} \tag{32}
$$

$$
= \frac{3}{2}\sum_{k=1}^{K}\sum_{h=1}^{H-1} \mathbb{1}\left\{\mathcal{G}_k\right\} \mathbb{V}^{\pi^k}_{h+1,k} + \sum_{k=1}^{K}\sum_{h=1}^{H-1} \mathbb{1}\left\{\mathcal{G}_k\right\}\left(\hat{\overline{\mathbb{V}}}_{h+1,k} - \frac{3}{2}\mathbb{V}^{\pi^k}_{h+1,k}\right)
$$

$$\leq \widetilde{O}\left(HT + H^2\sqrt{T} + H^3 + H^3 S^2 A + H\sum_{k=1}^{K}\sum_{h=1}^{H-1} \mathbb{1}\{\mathcal{G}_k\}\left|\overline{\delta}_{h+1,k}^{\pi^k}(s_{h+1}^k)\right|\right), \qquad (33)$$

By combining equations (31) and (33), we have

$$\sum_{k=1}^{K}\sum_{h=1}^{H-1} \mathbb{1}\{\mathcal{G}_k\}\left(\hat{\mathbb{V}}_{h+1,k}^* + \hat{\overline{\mathbb{V}}}_{h+1,k}\right)$$

$$\leq \widetilde{O}\left(HT + H^2\sqrt{T} + H^3 S^2 A + H\sum_{k=1}^{K}\sum_{h=1}^{H-1} \mathbb{1}\{\mathcal{G}_k\}\left(\delta_{h+1,k}^{\pi^k}(s_{h+1}^k) + \left|\overline{\delta}_{h+1,k}^{\pi^k}(s_{h+1}^k)\right|\right)\right). \quad (34)$$

Then, by referring to definitions of $\sqrt{e_{\mathrm{Be}}^k(h, s_h^k, a_h^k)}$ and $\gamma_{\mathrm{Be}}^k(h, s_h^k, a_h^k)$, with probability at least $1 - \delta/3$, we have

$$\sum_{k=1}^{K}\sum_{h=1}^{H-1} \mathbb{1}\{\mathcal{G}_k\}\left(\delta_{h+1,k}^{\pi^k}(s_{h+1}^k) + \left|\overline{\delta}_{h+1,k}^{\pi^k}(s_{h+1}^k)\right|\right)$$

$$\leq \sum_{h=1}^{H-1}\left(\sum_{k=1}^{K} \mathbb{1}\{\mathcal{G}_k\}\left(\left|\delta_{h+1,k}^{\pi^k}(s_{h+1}^k)\right| + \left|\overline{\delta}_{h+1,k}^{\pi^k}(s_{h+1}^k)\right|\right)\right)$$

$$\leq \mathbb{1}\{\mathcal{G}_k\}3C_1 e^{3C_1} H\sum_{k=1}^{K}\sum_{h=1}^{H}\left(\sqrt{e_{\mathrm{ty}}^k(h, s_h^k, a_h^k)} + \gamma_{\mathrm{ty}}^k(h, s_h^k, a_h^k)\right) + \widetilde{O}\left(H^5 S^2 A + H^2\sqrt{T}\right)$$

(By referring to the proof of Lemma 20)

$$\leq \widetilde{O}\left(H^5 S^2 A + H^2\sqrt{T} + \sqrt{H^3 SAT} + H\sum_{k=1}^{K}\sum_{h=1}^{H-1} \mathbb{1}\{\mathcal{G}_k\} U_{h,k}\right) \qquad \text{(By Lemma 23 and 24)}$$

$$\leq \widetilde{O}\left(H^5 S^2 A + H^2\sqrt{SAT} + H\sum_{k=1}^{K}\sum_{h=1}^{H-1} \mathbb{1}\{\mathcal{G}_k\} U_{h,k}\right). \qquad (35)$$

By plugging equation (35) into equation (34), we can have

$$\sum_{k=1}^{K}\sum_{h=1}^{H-1} \mathbb{1}\{\mathcal{G}_k\}\left(\hat{\mathbb{V}}_{h+1,k}^* + \hat{\overline{\mathbb{V}}}_{h+1,k}\right)$$

$$\leq \widetilde{O}\left(HT + H^2\sqrt{T} + H^3 S^2 A + H^6 S^2 A + H^3\sqrt{SAT} + H^2\sum_{k=1}^{K}\sum_{h=1}^{H-1} \mathbb{1}\{\mathcal{G}_k\} U_{h,k}\right)$$

$$\leq \widetilde{O}\left(HT + H^3\sqrt{SAT} + H^6 S^2 A + H^2\sum_{k=1}^{K}\sum_{h=1}^{H-1} \mathbb{1}\{\mathcal{G}_k\} U_{h,k}\right)$$

$$\leq \widetilde{O}\left(HT + H^2\sum_{k=1}^{K}\sum_{h=1}^{H-1} \mathbb{1}\{\mathcal{G}_k\} U_{h,k}\right) \qquad \text{(When } T \geq \Omega\left(H^5 S^2 A\right)\text{)}$$

Now, by plugging the above result into equation (29), when $T \geq \Omega\left(H^5 S^2 A\right)$, it holds that

$$\sum_{k=1}^{K}\sum_{h=1}^{H-1} \mathbb{1}\{\mathcal{G}_k\} U_{h,k} \leq \sqrt{\widetilde{O}\left(HSA\left(HT + H^2\sum_{k=1}^{K}\sum_{h=1}^{H-1} \mathbb{1}\{\mathcal{G}_k\} U_{h,k}\right)\right)}$$

$$\leq \widetilde{O}\left(H\sqrt{SAT} + H^{1.5}\sqrt{\sum_{k=1}^{K}\sum_{h=1}^{H-1} \mathbb{1}\{\mathcal{G}_k\} U_{h,k}}\right).$$

It is easy to check that the above inequality implies $\sum_{k=1}^{K}\sum_{h=1}^{H-1} \mathbb{1}\{\mathcal{G}_k\} U_{h,k} \leq \widetilde{O}\left(H\sqrt{SAT}\right)$ and thus the proof is complete. $\qquad\square$

## G.1 Auxiliary Lemmas

The lemmas used for proving Lemma 25 are presented as the following.

**Lemma 26** (Lemma 8 in [7]). *For any $\delta > 0$, with probability at least $1 - \delta$, it holds that*

$$\sum_{k=1}^{K} \sum_{h=1}^{H-1} \mathbb{1}\left(\mathcal{G}_k\right) \mathbb{V}_{h+1,k}^{\pi^k} \leq \tilde{O}\left(HT + H^2\sqrt{T} + H^3\right).$$

**Lemma 27.** *For any $\delta > 0$, with probability at least $1 - \delta$, for any $k \in [K]$, $h \in [H]$, it holds that*

$$\hat{\mathbb{V}}_{h+1,k}^* \leq \frac{3}{2}\mathbb{V}_{h+1,k}^* + \frac{2H^2 S \log\left(2HS^2 AK/\delta\right)}{n_k\left(h, s_h^k, a_h^k\right)},$$

$$\hat{\overline{\mathbb{V}}}_{h+1,k} \leq \frac{3}{2}\overline{\mathbb{V}}_{h+1,k} + \frac{2H^2 S \log\left(2HS^2 AK/\delta\right)}{n_k\left(h, s_h^k, a_h^k\right)}.$$

*Proof.* The proof apply some techniques in [55]. Fix some $\delta > 0$ and let $L = \log\left(2HS^2 AK/\delta\right)$ for simplicity. First, by Lemma 30, for some tuple $(k, h, s, a, s')$, we have

$$\mathbb{P}\left(\tilde{P}_{h,s,a}^k(s') \geq \frac{3}{2}P_{h,s,a}(s') + \frac{2L}{n_k\left(h, s, a\right)}\right)$$

$$\leq \mathbb{P}\left(\tilde{P}_{h,s,a}^k(s') - P_{h,s,a}(s') \geq \sqrt{\frac{2P_{h,s,a}(s')L}{n_k\left(h, s, a\right)}} + \frac{L}{n_k\left(h, s, a\right)}\right)$$

$$\text{(Since } a + b \geq 2\sqrt{ab} \text{ for } a, b \geq 0\text{)}$$

$$\leq \frac{\delta}{HS^2 AK}.$$

Then, a union bound says that its complement holds for any $(k, h, s, a, s')$ with probability at least $1 - \delta$. Thus, we have

$$\hat{\mathbb{V}}_{h+1,k}^* = \sum_{s' \in \mathcal{S}} \tilde{P}_{h,s_h^k,a_h^k}^k(s') \left(V_{h+1}^*(s') - \left\langle \tilde{P}_{h,s_h^k,a_h^k}^k, V_{h+1}^* \right\rangle\right)^2$$

$$\leq \sum_{s' \in \mathcal{S}} \tilde{P}_{h,s_h^k,a_h^k}^k(s') \left(V_{h+1}^*(s') - \left\langle P_{h,s_h^k,a_h^k}, V_{h+1}^* \right\rangle\right)^2$$

$$\text{(Since } \mathbb{E}\left[X\right] \text{ is the minimizer of } \min_x \mathbb{E}\left[(X - x)^2\right]\text{)}$$

$$\leq \sum_{s' \in \mathcal{S}} \left(\frac{3}{2}P_{h,s,a}(s') + \frac{2L}{n_k(h, s, a)}\right) \left(V_{h+1}^*(s') - \left\langle P_{h,s_h^k,a_h^k}, V_{h+1}^* \right\rangle\right)^2$$

$$\leq \frac{3}{2}\mathbb{V}_{h+1,k}^* + \frac{2H^2 S \log\left(2HS^2 AK/\delta\right)}{n_k(h, s_h^k, a_h^k)}.$$

For $\hat{\overline{\mathbb{V}}}_{h+1,k}$, we just need to follow a similar argument and thus the proof is complete. $\square$

**Lemma 28.** *For any $\delta > 0$, with probability at least $1 - \delta$, it holds that*

$$\sum_{k=1}^{K} \sum_{h=1}^{H-1} \mathbb{1}\left\{\mathcal{G}_k\right\} \left(\hat{\mathbb{V}}_{h+1,k}^* - \frac{3}{2}\mathbb{V}_{h+1,k}^{\pi^k}\right) \leq \tilde{O}\left(H\sum_{k=1}^{K} \sum_{h=1}^{H-1} \mathbb{1}\left\{\mathcal{G}_k\right\} \delta_{h+1,k}^{\pi^k}(s_{h+1}^k) + H^2\sqrt{T} + H^3 S^2 A\right).$$

*Proof.* We begin by applying Lemma 27. Thus, with probability at least $1 - \frac{\delta}{2}$, we have

$$\sum_{k=1}^{K} \sum_{h=1}^{H-1} \mathbb{1}\left\{\mathcal{G}_k\right\} \left(\hat{\mathbb{V}}_{h+1,k}^* - \frac{3}{2}\mathbb{V}_{h+1,k}^{\pi^k}\right)$$

$$\leq \sum_{k=1}^{K} \sum_{h=1}^{H-1} \mathbb{1}\left\{\mathcal{G}_k\right\} \left(\frac{3}{2}\mathbb{V}_{h+1,k}^* - \frac{3}{2}\mathbb{V}_{h+1,k}^{\pi^k}\right) + \frac{4H^2 S \log\left(4HS^2 AK/\delta\right)}{3n_k(h, s_h^k, a_h^k)} \qquad \text{(By Lemma 27)}$$

$$\leq \sum_{k=1}^{K} \sum_{h=1}^{H-1} \mathbb{1}\{\mathcal{G}_k\} \left(\frac{3}{2}\mathbb{V}_{h+1,k}^* - \frac{3}{2}\mathbb{V}_{h+1,k}^{\pi^k}\right) + \widetilde{O}\left(H^3 S^2 A\right) \qquad \text{(By Lemma 24)}$$

$$\leq \sum_{k=1}^{K} \sum_{h=1}^{H-1} \mathbb{1}\{\mathcal{G}_k\} \frac{3}{2}\mathbb{E}_{s'\sim P_{h,s_h^k,a_h^k}} \left[\left(V_{h+1}^*(s')\right)^2 - \left(V_{h+1,k}^{\pi^k}(s')\right)^2\right] + \widetilde{O}\left(H^3 S^2 A\right)$$
$$\text{(Since } V_{h+1,k}^{\pi^k} \leq V_{h+1}^*)$$

$$\leq 3H \sum_{k=1}^{K} \sum_{h=1}^{H-1} \mathbb{1}\{\mathcal{G}_k\} \mathbb{E}_{s'\sim P_{h,s_h^k,a_h^k}} \left[\delta_{h+1,k}^{\pi^k}(s')\right] + \widetilde{O}\left(H^3 S^2 A\right)$$
$$\text{(Since } V_{h+1,k}^{\pi^k} \leq V_{h+1}^* \leq H \text{ and } a^2 - b^2 = (a+b)(a-b))$$

$$\leq \widetilde{O}\left(H \sum_{k=1}^{K} \sum_{h=1}^{H-1} \mathbb{1}\{\mathcal{G}_k\} \delta_{h+1,k}^{\pi^k}(s_{h+1}^k) + H^2\sqrt{T} + H^3 S^2 A\right). \qquad \text{(By Lemma 22)}$$

The last line above holds because by Lemma 22, with probability at least $1 - \frac{\delta}{2}$, we have

$$\left|\sum_{k=1}^{K} \sum_{h=1}^{H-1} \mathbb{1}\{\mathcal{G}_k\} \left(\mathbb{E}_{s'\sim P_{h,s_h^k,a_h^k}} \left[\delta_{h+1,k}^{\pi^k}(s')\right] - \delta_{h+1,k}^{\pi^k}(s_{h+1}^k)\right)\right| \leq \widetilde{O}\left(H\sqrt{T}\right).$$

$\square$

**Lemma 29.** *For any $\delta > 0$, w ith probability at least $1 - \delta$, it hold that*

$$\sum_{k=1}^{K} \sum_{h=1}^{H-1} \mathbb{1}\{\mathcal{G}_k\} \left(\hat{\bar{\mathbb{V}}}_{h+1,k} - \frac{3}{2}\mathbb{V}_{h+1,k}^{\pi^k}\right) \leq \widetilde{O}\left(H \sum_{k=1}^{K} \sum_{h=1}^{H-1} \mathbb{1}\{\mathcal{G}_k\} \left|\bar{\delta}_{h+1,k}^{\pi^k}(s_{h+1}^k)\right| + H^2\sqrt{T} + H^3 S^2 A\right).$$

*Proof.* Similarly, we begin by applying Lemma 27 and with probability at least $1 - \frac{\delta}{3}$, we have

$$\sum_{k=1}^{K} \sum_{h=1}^{H-1} \mathbb{1}\{\mathcal{G}_k\} \left(\hat{\bar{\mathbb{V}}}_{h+1,k} - \frac{3}{2}\mathbb{V}_{h+1,k}^{\pi^k}\right)$$

$$\leq \sum_{k=1}^{K} \sum_{h=1}^{H-1} \mathbb{1}\{\mathcal{G}_k\} \left(\frac{3}{2}\overline{\mathbb{V}}_{h+1,k} - \frac{3}{2}\mathbb{V}_{h+1,k}^{\pi^k}\right) + \frac{4H^2 S \log\left(6HS^2 AK/\delta\right)}{3n_k(h, s_h^k, a_h^k)}$$

$$\leq \sum_{k=1}^{K} \sum_{h=1}^{H-1} \mathbb{1}\{\mathcal{G}_k\} \left(\frac{3}{2}\overline{\mathbb{V}}_{h+1,k} - \frac{3}{2}\mathbb{V}_{h+1,k}^{\pi^k}\right) + \widetilde{O}\left(H^3 S^2 A\right) \qquad \text{(By Lemma 24)}$$

$$= \frac{3}{2}\underbrace{\sum_{k=1}^{K} \sum_{h=1}^{H-1} \mathbb{1}\{\mathcal{G}_k\} \left(\left\langle P_{h,s_h^k,a_h^k}, \left(\overline{V}_{h+1,k}\right)^2\right\rangle - \left\langle P_{h,s_h^k,a_h^k}, \left(V_{h+1,k}^{\pi^k}\right)^2\right\rangle\right)}_{(a)}$$

$$+ \frac{3}{2}\underbrace{\sum_{k=1}^{K} \sum_{h=1}^{H-1} \mathbb{1}\{\mathcal{G}_k\} \left(\left\langle P_{h,s_h^k,a_h^k}, V_{h+1,k}^{\pi^k}\right\rangle^2 - \left\langle P_{h,s_h^k,a_h^k}, \overline{V}_{h+1,k}\right\rangle^2\right)}_{(b)} + \widetilde{O}\left(H^3 S^2 A\right).$$
$$\text{(By definition of variance)}$$

We will bound (a) and (b) separately. For term (a), with probability at least $1 - \frac{\delta}{3}$, we have

$$(a) = \sum_{k=1}^{K} \sum_{h=1}^{H-1} \mathbb{1}\{\mathcal{G}_k\} \left\langle P_{h,s_h^k,a_h^k}, \left(\overline{V}_{h+1,k}\right)^2 - \left(V_{h+1,k}^{\pi^k}\right)^2\right\rangle$$

$$\leq \sum_{k=1}^{K} \sum_{h=1}^{H-1} \mathbb{1}\{\mathcal{G}_k\} \left\langle P_{h,s_h^k,a_h^k}, \left|\overline{V}_{h+1,k} - V_{h+1,k}^{\pi^k}\right| \left|\overline{V}_{h+1,k} + V_{h+1,k}^{\pi^k}\right|\right\rangle$$
$$\text{(Since } a^2 - b^2 = (a+b)(a-b))$$

$$\leq 3H \sum_{k=1}^{K} \sum_{h=1}^{H-1} \mathbb{1}\{\mathcal{G}_k\} \left\langle P_{h,s_h^k,a_h^k}, \left| \overline{V}_{h+1,k} - V_{h+1,k}^{\pi^k} \right| \right\rangle \quad \text{(Since } \left\| \overline{V}_{h+1,k} \right\|_\infty \leq 2H \text{ under } \mathcal{G}_k)$$

$$\leq 3H \sum_{k=1}^{K} \sum_{h=1}^{H-1} \mathbb{1}\{\mathcal{G}_k\} \left| \overline{\delta}_{h+1,k}^{\pi^k}(s_{h+1}^k) \right| + \widetilde{O}\left( H^2 \sqrt{T} \right). \quad \text{(By Lemma 22)}$$

For term (b), with probability at least $1 - \frac{\delta}{3}$, we have

$$(b) = \sum_{k=1}^{K} \sum_{h=1}^{H-1} \mathbb{1}\{\mathcal{G}_k\} \left\langle P_{h,s_h^k,a_h^k}, V_{h+1,k}^{\pi^k} + \overline{V}_{h+1,k} \right\rangle \left\langle P_{h,s_h^k,a_h^k}, V_{h+1,k}^{\pi^k} - \overline{V}_{h+1,k} \right\rangle$$

$$\leq 3H \sum_{k=1}^{K} \sum_{h=1}^{H-1} \mathbb{1}\{\mathcal{G}_k\} \left\langle P_{h,s_h^k,a_h^k}, \left| V_{h+1,k}^{\pi^k} - \overline{V}_{h+1,k} \right| \right\rangle$$

$$\leq 3H \sum_{k=1}^{K} \sum_{h=1}^{H-1} \mathbb{1}\{\mathcal{G}_k\} \left| \overline{\delta}_{h+1,k}^{\pi^k}(s_{h+1}^k) \right| + \widetilde{O}\left( H^2 \sqrt{T} \right). \quad \text{(By Lemma 22)}$$

Therefore, in summary, we have with probability at least $1 - \delta/2$,

$$\sum_{k=1}^{K} \sum_{h=1}^{H-1} \mathbb{1}\{\mathcal{G}_k\} \left( \widehat{\overline{\mathbb{V}}}_{h+1,k} - \frac{3}{2} \mathbb{V}_{h+1,k}^{\pi^k} \right) \leq \widetilde{O}\left( H \sum_{k=1}^{K} \sum_{h=1}^{H-1} \mathbb{1}\{\mathcal{G}_k\} \left| \overline{\delta}_{h+1,k}^{\pi^k}(s_{h+1}^k) \right| + H^2 \sqrt{T} + H^3 S^2 A \right).$$

$\square$

# H  Proof of the Main Theorems

In this section, we state and prove our two main theorems.

**Theorem 1.** *If the Hoeffding-type noise is used, then for any MDP $M = (H, \mathcal{S}, \mathcal{A}, P, R, s_1)$, for any $\delta > 0$, with probability at least $1 - \delta$, Algorithm 1 satisfies*

$$\mathrm{Reg}(M, K, \mathsf{SSR}_{\mathrm{Ho}}) \leq \widetilde{O}\left( H^{1.5} \sqrt{SAT} + H^4 S^2 A \right).$$

*In particular, when $T \geq \widetilde{\Omega}\left( H^5 S^3 A \right)$, it holds that $\mathrm{Reg}(M, K, \mathsf{SSR}_{\mathrm{Ho}}) \leq \widetilde{O}\left( H^{1.5} \sqrt{SAT} \right)$.*

*Proof.* By using the result of Lemma 20, under Hoeffding-type noise, with probability at least $1 - \delta$, we have

$$\mathrm{Reg}\left( M, K, \mathsf{SSR}_{\mathrm{Ho}} \right)$$

$$\leq \mathbb{1}\{\mathcal{G}_k\} 3 C_1 e^{3C_1} \sum_{k=1}^{K} \sum_{h=i}^{H} \left( \sqrt{e_{\mathrm{Ho}}^k(h, s_h^k, a_h^k)} + \gamma_{\mathrm{Ho}}^k(h, s_h^k, a_h^k) \right) + \widetilde{O}\left( H^4 S^2 A + H \sqrt{T} \right)$$

$$\leq 6 C_1 e^{3C_1} \sum_{k=1}^{K} \sum_{h=1}^{H-1} \left( H \sqrt{\frac{\log(2HSAk^2)}{n_k(h,s,a)+1}} + \frac{H}{n_k(h,s,a)+1} \right) + \widetilde{O}\left( H^4 S^2 A + H \sqrt{T} \right)$$

$$= \widetilde{O}(H^{1.5} \sqrt{SAT} + H^4 S^2 A).$$

Here, the second inequality is from the definitions of $\sqrt{e_{\mathrm{Ho}}^k(h, s_h^k, a_h^k)}$ and $\gamma_{\mathrm{Ho}}^k(h, s_h^k, a_h^k)$, and the last step is from Lemma 23 and 24.

$\square$

**Theorem 2.** *For Bernstein-type noise and $T \geq \widetilde{\Omega}\left( H^5 S^2 A \right)$, then for any MDP $M = (H, \mathcal{S}, \mathcal{A}, P, R, s_1)$, for any $\delta > 0$, with probability at least $1 - \delta$, Algorithm 1 satisfies*

$$\mathrm{Reg}(M, K, \mathsf{SSR}_{\mathrm{Be}}) \leq \widetilde{O}\left( H \sqrt{SAT} + H^4 S^2 A \right).$$

*In particular, if we further have $T \geq \widetilde{\Omega}\left( H^6 S^3 A \right)$, it then holds that $\mathrm{Reg}(M, K, \mathsf{SSR}_{\mathrm{Be}}) \leq \widetilde{O}\left( H \sqrt{SAT} \right)$.*

*Proof.* Similar to the proof of Theorem 1, under Bernstein-type noise, it holds with probability at least $1 - \frac{\delta}{2}$ that

$$\text{Reg}\left(M, K, \mathsf{SSR}_{\text{Be}}\right)$$

$$\leq \mathbb{1}\left\{\mathcal{G}_k\right\} 3C_1 e^{3C_1} \sum_{k=1}^{K} \sum_{h=i}^{H} \left(\sqrt{e_{\text{Be}}^k(h, s_h^k, a_h^k)} + \gamma_{\text{Be}}^k(h, s_h^k, a_h^k)\right) + \widetilde{O}\left(H^4 S^2 A + H\sqrt{T}\right)$$

$$\leq \widetilde{O}\left(\sum_{k=1}^{K} \sum_{h=1}^{H-1} \left(\mathbb{1}\left\{\mathcal{G}_k\right\} U_{h,k} + \sqrt{\frac{\log(2HSAk^2)}{n_k(h,s,a)+1}} + \frac{H}{n_k(h,s,a)+1}\right)\right) + \widetilde{O}(H^4 S^2 A + H\sqrt{T})$$

$$= \widetilde{O}(H\sqrt{SAT} + H^4 S^2 A),$$

where the last step is from Lemma 25.

$\square$

# I   Technical Lemmas

**Lemma 30** (Bennet's Inequality)**.** *Let* $Z_1, \ldots, Z_n$ *be i.i.d. random variables bounded in* $[0, 1]$*. Then, for any* $\delta > 0$*, we have*

$$\mathbb{P}\left(\left|\frac{1}{n}\sum_{i=1}^{n} Z_i - \mathbb{E}[Z]\right| \geq \sqrt{\frac{2\text{Var}(Z)\log(2/\delta)}{n}} + \frac{\log(2/\delta)}{n}\right) \leq \delta.$$

**Lemma 31** (from [32])**.** *Let* $Z_1, \ldots, Z_n$ *with* $n \geq 2$ *be i.i.d. random variables bounded in* $[0, H]$*. Define* $\bar{Z} = \frac{1}{n}\sum_{i=1}^{n} Z_i$ *and* $\hat{V}_n = \frac{1}{n}\sum_{i=1}^{n}\left(Z_i - \bar{Z}\right)^2$*. Then, for any* $\delta > 0$*, we have*

$$\mathbb{P}\left(\left|\mathbb{E}[\bar{Z}] - \sum_{i=1}^{n} Z_i\right| \geq \sqrt{\frac{2\hat{V}_n\log(2/\delta)}{n-1}} + \frac{7\log(2/\delta)}{3(n-1)}\right) \leq \delta.$$

**Lemma 32.** *Let* $X$ *be arbitrary random variable bounded in* $[a, b]$ *for some* $a, b \in \mathbb{R}$*. Then, we have* $\text{Var}(X) \leq \frac{(b-a)^2}{4}$*.*

**Lemma 33.** *For any* $\delta > 0$*, with probability at least* $1 - \delta$*, it holds for all* $k, h, s, a, s'$ *that*

$$\left|\hat{P}_{h,s,a}^k(s') - P_{h,s,a}(s')\right| \leq \sqrt{\frac{4P_{h,s,a}(s')(1 - P_{h,s,a}(s'))\log(2HS^2AK/\delta)}{n_k(h,s,a)+1}} + \frac{3\log(2HS^2AK/\delta)}{n_k(h,s,a)+1}.$$

*Proof.* Let $\delta' = \frac{\delta}{HS^2AK}$ and fix $(k, h, s, a, s')$ such that $n_k(h,s,a) \geq 1$. Then, we have

$$\mathbb{P}\left(\left|\hat{P}_{h,s,a}^k(s') - P_{h,s,a}(s')\right| \geq \sqrt{\frac{4P_{h,s,a}(s')(1 - P_{h,s,a}(s'))\log(2/\delta')}{n_k(h,s,a)+1}} + \frac{3\log(2/\delta')}{n_k(h,s,a)+1}\right)$$

$$\leq \mathbb{P}\left(\left|\tilde{P}_{h,s,a}^k(s') - P_{h,s,a}(s')\right| \geq \sqrt{\frac{4P_{h,s,a}(s')(1 - P_{h,s,a}(s'))\log(2/\delta')}{n_k(h,s,a)+1}} + \frac{3\log(2/\delta') - 1}{n_k(h,s,a)+1}\right)$$

$$\leq \mathbb{P}\left(\left|\tilde{P}_{h,s,a}^k(s') - P_{h,s,a}(s')\right| \geq \sqrt{\frac{4P_{h,s,a}(s')(1 - P_{h,s,a}(s'))\log(2/\delta')}{n_k(h,s,a)+1}} + \frac{2\log(2/\delta')}{n_k(h,s,a)+1}\right)$$

$$\leq \mathbb{P}\left(\left|\tilde{P}_{h,s,a}^k(s') - P_{h,s,a}(s')\right| \geq \sqrt{\frac{2P_{h,s,a}(s')(1 - P_{h,s,a}(s'))\log(2/\delta')}{n_k(h,s,a)}} + \frac{\log(2/\delta')}{n_k(h,s,a)}\right)$$

$$\text{(Since } n + 1 \leq 2n \text{ for } n \geq 1)$$

$$\leq \delta' = \frac{\delta}{HS^2AK}. \qquad\qquad \text{(By Lemma 30, the Bennet's inequality)}$$

Then, the proof is complete by taking a union bound over all possible $(k, h, s, a, s')$. $\square$

## J  Numeric Simulations

In this section, we empirically compare RLSVI [41], UCBVI [7] and our algorithm SSR on the famous deep sea environment, which is a tabular environment frequently used to test an algorithm's ability to do efficient exploration [35, 37, 47].

Deep sea, as shown in Figure 1, is a grid-like deterministic environment with $N \times N$ cell states, action space $\{0, 1\}$ and action mask $M_{ij} \sim \text{Bernoulli}(0.5)$, $(i, j) \in \mathcal{S}$, whose values are sampled when initializing the environment. At each cell $(i, j)$. Action $M_{ij}$ represents going "right", which leads the agent to the lower right cell, and $1 - M_{ij}$ represents going "left", which leads the agent to the lower left cell. An episode of this environment will end after $N$ steps. When going "left" or going "right" at the off-diagonal, the agent will receive 0 reward; when going "right" along the diagonal before reaching the lower right corner, the agent will receive negative reward $-\frac{0.01}{N}$. Finally, when reaching the lower right corner, depending on the environment initialization, the agent will either receive reward $+1$ or $-1$. In our experiment, we set this to $+1$, which results in an obvious optimal policy "always going right" with total reward 0.99 per episode.

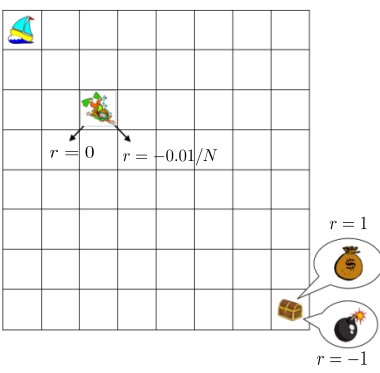

The experiment results are shown in Figure 2.[3] From the plots, we can see that in both settings, SSR performs significantly better than RLSVI as predicted by our theory. Specifically, because of the instability incurred by the independent random seeds and large perturbation magnitude, RLSVI almost never reaches the lower right corner in both settings and thus incurs linear regret. On the other hand, SSR obtains a much lower sub-linear regret because it can explore consistently with the single random seed.

Meanwhile, in both settings, SSR performs comparably with the UCBVI, which is expected since both algorithms achieve the minimax lower bound and our analysis does not indicate that one is better than the other.

Figure 1: An example deep sea environment with $N = 8$ [37].

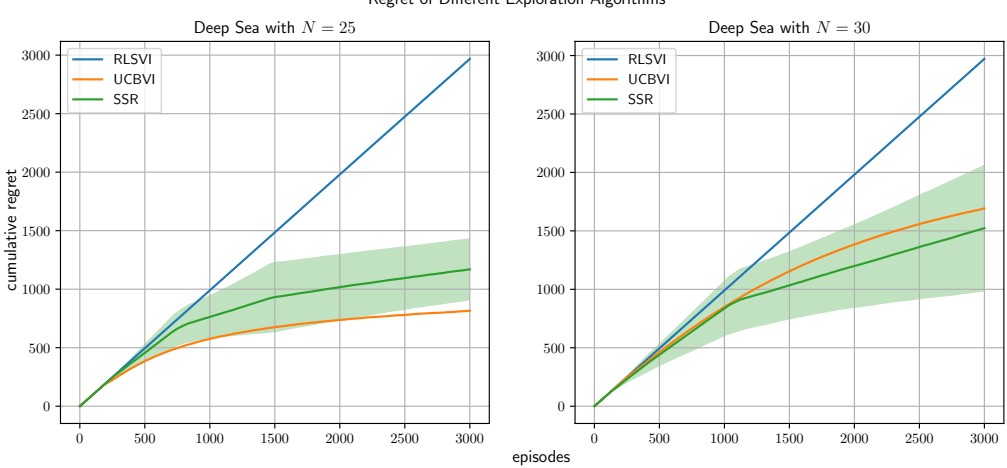

Figure 2: Empirical evaluation of RLSVI, UCBVI and SSR in deep sea environments with $N = 25$ and $N = 30$. The results are averaged over 10 repeated trials and the shaded area represents the standard deviation. For simplicity, we use Hoeffding-type bonus for both UCBVI and SSR.

---

[3]Bonuses for all three algorithms are scaled down from the theoretical values by a factor of $7 \times 10^4$ since without scaling, none of them can learn anything even in the deep sea with $N = 5$.

Finally, we also do an ablation study to show that the better performance of SSR over the RLSVI indeed comes from the single seed randomization instead of smaller noise magnitude. In particular, we run both algorithms in a deep sea environment with $N = 25$ and apply the same noise magnitude, whose results are shown in Figure 3. We can see that although using the same noise magnitude, SSR still significantly outperforms RLSVI.

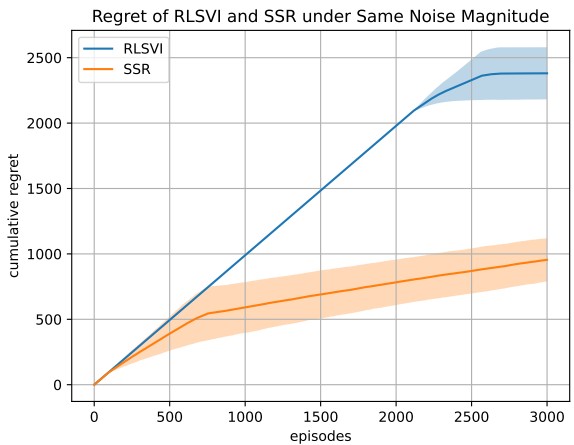

Figure 3: Empirical evaluation of RLSVI and SSR in deep sea environments with $N = 25$, where both algorithms use the same noise magnitude.