# OpenReview forum: "Near-Optimal Randomized Exploration for Tabular Markov Decision Processes"
_NeurIPS.cc/2022/Conference — NeurIPS 2022 Accept_

### Official Review · Reviewer_WUa3 · 2022-07-11

**Rating:** 6
**Confidence:** 4
**Soundness:** 4 excellent
**Presentation:** 3 good
**Contribution:** 3 good

**Summary:**

Authors propose novel algorithm for reinforcement learning with randomized exploration. Algorithm achieves minimax optimal regret bound up to poly-logarithmic factors, and it is the first such result for RL algorithms based not on the principle of Optimism in the face of Uncertainty. The most crucial algorithmic features that allow authors to achieve their result are
1) use a single Gaussian noise up to scaling for all transitions;
2) specific Bernstein-shaped magnitude of noise;
3) novel clipping procedure that was used at the first time for algorithms with randomized exploration.

**Questions:**

- The main suggestion was listed in the section on Strength and Weaknesses: it would be very nice to have an additional empirical comparison between SSR algorithm and other baselines such as UCBVI, classical RLSVI, and PSRL.
- Additionally there are several small issues with the text in the appendix:
    - In the formula after line 577 $\alpha_k$ was used without definition except the table of notations;
    - In Lemma 10 it could be great to make $z$ a parameter of function $f$;
    - In the beginning of the section of regret decomposition there was a lot of added subscript $k$ into value function that corresponds to $V^\pi$.

**Limitations:**

Authors addressed all the limitations of their paper.

**Strengths And Weaknesses:**

Strengths:

1) The first algorithm based on randomized exploration that achieves minimax optimal regret; It vastly extends the theoretical perspectives of randomized exploration.

2) Novel proof technique that allows to control pessimism error for randomized algorithms;

Weaknesses:

1) Lack of empirical study. Authors claimed that use of single seed can significantly increase the stability of the algorithm, and it would be very interesting to observe this effect practically on tabular environment with non-trivial exploration problem (N-rooms for example).

---

> ### Author Response · Authors · 2022-08-01
> **Response**
>
> Thank you very much for your careful reading and recognition of our novelty! Please find our response to your questions and concerns below:
> - **Empirical study:** Please refer to the experiments section at the top for all reviewers.
> - **Appendix issues:** Thanks a lot for the careful reading and constructive suggestions for our appendix. We will fix these issues in our final version.

---

> > ### Comment · Reviewer_WUa3 · 2022-08-08
> > **Thanks for your response**
> >
> > Thanks for your reply! Unfortunately, in the experimental section I did not found the proof of your claim about increasing of stability of the algorithm by single seed randomization. It would be interesting to observe in the final version an ablation study of your algorithm. Now it is not clear what make algorithm better: single seed randomization or just smaller magnitude of the noise.
> > However, I still think this paper is a very interesting contribution to the community and would like to keep my score.

---

> > > ### Author Response · Authors · 2022-08-09
> > > **Further response**
> > >
> > > Thank you very much for your constructive suggestion! We will add an ablation study of our algorithm in the final version.

---

> ### Author Response · Authors · 2022-08-07
> **Any other questions to be addressed？**
>
> Thanks again for your review. We hope our answers could increase your confidence. As the discussion period is close to the end and we have not yet heard back from you, we would be glad to see if our rebuttal response has addressed your concerns questions/concerns.
> We are more than happy to discuss further if you have any further concerns and issues, please kindly let us know your feedback. Thank you for your time and help!

---

### Official Review · Reviewer_GzaW · 2022-07-12

**Rating:** 8
**Confidence:** 4
**Soundness:** 4 excellent
**Presentation:** 4 excellent
**Contribution:** 4 excellent

**Summary:**

The paper considers the problem of regret minimization in RL using randomized algorithms. For this, the authors provide a single-seed-based algorithm that is able to obtain $\tilde{O}(H\sqrt{SAT})$ regret bound using Bernstein concentration-based variance scaling.

**Questions:**

How does the SSR algorithm compare against the PSRL algorithm by [5,6]?

**Limitations:**

Yes.

**Strengths And Weaknesses:**

Strength: The result of the paper and the simplicity of the algorithm using a single seed in an episode is the biggest result. Further, providing an analysis based on absolute value analysis of the difference between the optimal policy of the true problem setup and of the optimal policy of the sampled value function is interesting and can be used in many future works working on sampled MDPs.

Weakness:

1. The additive term of $O(H^4S^2A)$ is quite large.
2. The paper does not addresses the works in the area of Thompson Sampling and worst-case regret bounds for posterior sampling based algorithms.

---

> ### Author Response · Authors · 2022-08-01
> **Response**
>
> Thank you very much for your careful reading and recognition of our novelty! Please find our response to your questions and concerns below:
> - **Lower order term:** We agree that our lower order term $\widetilde{O}(H^4S^2A)$ is sub-optimal. However, it is a big open problem in reinforcement learning theory to get rid of the lower order terms and can serve as a promising future direction. In particular, our lower order term is the same as UCB-VI and more discussion about this topic can be found in [31].
> - **Thompson sampling algorithms and PSRL[5,6]:** We did not fully address the works in the area of Thompson sampling because we mainly focus on the standard frequentist regret bound and it would be hard to compare with an algorithm (such as PSRL in [5,6]) which studied different settings and objectives. On the other hand, we indeed compared with the algorithm of Thompson sampling under comparable settings and objectives such as [41] (in Section 4.1).

---

### Official Review · Reviewer_bZZ8 · 2022-07-16

**Rating:** 4
**Confidence:** 4
**Soundness:** 2 fair
**Presentation:** 2 fair
**Contribution:** 3 good

**Summary:**

The paper studies a finite state-action MDP setting with time inhomogeneous transitions and costs. The authors then propose a single seed randomization algorithm that . Regret analysis then purports to achieve $\tilde{O}(H\sqrt{SAT})$ bound on regret in the finite MDP setting using randomized algorithms (as opposed to variants of the UCB family for MDPs).


**Questions:**

1. Steps are missing in Algorithm 1.
    - Trajectory generation must be mentioned somewhere between Steps 5 and 6.
    - Please point to equations 5 and 6 to show how $\sigma^k_{ty}$ is generated. I was searching around for an expression for $\sigma^k_{ty}$ for quite some time.

2. In line 225, you claim that "the main advantage of randomized exploration lies in the algorithm component." Yet, the results show that it is the other point, i,e, the Bernstein noise that really does the $H^{1.5}$ to $H$ reduction. In light of this, how would you justify your claim in Line 225?

3. Can you say anything about expected regret?
     -  I cannot find any numerical results in the paper. In the absence of simulations, how does one compare SSR empirically with other randomized (and UCB-based) algorithms? Regret analysis only provides upper bounds which could be rather loose.
    - Moreover, simulations could also shed light on the matter of **expected regret**. The present analysis says nothing about what happens on the complement, i.e., the subset of sample paths with measure $\approx \delta.$

**Limitations:**

The work is theoretical and I cannot find any potential negative societal impact.

**Strengths And Weaknesses:**

Strengths:
1. The paper addresses an important lacuna in finite MDP Reinforcement Learning. UCB algorithms have been known to show $\tilde{O}(H\sqrt{SAT})$ regret performance for quite some time now and an extension to Thompson Sampling-type algorithms is quite necessary.

Weaknesses:
1. I'm not convinced the setting of the problem is entirely natural. Essentially, the model is one where the agent interacts with the environment over H trajectories, and the exact same sequence of transition matrices and reward functions materializes every time.
- Using a model just because it has been previously used does _not_ constitute sufficient justification.

2. The algorithm and its analysis are lacking in novelty, in my opinion.
- The concept of clipping and are introduced in [2, Algorithm 1] (see line 15)
- With regards to $\sigma^k_{ty}$ in Step 6: Moving from Hoeffding to Bernstein type randomization is rather dated and can be found in  UCB-type algorithms previously proposed in the literature.
- It appears to me that using a single seed for the trajectory is the only truly novel idea.

---

> ### Author Response · Authors · 2022-08-01
> **Response**
>
> Thank you very much for your careful reading and constructive suggestions! Please find our response to your questions and concerns below:
> - **Tabular MDP settings:** The tabular MDP setting studied in this paper is a **benchmark setting** in the RL theory community to evaluate the theoretical properties of new algorithms and techniques. All the relevant papers cited [2,7,8,9,12,14,15,22,23,31,33,41,42,48,50,51,53,54] use the same setting.
> If the transition matrix and reward are changing, then the setting becomes **non-stationary MDP** [Wei and Luo, 2021], and how to adapt UCB/TS to non-stationary MDP remains largely unsolved. However, non-stationary MDP is beyond the scope of this paper.
> - **Novelty:** We disagree we lack novelty. We want to emphasize that much more technical novelty lies in our analysis, in addition to the single seed in algorithm design.
>     - In particular, clipping used in existing literature [2,7] directly truncates the value estimate by the upper bound $H-h+1$, while we are the first one to use the new two-side clipping (with threshold $2(H-h+1)$) and this is crucial to our analysis.
>     - In the analysis, we use both optimism and pessimism to derive a novel recursive bound on the absolute value of the policy estimation error. Meanwhile, it is also the first time that pessimism is used for worst-case regret analysis in online reinforcement learning.
> - **Algorithm writing:** Thanks for the suggestion! We have added a hyperlink to equation (5) and (6) for $\sigma^k_{\mathrm{ty}}$. As for trajectory generation, we believe it has been described in line 9 and 10 of Algorithm 1. Please let us know if you were referring to something else?
> - **Advantage of randomized exploration:** We are sorry for the confusion. However, in the paragraph of line 225, we did not intend to use our regret bounds to justify that the algorithmic component is the main advantage of randomized exploration. Instead, it serves as our motivation to study randomized exploration since the *advantage* here refers to the ease of implementation instead of any superiority in regret bound.
> - **Expected regret and experiments:** Please refer to the experiments section at the top for all reviewers. As for the expected regret, we believe that high-probability regret is a strictly stronger measure than the expected regret since by simply taking $\delta=\frac{1}{T}$, the expected regret will have the same leading order as the high-probability regret.
>
> ```
> Wei, Chen-Yu, and Haipeng Luo. "Non-stationary reinforcement learning without prior knowledge: An optimal black-box approach." Conference on Learning Theory. PMLR, 2021.
> ```

---

> > ### Comment · Reviewer_bZZ8 · 2022-08-09
> > **Reviewer response Part 2**
> >
> > Thank you for the clarifications and taking my suggestions into consideration.
> >
> > I have a couple of questions.
> > 1. The MDP model in the paper uses a sequence of transition matrices and rewards that change along an episode, which is why they're indexed by $h\in[H].$ This, by definition is a non stationary process.
> > * My question was about justifying the use of this model. The same sequence of transition matrices and reward functions appears in _every_ episode. I would like to see some independent justification of this model, i.e., more than the claim that "others use it." Where in practice does one encounter this model (or something that at least approximates it)?
> > 2. I am not convinced with the explanation about technical and algorithmic novelty.
> >  ---
> > Post rebuttal, the reviewer would like to maintain their rating at 4.

---

> > > ### Author Response · Authors · 2022-08-09
> > > **Further Response**
> > >
> > > Thank you very much for your response! Please find our answers to your questions below:
> > > - **Non-stationary MDP**: In RL theory, **non-stationary MDP** means that transition probability $P$ and reward function $r$ change from episodes to episodes, which is a notion of non-stationarity slightly different from the standard stochastic process literature. In our paper, the model we use is called **time-inhomogeneous** MDP, which is more general than the **time-homogeneous** MDP (meaning that $P$ and $r$ remain fixed for all time steps), which is usually the standard textbook setting such as [Sutton et al. (2018)]. Therefore, our bounds still hold for the time-homogeneous MDP setting. In recent years, RL theory community is interested in time-inhomogenoeus MDP because of its generality.
> > > - **Real-world model**: We believe the Atari games, widely used as benchmarks for many practical algorithms, is a family of examples where "the same sequence of transition matrices and reward functions appears in every episode". At every time the game restarts (an episode ends), the agent will face the same game setting (same transition matrices and reward functions).
> > > - **Technical novelty:** We are happy to explain our novelties in further details if you have more specific questions.
> > >
> > > ```
> > > [Sutton et al. (2018)] Sutton, Richard S., and Andrew G. Barto. Reinforcement learning: An introduction. MIT press, 2018.
> > > ```

---

> ### Author Response · Authors · 2022-08-07
> **Any other questions to be addressed？**
>
> Thanks again for your review. We hope our answers could increase your confidence. As the discussion period is close to the end and we have not yet heard back from you, we would be glad to see if our rebuttal response has addressed your concerns questions/concerns.
> We are more than happy to discuss further if you have any further concerns and issues, please kindly let us know your feedback. Thank you for your time and help!

---

### Official Review · Reviewer_EVm9 · 2022-07-17

**Rating:** 7
**Confidence:** 4
**Soundness:** 3 good
**Presentation:** 4 excellent
**Contribution:** 3 good

**Summary:**

This paper studies episodic RL in tabular MDPs in the regret minimization setting, and presents an algorithm, called SSR, which follows the principle of exploration via randomized value functions. The paper presents two variants of SSR that differ in the choice of the noise. The variant that generates noise sequences based on Bernstein’s inequality is shown to achieve a worst-case regret bound (with high probability) that matches the lower bound up to logarithmic factors. This regret bound improves the best known regret bounds for randomized exploration by a factor of $\widetilde O(H\sqrt{S})$.

**Questions:**

See the weaknesses above. Also address minor comments if relevant.

**Limitations:**

There is no negative societal impact associated to this work. The most important limitations of the presented methods in the paper are adequately and clearly discussed in the paper.


**Strengths And Weaknesses:**

The paper studies a classic but important problem in RL. Despite all the recent advances in the theoretical aspects of episodic RL, the studied setup is interesting in view of the empirical success of exploration via randomized value function and considering that the theoretical picture is not yet complete. The paper makes a good step in this direction and thus makes a nice addition to the literature on theoretical episodic RL.

The paper is written well and admits a good organization. The presentation of the various elements (model, algorithm, and results) is clear and precise. All these makes the paper a nice read.

Amongst the technical and algorithmic novelties is the use of a single seed for randomization and the use of a new clipping strategy. The latter is shown to maintain both optimism and pessimism, which turns out to be crucial in achieving the improved regret bounds.

Some weaknesses and questions:

- The regret bound of the Bernstein-type variant in Theorem 2 is stated to hold when $T$ is larger than a (rather large) polynomial of $H$ and $S$, whereas this is not the case for the other variant. Could you elaborate further on this? And what can be said about regret for when $T$ is smaller than this threshold?

- I would like to appreciate the authors to have included the lower order terms in the regret bounds and to have clarified the time after which the desired $\sqrt{T}$-regime kicks in. Under both variants, the lower order terms scale as $H^4S^2A$, so that the desired $\sqrt{T}$-regime kicks in after a rather large $T$ (albeit polynomial in $H$ and $S$). I may urge the authors to compare SSR against state-of-the-art in terms of such critical values of $T$ as well.

- The statement ”UCB-type algorithms … suffer from difficult implementation ...” is not entirely correct. While this statement could be valid for model-based UCB-type algorithms, model-free ones do not require complicated planning procedures (such as EVI in UCRL2 or alike). Rather, they just take actions greedily with a Q-function, which is cheaply implementable. I may ask the authors to further elaborate on this, and if necessary, make the statement more precise.

- The paper nicely closes the gap between the regret lower and upper bounds, implying that for large enough time horizons, both UCB-type and randomized approaches could achieve the worst-case optimal regret bound. Then the following question naturally arises: In the considered episodic setting, which approach is empirically superior? Unfortunately, the paper does not provide any experimental result, so it remains inconclusive as to whether any of the two approaches provide a superior empirical performance.

Minor comments:

In several places, it is stated that optimism and pessimism is guaranteed to hold with constant probability. In technical sections (e.g., Section 4 and later), it might be a good idea to explicitly state such a constant probability to enrich the discussion.

Some typos:

Line 89: Algorithms …. has been ==> … have been

Line 130 (and elsewhere): in tabular setting ==> in the tabular setting

Line 158: $h=1, \ldots H$ ==> $h=1, \ldots, H$

Line 162: noisy observation ==> noisy observations

Line 178: has to large ==> has to be large

Line 188: in suboptimal regret bound ==> in a suboptimal regret bound

Line 192: number of clipping ==> … clippings

Line 194: is closed to ==> is close to

Line 198: uncertain-based ==> uncertainty-based

Line 212: … still maintain ==> still maintains

Line 219: Full stop missing

Line 221: noise ==> noises

------ Post Rebuttal ------

I would like to thank the authors for the rebuttal. My questions and concerns are adequately addressed. As a result, I maintain my score of 7, but with an increased confidence of 4.

---

> ### Author Response · Authors · 2022-08-01
> **Response**
>
> Thank you very much for your careful reading and constructive suggestions! We will fix the typos and address all concerns in more details in the final version. Please find our response to your questions and concerns below:
> - **Condition for $T$ in Theorem 2:** The requirement of $T\geq \Omega(H^5S^2A)$ in Theorem 2 comes from line 774 in the proof of Lemma 25. From a high-level perspective, this is needed because the Bernstein bonus uses the sum of variance to control the regret, which is more refined than the Hoeffding bonus but only when $T$ is large enough. That is, if $T\leq H^5S^2A$, then our analysis cannot guarantee the optimal order of $H$ in the leading term.
> - **Lower order term:** Our lower order term is the same as UCB-VI and we will discuss more on this in the final version. Meanwhile, it remains a major open problem in reinforcement learning theory to get rid of the lower order terms and some detailed discussion of this topic can be found in [31].
> - **Implementation of UCB-type algorithms:** We want to clarify that we believe UCB-type algorithms have difficulty in implementation mainly because of their non-trivial construction of confidence interval, which is true for both model-based and model-free algorithms. This is discussed in the paragraph starting from line 222 and we will try to make it more clear.
> - **Experiments:** Please refer to the experiments section at the top for all reviewers.

---

> ### Author Response · Authors · 2022-08-07
> **Any other questions to be addressed？**
>
> Thanks again for your review. We hope our answers could increase your confidence. As the discussion period is close to the end and we have not yet heard back from you, we would be glad to see if our rebuttal response has addressed your concerns questions/concerns.
> We are more than happy to discuss further if you have any further concerns and issues, please kindly let us know your feedback. Thank you for your time and help!

---

### Author Response · Authors · 2022-08-01
**Experiments Added**

We have fixed all the typos and added a section for experiments at **the end of the appendix**. In brief, we empirically compare RLSVI, UCBVI and our algorithm SSR in the deep sea environment, which is frequently used to test an algorithm's ability to do efficient exploration [35, 37]. We did not compare them with the PSRL [5, 6] because it is designed for different settings and objectives, which is not comparable to our algorithm.

In brief, SSR performs comparably as the UCBVI and significantly better than the RLSVI as suggested by our theory. More details can be found in our updated supplementary file.

---

### Meta-Review · Area_Chair_stfb · 2022-08-20

**Recommendation:** Accept
**Confidence:** Certain

**Metareview:**

Thank the authors for their submission.

The paper studies regret minimization in finite-horizon tabular Markov decision processes.
It is the first to show an optimal (up to logarithmic factors) regret bound of $\widetilde O(H \sqrt{|S| |A| T)}$ to Thompson sampling-type algorithms. A good addition to the TS literature, showing another case in which TS algorithms can have the same regret guarantees as optimistic algorithms. The paper is well-written.

**Award:**

No

---

### Decision · Program_Chairs · 2022-09-14

Accept